# Changing the rotational direction of a wind turbine under veering inflow: A parameter study

Antonia Englberger[1], Julie K. Lundquist[2,3], and Andreas Dörnbrack[1]

[1]German Aerospace Center, Institute of Atmospheric Physics, Oberpfaffenhofen, Germany
[2]Department of Atmospheric and Oceanic Sciences, University of Colorado Boulder, Boulder, USA
[3]National Renewable Energy Laboratory, Golden, Colorado, USA

**Correspondence:** Antonia Englberger (antonia.englberger@dlr.de)

**Abstract.** All current-day wind turbine blades rotate in clockwise direction as seen from an upstream perspective. The choice of the rotational direction impacts the wake if the wind profile changes direction with height. Here, we investigate the respective wakes for veering and backing winds in both hemispheres by means of large-eddy simulations. We quantify the sensitivity of the wake to the strength of the wind veer, the wind speed, and the rotational frequency of the rotor in the Northern Hemisphere. A veering wind in combination with counterclockwise rotating blades results in a larger streamwise velocity output, a larger spanwise wake width, and a larger wake deflection angle at the same downwind distance in comparison to a clockwise rotating turbine in the Northern Hemisphere. In the Southern Hemisphere, the same wake characteristics occur if the turbine rotates counterclockwise. These downwind differences in the wake result from the amplification or weakening/reversion of the spanwise wind component due to the effect of the superimposed vortex of the rotor rotation on the inflow's shear. An increase of the directional shear or the rotational frequency of the rotor under veering wind conditions increases the difference in the spanwise wake width and the wake deflection angle between clockwise and counterclockwise rotating actuators, whereas the wind speed lacks a significant impact.

# 1 Introduction

Most modern industrial-scale wind turbines rotate clockwise, as seen from a viewer looking downwind. Traditional Danish windmills turned counterclockwise (Maegaard et al., 2013), who preferred the thin end of the laths pointing towards the left on the blades as they were built by right-handed millers. This rotational direction was adapted by the wind-turbine pioneer Christian Riisager and subsequently by the company Tvind. In 1978, Erik Grove-Nielson designed the first 5-m fibreglass blades. He and his wife Tove chose a clockwise rotational direction of the blades to distinguish their product from Tvind. Descendants of the Riisager wind turbine (Wind Matic and Tellus) rotate counterclockwise while those of Grove-Nielson (Vestas, Bonus (now Siemens), Nordtank and Enercon) rotate clockwise. Three of the four clockwise rotating blade manufacturers became market leaders in the international wind power industry, and the clockwise rotating blades, eventually, became the global standard (Maegaard et al., 2013). The clockwise blade rotation is, therefore, a historical coincidence without physical motivation.

Rotating blades encounter a variety of wind conditions. In a convective regime during daytime above the surface layer, there is no significant change of the incoming wind direction or wind speed with height and the inflow conditions are uniform over the whole rotor area. A nocturnal stably stratified regime, however, often generates wind profiles with changing magnitude (vertical wind shear) and direction (wind veer) (Lindvall and Svensson, 2019). Vertical variations of both quantities reflect the balance between Coriolis force and friction. Friction affects the lowest part of the wind profile and contributes as internal friction in the flow while the rotational direction of the wind vector in the Ekman spiral aloft depends on the hemisphere. In the Northern Hemisphere (NH) (Southern Hemisphere (SH)), winds tend to rotate clockwise (counterclockwise) with height (Stull, 1988). Veer occurs on many nights both onshore (Walter et al., 2009; Rhodes and Lundquist, 2013; Sanchez Gomez and Lundquist, 2020) and offshore (Bodini et al., 2020, 2019). According two years of meteorological tower measurements in Lubbock (Texas) (Walter et al., 2009) and three months of lidar observations in north-central Iowa (Sanchez Gomez and Lundquist, 2020), veer occurs in well over 70% of those SBL occurrences ($\approx$ 76% in Walter et al. (2009) and $\approx$ 78% in Sanchez Gomez and Lundquist (2020)). In the remaining 22% (Sanchez Gomez and Lundquist, 2020) to 24% (Walter et al., 2009), a backing wind occurs. A backing wind is characterized by a counterclockwise wind direction change with height in the NH.

The frequency of occurrence of a veering wind depends on many criteria. A wind direction change with height occurs mainly at night. Secondly, seasonal differences occur. Thirdly, the frequency of occurrence of veering or backing is location specific. In Lubbock (Texas) (Walter et al., 2009) and in north-central Iowa (Sanchez Gomez and Lundquist, 2020), a veering wind occurs in three out of four nights. In their global climatology of veer based on radiosonde data, Lindvall and Svensson (2019) find stronger veer (or backing for the SH) in midlatitudes (see their Fig. 3). Of course, topography can change the frequency of occurrence significantly. Each location has its own percentage values of the occurrence of a veering wind. In this study, a directional shear of $0.08°$ m$^{-1}$ is applied in the reference case, as it corresponds to the mean of the frequency of occurrence of a veering inflow in Walter et al. (2009). Further, seasonal dependence occurs (Bodini et al., 2019, 2020). The longer lasting nights during winter are characterised by smaller mean values of the directional shear (minimum winter values in December of $0.03°$ m$^{-1}$ (Bodini et al., 2020, Fig. 4) according to 13 months of offshore lidar measurements in Massachusetts). The

shorter nights during summer, however, are characterised by larger mean values of the directional shear (maximum summer values in June of $0.095°$ m$^{-1}$ (Bodini et al., 2020, Fig. 4)). The occurrence of a specific directional shear (e.g. $ds = 0.08°$ m$^{-1}$) for a specific wind speed (in between 9 m s$^{-1}$ and 11 m s$^{-1}$, similar to the reference case used here) is much larger during winter (85% of the veering cases are characterized by $ds \leq 0.08°$ m$^{-1}$) in comparison to the summer (45% (Bodini et al., 2020,

Fig. 5)).

The wind turbine's wake characteristics in a veering wind regime differ for counterclockwise and clockwise rotating blades (Englberger et al., 2019). The induced vortex component of the near wake's flow is determined by the rotation of the blades. The wake rotates opposite to the blade rotation due to aerodynamics and design of the wind-turbine blades (Zhang et al., 2012). In contrast, the rotational direction of the far wake is determined by the Ekman spiral. If a northern hemispheric Ekman

spiral interacts with clockwise rotating blades, the spanwise flow component in the wake weakens or even reverses due to a superposition with the vortex of the near wake and attenuates to the inflow in the far wake. In this case, the near wake's counterclockwise rotation diminishes and becomes clockwise. After this reversion or likewise in the case of a reduction of the spanwise wake component, the wake's rotation strength intensifies downwind. Conversely, if the same inflow interacts with counterclockwise rotating blades, the spanwise flow component is amplified in the near wake, because the rotational direction

persists in the whole wake and the wake's rotation strength weakens downwind.

The modification of the spanwise flow component also impacts the streamwise velocity in the wake. It affects the velocity deficit in the near wake, the streamwise wake elongation of the wake, the spanwise wake width, and the deflection angle of the wake (Englberger et al., 2019). There also exists a rotational direction impact on the streamwise velocity component in a flow regime without wind veer (Vermeer et al., 2003; Shen et al., 2007; Sanderse, 2009; Kumar et al., 2013; Hu et al., 2013; Yuan

et al., 2014; Mühle et al., 2017). Vasel-Be-Hagh and Archer (2017) simulated a wind farm similar to Lillegrund in Sweden with alternative rotational direction of the rotors, starting with clockwise in the first row. Including wind shear but no wind veer in the inflow conditions, the power output was 1.4% larger in comparison to only clockwise rotating rotors in the wind farm. However, compared to the wake differences for clockwise and counterclockwise rotating rotors in a flow regime with wind veer, the differences are small in the case of no wind veer (Englberger et al., 2019). Therefore, the 1.4% in Vasel-Be-Hagh and

Archer (2017) can be considered as a lower limit, the consideration of veer amplifies this difference.

In this study, we investigate the relationship between the upstream wind profile and the direction of the turbine rotation using large-eddy simulations (LESs). Both clockwise and counterclockwise rotating actuators are embedded in a veering as well as a backing inflow for both hemispheres. In the case of a veering inflow in the NH, we carry out a parameter study investigating the impact of the magnitude of the geostrophic wind, the directional shear, and the rotational frequency of the rotor. The results of

the rotational direction impact on the wake are interpreted for all simulations with a theoretical analysis considering a Rankine vortex representation of the wake. To our knowledge, this is the first parameter study which investigates the interactions of wake rotational direction in combination with an Ekman spiral.

Our previous study (Englberger et al., 2019) lays the groundwork for this investigation, describing in detail the rotational direction impact in a stably-stratified regime under veered inflow conditions and in an evening boundary layer regime under

non-veered conditions (in the Northern Hemisphere). Further, that work explains the physical mechanism responsible for the

rotational direction impact of the blades on the wake by simple analysis of a linear superposition of the veering inflow wind field with a Rankine vortex.

This paper is organised as follows. The numerical model EULAG, the wind-turbine simulation setup, and the metrics applied in this work are described in Sect. 2. The analysis predictions are introduced in Sect. 3. The corresponding idealized simulations investigating the rotational direction impact on the wake follow in Sect. 4. A comparison of the simulation results to the analysis predictions is given in Sect. 5 and a conclusion follows in Sect. 6.

## 2 Numerical Model Framework

### 2.1 The Numerical Model EULAG

The wind-turbine simulations, with prescribed wind and turbulence conditions, are conducted with the flow solver EULAG (Prusa et al., 2008). For a comprehensive description and discussion of EULAG we refer to Smolarkiewicz and Margolin (1998) and Prusa et al. (2008).

The Boussinesq equations for a flow with constant density $\rho_0 = 1.1\ \text{kg m}^{-3}$ are solved for the Cartesian velocity components $u, v, w$ and for the potential temperature perturbations $\Theta' = \Theta - \Theta_e$ (Smolarkiewicz et al., 2007),

$$\frac{d\mathbf{v}}{dt} = -\boldsymbol{\nabla}\left(\frac{p'}{\rho_0}\right) + \mathbf{g}\frac{\Theta'}{\Theta_0} + \boldsymbol{\mathcal{V}} + \boldsymbol{\beta}_\mathbf{v}\frac{\mathbf{F}_{WT}}{\rho_0}, \tag{1}$$

$$\frac{d\Theta'}{dt} = \mathcal{H} - \mathbf{v}\nabla\Theta_f, \tag{2}$$

$$\boldsymbol{\nabla} \cdot (\rho_0 \mathbf{v}) = 0, \tag{3}$$

with $\Theta_e$ representing the environmental/background state and $\Theta_0$ representing the constant reference value of 300 K. In Eqs. (1), (2) and (3), $d/dt$, $\boldsymbol{\nabla}$ and $\boldsymbol{\nabla} \cdot$ represent the total derivative, the gradient and the divergence, respectively. The quantity $p'$ represents the pressure perturbation with respect to the environmental state. Further, $\mathbf{g}$ represents the vector of acceleration due to gravity. The subgrid-scale terms $\boldsymbol{\mathcal{V}}$ and $\mathcal{H}$ symbolise viscous dissipation of momentum and diffusion of heat. $\mathbf{F}_{WT}$ corresponds to the turbine-induced force and $\boldsymbol{\beta_v}$ to the rotational direction. All following simulations are performed without an explicit subgrid-scale closure as implicit LES (Grinstein et al., 2007), to remove any question of the influence of the subgrid-scale closure on the results. Further, we apply a free-slip vertical boundary condition.

The turbine-induced forces ($\mathbf{F}_{WT}$) in Eq. (1) are parametrized with the blade element momentum (BEM) method as actuator disc, including a nacelle and excluding the tower. The BEM method enables the calculation of the steady loads, thrust, and power for different wind speeds and rotational speeds of the blades. The airfoil data of the 10 MW reference wind turbine from DTU (Bak et al., 2013) are applied, whereas the radius of the rotor as well as the chord length of the blades are scaled to a rotor with a diameter of 100 m. For a more detailed description of the wind-turbine parametrization and all values used in the wind-turbine parametrization we refer to parametrization B of Englberger and Dörnbrack (2017).

The actuator disc rotates in clockwise or counterclockwise direction, depending on the choice of $\boldsymbol{\beta_v} \in \{-1, 1\}$. The rotor rotation is not directly simulated, instead, the rotor forces are exerted directly on the velocity fields in Eq. (1). A clockwise rotating rotor initiates a counterclockwise wake rotation and vice versa, following conservation of angular momentum (Zhang et al., 2012). In this work, a common clockwise rotor rotation 'cr' is defined as $\beta_v = 1$ and $\beta_w = -1$ and a counterclockwise rotor rotation 'ccr' as $\beta_v = -1$ and $\beta_w = 1$, with $\beta_u = 1$ in both cases.

## 2.2 Setup of the Wind-Turbine Simulations

Wind-turbine simulations on $512 \times 64 \times 64$ grid points with a horizontal and vertical resolution of 5 m and open boundaries are performed for veering and non-veering inflow lasting 40 min. The rotor of the wind turbine has a diameter $D$ as well as a hub height $z_h$ of 100 m and is located at 300 m downwind from the inflow boundary and centred in the spanwise $y$-direction.

A veering wind profile can be described by the Ekman spiral

$$u_{Ekman}(z) = u_g \cdot (1 - \exp(-z\gamma)\cos(z\gamma)), \tag{4}$$

$$v_{Ekman}(z) = u_g \cdot (\exp(-z\gamma)\sin(z\gamma)), \tag{5}$$

following Stull (1988), with a geostrophic wind $u_g$ and

$$\gamma = \sqrt{\frac{f}{2\kappa}}$$

representing a Coriolis parameter $f = 1.0 \times 10^{-4}$ s$^{-1}$ and an eddy viscosity coefficient $\kappa$.

Wind direction change between two heights is defined as directional shear. As we assume the directional shear to be an impact factor for the interaction process of a rotating system with veering inflow, a modified version of the Ekman spiral is applied as $v_f$ in the simulations in this work. Further, the negative vertical gradient of the streamwise velocity in the supergeostropic component of the Ekman spiral is not considered in $u_f$.

The simulations are initialized with the streamwise velocity profile

$$u_f(z) = u_g * (1 - \exp(-z\gamma)), \tag{6}$$

with an eddy viscosity coefficient $\kappa = 0.06$ m$^2$ s$^{-1}$. The corresponding spanwise velocity profile is

$$v_f(z) = 0 \tag{7}$$

in the case of no veering inflow with $\frac{\partial v_f}{\partial z} = 0$, and

$$v_f(z) = u_f(z) \cdot \tan(\phi_{wind}(z)) \tag{8}$$

in the case of veering inflow with $\frac{\partial v_f}{\partial z} \neq 0$ with a given directional shear

$$ds = \frac{\Delta\phi}{100\ m}, \tag{9}$$

with $\Delta\phi = \phi_{150\,m} - \phi_{50\,m}$ and

$$\phi(z) = \pm 2\Delta\phi\left(1 - \frac{z}{D}\right) \tag{10}$$

in the lowest 200 m and constant above. The influence of the Coriolis force on the flow field is only included in the simulations via Eqs. 6, 8. Note that no Coriolis force is applied in the numerical model (Eq. 1).

For $u_f$ and $v_f$, we consider the NH ($f > 0$) and the SH ($f < 0$), a veering ($\Delta\phi > 0$ in NH, $\Delta\phi < 0$ in SH), and a backing ($\Delta\phi < 0$ in NH, $\Delta\phi > 0$ in SH) wind. In the reference simulation (with a veering wind in the NH), the directional shear is $0.08^\circ\,\mathrm{m}^{-1}$ with $v_f(z_h) = 0$. The initial vertical velocity is

$$w_f(z) = 0 \tag{11}$$

in all simulations. The flow components $u_f$, $v_f$, and $w_f$ are used for specifying the initial conditions. The pressure solver in
EULAG further applies $u_f$ and $v_f$ as boundary conditions. The potential temperature is

$$\Theta_e(z) = \Theta_0 + \frac{3\,K}{200\,m}z \tag{12}$$

in the lowest 200 m and 303 K above.

For a veering wind in the NH, we modify the geostrophic wind, the directional shear and the rotational frequency of the rotor. We sample winds with a geostrophic wind component of $u_g = 6\,\mathrm{m\,s}^{-1}$, $u_g = 10\,\mathrm{m\,s}^{-1}$ (reference simulation), and $u_g = 14\,\mathrm{m\,s}^{-1}$,
referred to with the acronyms u6 and u14 in the simulation nomenclature. Further, we apply a directional shear of $0.04^\circ\,\mathrm{m}^{-1}$, $0.08^\circ\,\mathrm{m}^{-1}$, $0.12^\circ\,\mathrm{m}^{-1}$, $0.16^\circ\,\mathrm{m}^{-1}$, and $0.20^\circ\,\mathrm{m}^{-1}$ corresponding to weak (ds4), moderate (reference simulation), moderate to strong (ds12), strong (ds16) and very strong (ds20) shear. As additional parameter, the rotational frequency ranges from $\Omega = 0.058\,\mathrm{s}^{-1}$, $\Omega = 0.12\,\mathrm{s}^{-1}$, $\Omega = 0.175\,\mathrm{s}^{-1}$, to $\Omega = 0.23\,\mathrm{s}^{-1}$, corresponding to low ($\Omega l$), moderate (reference simulation), high ($\Omega h$), and very high ($\Omega vh$) in the simulation nomenclature.

This work is a parameter study investigating the impacts of the inflow (directional shear, wind speed) and the rotating system (rotational frequency) on the wake. The wake's impact by the rotational direction of the rotor depends on the mean wind profile, which is determined by the geostrophic wind (Eqs. 6, 8) and the directional shear (Eq. 9). Turbulence modifies the strength of the wakes, but not the occurrence (Appendix and Section 5). Therefore, we perform the simulations as implicit LES with no explicit subgrid-scale closure model. Moreover, we apply the turbulence parametrization by Englberger and
Dörnbrack (2018b) to perturb the flow field during the numerical integration. This turbulence parametrization provides a computationally fast method for wind-turbine simulations with open horizontal boundary conditions on a small domain. It includes stability-dependent atmospheric characteristics in the inflow. This makes the method very suitable for parameter studies. We superimpose upon the inflow wind field turbulent fluctuations of a neutral boundary layer precursor simulation (Englberger and Dörnbrack, 2017), as represented by term I in Eq. 13, where $\mathbf{u}_p\big|_{i^*,j,k}$ is the velocity vector of a neutral
boundary layer equilibrium state at each grid point $i$, $j$, and $k$.

$$\mathbf{u}_p^*\big|_{i=1,j,k}^{\delta} = \alpha_0 \cdot \boldsymbol{\alpha}_{i^*,j,k} \cdot \underbrace{\left(\mathbf{u}_p\big|_{i^*,j,k} - \frac{1}{n\cdot m}\sum_{i=1}^{n}\sum_{j=1}^{m}\mathbf{u}_p\big|_{i,j,k}\right)}_{I}. \tag{13}$$

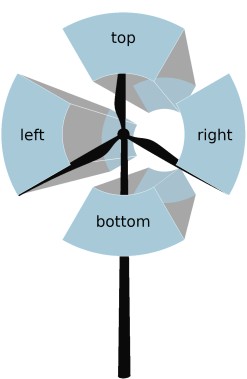

**Figure 1.** Schematic illustration of the top sector, the bottom sector, as well as the left and right sectors, defined from a view looking downwind towards the wind turbine on the disc.

The indices of the grid points are denoted by $i = 1 \ldots n$, $j = 1 \ldots m$, and $k = 1 \ldots l$ in the $x$, $y$, and $z$ directions, respectively. The star refers to a streamwise shift by one grid point at every time step $\delta$ with $i^* = i + \delta$, whereas $i^* = [1, n]$ and $\delta^*$ represents the passed number of timesteps. The prefactor $\alpha_0$ represents the amplitude of the turbulence perturbations and $\boldsymbol{\alpha}_{i^*, j, k}$ represents adjustable stratification-dependent parameters for convective and stable regimes as well as the transitions between them.

5 The stratification-dependent parameters were retrieved from a 30-h diurnal cycle simulation from Englberger and Dörnbrack (2018a).

In the following simulations we apply a nighttime representations using values of $\alpha = 0.3$, $\alpha_u = 0.15$, $\alpha_v = 0.24$, and $\alpha_w = 0.13$ (Englberger and Dörnbrack, 2018b, Table 1). A rather similar set-up including the turbulence parametrization has been applied in Englberger and Lundquist (2020).

10 ## 2.3 Metrics

For the investigation of the rotational direction impact on the wake, the following characteristics are calculated from the simulation results: the spatial distribution of the time-averaged discrete streamwise velocity $\overline{u_{i,j,k}}$, the time-averaged discrete spanwise velocity $\overline{v_{i,j,k}}$, and the streamwise velocity deficit

$$VD_{i,j,k} \equiv \frac{\overline{u_{1,j,k}} - \overline{u_{i,j,k}}}{\overline{u_{1,j,k}}}. \tag{14}$$

15 The characteristics are averaged over the last 30 min of the 40-min wind-turbine simulation. The 30-min temporal average is calculated online in the numerical model according to the method of Fröhlich (2006, Eq. 9.1).

In the following, the quantities $\overline{u_{i,j,k}}$ and $\overline{v_{i,j,k}}$ are evaluated and discussed for top and bottom sectors. They result from a division of the rotor area into four sections of $90°$, as shown in Fig. 1, including all grid points with a distance $r$ from the rotor center $0\,\mathrm{m} < \mathrm{r} \leq \mathrm{R}$. The left and right sectors are defined from a view looking downwind towards the wind turbine on the disc.

## 3 Theoretical analysis

In this section, we construct a simple analytical model of the interaction of the rotating wake of a wind turbine with a sheared ambient flow. The rotating wake is prescribed by a Rankine vortex, whereas the ambient flow is described by three different inflow conditions (no veer, veering wind, backing wind). In the case of a veering inflow, the relations are also evaluated for three different parameters (wind speed, directional shear, rotational velocity). The approach follows Englberger et al. (2019) and is modified to allow different directional shear values.

A rotating system can be described by a Rankine vortex with the radial dependence $r$, the rotational velocity $\omega$, and the angle $\vartheta$ in a $y$-$z$ plane:

$$v_v(z) = \pm \omega r \sin(\vartheta) \tag{15}$$

$$w_v(z) = \mp \omega r \cos(\vartheta) \tag{16}$$

The veering inflow is described by Eqs. 4 and 5, whereas no wind veer is described by Eq. 7. In this analysis we apply the simplified Eqs. 6 and 8 for the veering inflow, as they allow a variety of directional shear values. Both inflow cases result in a superposition of the spanwise components $v_f$ (Eqs. 7, 8) and $v_v$ (Eq. 15) in Eq. 17:

$$v(z, x_{down}) = \begin{cases} \text{no veer:} & v_v \cdot (1 - \dfrac{x_{down}}{x_\zeta}) \\ & = \pm \omega r \sin(\vartheta)(1 - \dfrac{x_{down}}{x_\zeta}) \\ \text{veer:} & v_f + v_v \cdot (1 - \dfrac{x_{down}}{x_\zeta}) \\ & = u_g \cdot \exp(-z\gamma) \cdot \tan\left(2\Delta\phi\left(1 - \dfrac{z}{D}\right)\right) \pm \omega r \sin(\vartheta)(1 - \dfrac{x_{down}}{x_\zeta}) \end{cases} \quad x_{WT} \leq x_{down} \leq x_\xi \tag{17}$$

In Eq. 17, a linear decrease of $v_v(z, x_{down})$ is assumed for a given downwind distance $x_{down}$ from the rotating system $x_{WT}$ up to $x_\xi$ with $v(z, x_\xi) = 0$. In this work, we only consider the spanwise flow component, as $w_f = 0$ (Eq. 11).

Figure 2 represents the spanwise velocity component $v$ resulting from Eq. 17 at $z = 125$ m and at $z = 75$ m at the rotor center in lateral direction. The rotating system has a rotor center $z_h = 100$ m and a rotor radius $R = 50$ m. The vertical positions are centered in the top and the bottom sectors of Fig. 1.

The wake resulting from a clockwise (cr) or counterclockwise (ccr) rotating rotor interacting with no wind veer are represented in Fig. 2(b). Following Eq. 17, the rotational direction of the rotor determines the sign of $v(z, x_{down})$. Therefore, the spanwise velocity component has the opposite sign in the top and the bottom rotor part of both cr and ccr in Fig. 2(b). Approaching $x_\xi$, $v(z, x_\xi) = v_f(z) = 0$.

In the case of veering inflow, however, the spanwise flow component impacts the wake (Eq. 17). The spanwise flow component results from the Ekman spiral, which is hemispheric dependent. In the NH, $f > 0$ and, therefore, the spanwise flow component $v_f(z_h - R/2) > 0$ in the lower rotor half and $v_f(z_h + R/2) < 0$ in the upper rotor half with $v_f(z_h) = 0$ (Eq. 8). This situation corresponds to a flow from right to left in the lower rotor half and from left to right in the upper rotor half, looking from upwind towards downwind (Fig. 1). If '+' is applied in Eq. 17, $v_v(z_h - R/2, x_{down}) = +\omega r \sin(270°) = -\omega r < 0$ in the

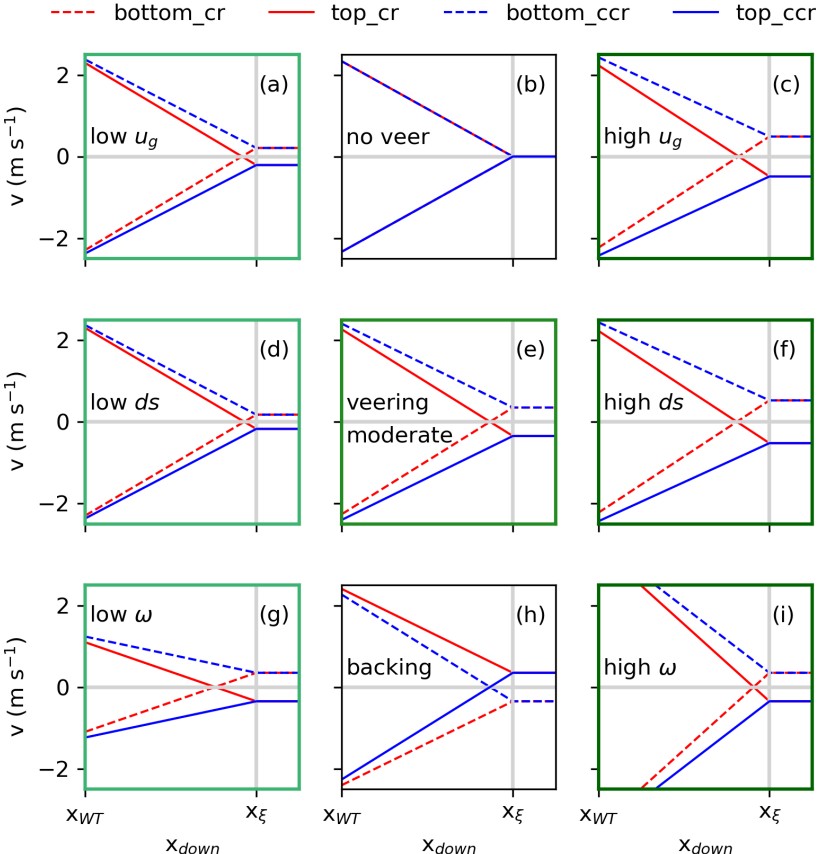

**Figure 2.** Representations of $v(z = 75m, x_{down})$ at the rotor center in lateral position as bottom and $v(z = 125m, x_{down})$ as top (Eq. 17) for a clockwise (cr) and a counterclockwise (ccr) rotating rotor in the case of no veer in (b), a veering wind in (e), and a backing wind in (h). In the case of a veering wind in (e), moderate parameters of $u_g = 10$ m s$^{-1}$, $ds = 0.08°$ m$^{-1}$, and $\omega = 0.12°$ s$^{-1}$ are applied. In the left and right column, only one parameter is changed compared to the veering wind situation in (e). Applying low parameters $u_g = 6$ m s$^{-1}$ in (a), $ds = 0.04°$ m$^{-1}$ in (d), and $\omega = 0.058°$ s$^{-1}$ in (g), and applying high parameters $u_g = 14$ m s$^{-1}$ in (c), $ds = 0.12°$ m$^{-1}$ in (f), and $\omega = 0.175°$ s$^{-1}$ in (i).

lower rotor half and $v_v(z_h + R/2, x_{down}) = +\omega r sin(90°) = \omega r > 0$ in the upper rotor half. However, if '-' is applied in Eq. 17, $v_v(z_h - R/2, x_{down}) = -\omega r sin(270°) = \omega r > 0$ in the lower rotor half and $v_v(z_h + R/2, x_{down}) = -\omega r sin(90°) = -\omega r < 0$ in the upper rotor half. The sign '+' corresponds to a counterclockwise wake rotation which arises from a clockwise rotor rotation (Zhang et al., 2012), whereas the sign '-' corresponds to a clockwise wake rotation arising from a counterclockwise rotor rotation ccr.

In the case of a clockwise rotating rotor, the rotor competes against the veer effect with $v_v(z_h - R/2, x_{down}) < 0$ superpositioning $v_f(z_h - R/2) > 0$ and $v_v(z_h + R/2, x_{down}) > 0$ superpositioning $v_f(z_h + R/2) < 0$. In both the top and bottom half of the rotor, the spanwise component of the inflow $v_f$ is weakened by the vortex component $v_v$ or even reversed, if $|v_v| > |v_f|$. Approaching downwind, the impact of $v_v$ decreases and $v(z, x_{down})$ approaches $v_f(z)$ at $x_{down} = x_\xi$ with $v_v(z, x_\xi) = 0$.

In the case of a counterclockwise rotating rotor, the wake vortex intensifies the inflow with $v_v(z_h - R/2, x_{down}) > 0$ superpositioning $v_f(z_h - R/2) > 0$ and $v_v(z_h + R/2, x_{down}) < 0$ superpositioning $v_f(z_h + R/2) < 0$. The vortex intensifies the inflow $v_f(z_h)$ in all rotor heights. Approaching downwind, the impact of $v_v$ decreases and $v(z, x_{down})$ approaches $v_f(z)$ at $x_{down} = x_\xi$ with $v_v(z, x_\xi) = 0$. At $x_{down} = x_\xi$, the situation is independent of the vortex and the wake has completely recovered.

The different behaviour of the spanwise wake component is presented in Fig. 2(e). In the case of cr, the vortex component

weakens the spanwise inflow component, resulting in a reversion of the sign of $v(z, x_{down})$ behind the rotor at $x_{down} < x_\xi$. In the case of ccr, however, the vortex component intensifies the spanwise inflow component. At $x_\xi$, both rotational directions show the same result, approaching towards the inflow conditions.

Figure 2(h) represents the situation for a backing wind. Only $\phi(z)$ (Eq. 10) and, therefore, the flow component $v_f(z)$ (Eq. 8) changes sign in both the top and bottom half of the rotor. The vortex component $v_v(z \pm R/2, x_{down})$ is not inflow dependent.

Therefore, the wake behaviour of ccr (cr) in the case of backing wind is comparable to cr (ccr) under veering inflow, resulting in a decrease (intensification) of $v_f(z)$ in the wake, following Eq. 17 with a '-' in Eq. 10.

This analysis shows a rotational direction dependent downwind behaviour of the spanwise flow component in the case of $\frac{\partial v_f}{\partial z} \neq 0$. The superposition of the Rankine vortex with a veering inflow (and likewise the backing wind) has three impacts. The veering inflow is determined by the geostrophic wind $u_g$ and the directional shear $ds$ over the rotor height. The vortex

component is determined by the rotational velocity $\omega$ of the rotor. The impact of $u_g$, $ds$ and $\omega$ on the expected mean behaviour of the spanwise wake component is presented in Fig. 2 for low values of the parameters in the left row and high values in the right row, whereas Fig. 2(e) represents the veering case for moderate parameter values.

A decrease of $u_g$ (Fig. 2(a)) or $ds$ (Fig. 2(d)) and likewise an increase of $u_g$ (Fig. 2(c)) or $ds$ (Fig. 2(f)) impacts the mean value of the spanwise wake field. Decreasing the atmospheric parameter values (Fig. 2(a), (d)), the values of $v_f(z_h \pm R/2)$

also decrease, leading to a downwind shift of the sign-changing point of $v(z, x_{down}) = 0$ (compare Fig. 2(a), (d) to (e)). A further decrease of $v_f(z \pm R/2)$ approaching $v_f(z \pm R/2) = 0$ of the non-veering inflow case, results in Fig. 2(b). An increase of $u_g$ (Fig. 2(c)) or $ds$ (Fig. 2(f)) results in an increase of $v_f(z \pm R/2)$ and an upward shift of the sign-changing point. If the atmospheric parameter values increase (decrease), the difference in the slope of the spanwise component between cr and ccr also increases (Fig. 2(c), (f)) (decreases (Fig. 2(a), (d))). Likewise, the slope in the case of cr increases for high values

(Fig. 2(c), (f) in comparison to low values (Fig. 2(a), (d)), whereas in the case of ccr, the slope decreases for larger values of the inflow parameters (Fig. 2(c), (f) vs. (e) vs. (a), (d)). This behaviour can be interpreted as an increase of the difference in the wake between cr and ccr if the atmospheric parameters increase.

The rotational velocity $\omega$ controls the magnitude of the spanwise vortex component. A decrease of $\omega$ (Fig. 2(g)) and likewise an increase (Fig. 2(i)) also influences the mean value of the spanwise wake field, especially in the near wake. A decrease of

$\omega$ decreases $v(z, x_{down})$ directly behind the rotor (Fig. 2(g)), whereas an increase results in an increase of $v(z, x_{down})$ in the near wake (Fig. 2(i)). Larger values of $\omega$ lead to a less rapid wake recovery in the near wake.

As a veering wind in the NH is comparable to a backing wind in the SH (following the definition via Eq. 10), all panels in Fig. 2 are also valid for the SH with red lines representing ccr_SH and blue lines representing cr_SH and dashed lines referring
to the top rotor part and solid lines to the bottom rotor part.

## 4   Idealized simulations: Rotational Direction Impact on the Wake

### 4.1   Veering vs. No Veering Inflow

The analysis of the preceding section predicts a rotational direction impact on the spanwise velocity component $v$ (and likewise the vertical component $w$) in the wake under veering (or backing) inflow, whereas the wake characteristics in the case of no
veer are independent of the rotational direction of the rotor. This rotational direction impact is investigated by LESs with veering and no veering inflow, with the simulation CR, CCR, CR_NV, and CCR_NV conducted with the parameters as listed in Table 1. The interactions between the wake rotation and the inflow are embodied in Fig. 3 in the crossstream and vertical velocities at $x = 3\,\mathrm{D}$. The first two columns represent CR and CCR in the case of veering inflow, whereas the last two columns correspond to no wind veer in the incoming flow field. The top row (Fig. 3(a) - (d)) represents the vectors $(v, w)$. The evolution
of $v$ and $w$ is represented in the second row ((e) - (h)) for $v$ and in the third row ((i) - (l)) for $w$. In the case of no wind veer, the sign of $v$ is opposite in the upper and the lower rotor half for CR_NV and CCR_NV (Fig. 3(g), (h)), as predicted by the analysis (Eq. 17). The same is valid for the sign of $w$ (Fig. 3(k), (l)). The numerical model shows a clockwise rotating wake in the case of CCR_NV (while looking from upwind towards downwind (Fig. 1)), and a counterclockwise rotating wake in the case of CR_NV.
Under veering inflow, the simulated wake rotates clockwise in the case of CCR (Fig. 3(b)) and counterclockwise in the case of CR (Fig. 3(a)), similar to the no veer case (Fig. 3(d), (c)). However, in comparison to the no veer case, the strength of rotation differs and is much more pronounced in the case of CCR (Fig. 3(b)) in comparison to CR (Fig. 3(a)). This rotation arises from the spanwise velocity component, as the vertical velocity (Fig. 3(i), (j)) is comparable to the non-veering cases (Fig. 3(k), (l)). The positive and negative perturbations in $v$ have the same positive and negative patterns in CR (Fig. 3(e) vs. (g)) and CCR
(Fig. 3(f) vs. (h)) as in CR_NV and CCR_NV in the corresponding rotor sector at $x = 3\,\mathrm{D}$, however, with smaller $|\,v\,|$ values in the upper and lower rotor sector in the case of CR and larger $|\,v\,|$ values in the case of CCR. This simulated amplification of the spanwise flow component in the case of CCR (Fig 3(f)) and weakening up to a reversion of the sign in the wake region at $x = 3\,\mathrm{D}$ in the case of CR (Fig. 3(e)) is in agreement with the predictions of the analysis (Eq. 17) and Fig. 2.

A downwind distance of $x = 3\,\mathrm{D}$ is visualized in Fig. 3 because a significant spanwise vortex impact on the spanwise flow
component in the wake can be expected. In the following, special emphasis is placed at $x = 7\,\mathrm{D}$, which is often considered a typical downwind distance for a hypothetical waked wind turbine in numerical simulation studies (e.g. Gaumond et al. (2014); Abkar et al. (2016)). At $x = 7\,\mathrm{D}$, the vortex impact is much smaller compared to $x = 3\,\mathrm{D}$ (Fig. 3), resulting in an increase of the impact of the atmospheric flow.

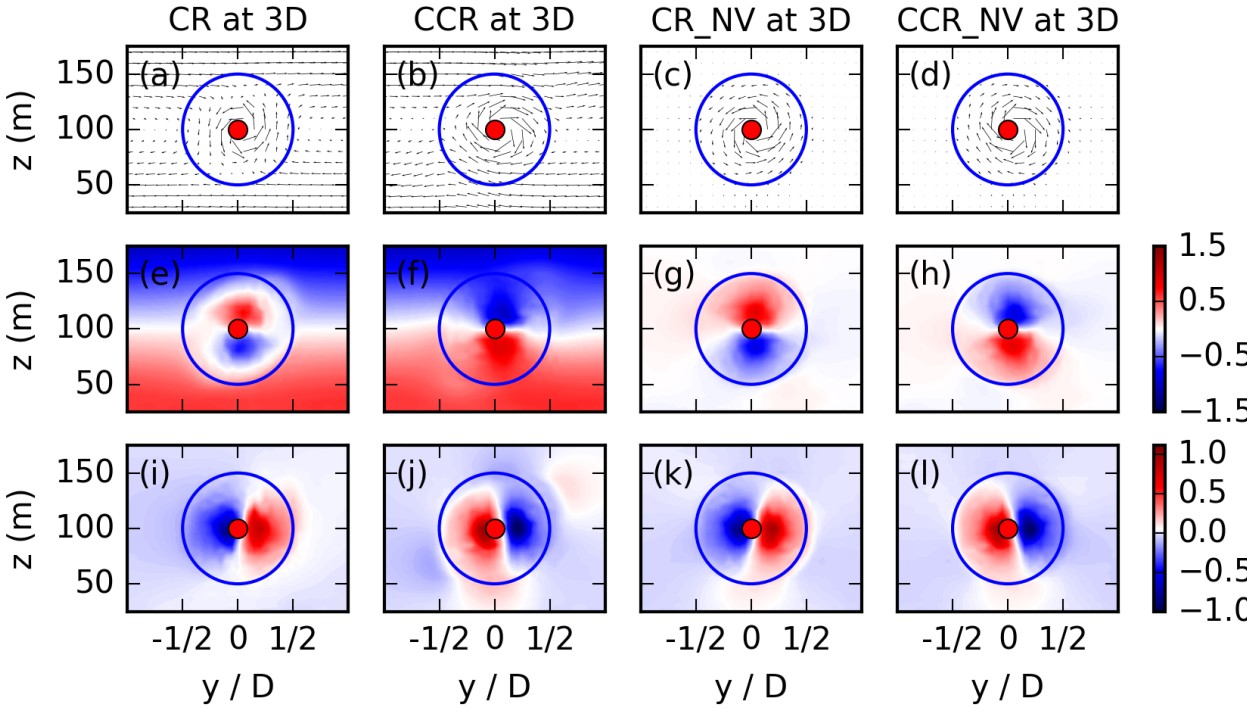

**Figure 3.** $y$-$z$-cross sections for veering and no veering (NV) inflow simulations at $x = 3$ D for CR (first column), CCR (second column), CR_NV (third column), and CCR_NV (last column). The first row ((a)-(d)) presents the $(v, w)$ vectors in the $y$-$z$-plane, the second row ((e)-(h)) the spanwise wake velocity $v$, and the third row ((i)-(j)) the vertical wake velocity $w$. The blue circle represents the circumference of the actuator disc. This picture is looking from upwind towards downwind on the wake (Fig. 1, corresponding to the left sector for $y < 0$ D and the right sector for $y > 0$ D.

As the rotational direction has a significant impact on the spanwise flow component at $x = 3$ D (Fig. 3), an impact on the streamwise flow component is also expected. The numerical results for the streamwise velocity component are presented for veering (CR, CCR) and non-veering (CR_NV, CCR_NV) inflow by $x$-$y$ cross sections of the streamwise velocity in the top half of the rotor disc at $z = 125$ m (Fig. 4(a)-(d)), at hub height at $z = 100$ m (Fig. 4(e)-(h)), and in the bottom half of the rotor

5    disc at $z = 75$ m (Fig. 4(i)-(l)).

The effect of wind veer on the streamwise velocity component of clockwise rotating wind turbines is investigated by comparing CR to CR_NV, in Fig. 4(a) vs. (c) at $z = 125$ m, (e) vs. (g) at $z = 100$ m, and (i) vs. (k) at $z = 75$ m. Inflow veer causes a more rapid wake recovery at all heights, based on comparison of the velocity deficit contours. Because enhanced $\frac{\partial v_f}{\partial z} \neq 0$ in the case of veering wind, it provides a source of resolved turbulence resulting in higher entrainment in comparison to the

10    no-veer case. Further, inflow wind veer causes wake deflection in relation to a vertical plan through the nacelle at $y = 0$ in both

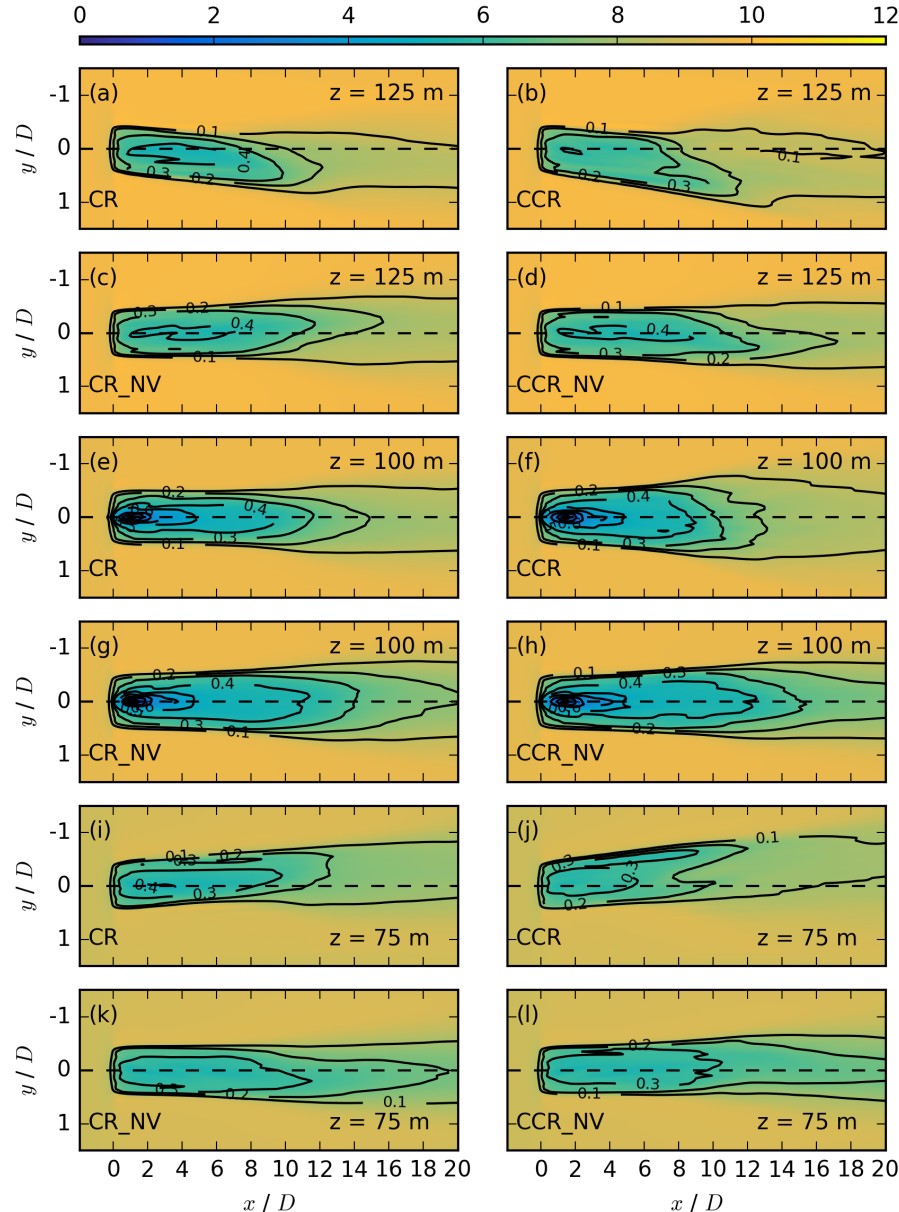

**Figure 4.** Contours of the streamwise velocity $\overline{u_{i,j,k_*}}$ in m s$^{-1}$ at $z = 125$ m in the first two rows, at $z = 100$ m the third and fourth row, and at $z = 75$ m in the last two rows for the simulations CR, CCR, CR_NV and CCR_NV, each averaged over 30 min. The black contours represent the velocity deficit $VD_{i,j,k_*}$ at the same vertical location.

the top half (Fig. 4(a) vs. (c)) and the bottom half (Fig. 4(i) vs. (k)) of the rotor disc. The wake in the veered simulation CR is deflected towards the right ($y > 0$ D) (left ($y < 0$ D)) in the upper (lower) rotor part (Fig. 4(a) (Fig. 4(i))). In the non-veered

**Table 1.** List of all performed simulations in this study for a clockwise (leftmost column) and a counterclockwise (rightmost column) rotor rotation. The parameters $u_g$, $ds$, and $\Omega$ refer to both rotational directions, whereas the only difference e.g. between CR and CCR in the first line is the rotational direction. Further, _NV represents no wind veer, _b a backing wind, _ds refers to varying the directional shear, _u refers to varying the geostrophic wind, and _$\Omega$ refers to varying the rotational frequency in the corresponding simulations.

| SIMULATIONS WITH DIFFERENT ROTATIONAL DIRECTIONS OF THE ROTOR | | | | |
|---|---|---|---|---|
| **CLOCKWISE** | $u_g$ | $ds$ | $\Omega$ | **COUNTERCLOCKWISE** |
| **CR** | $10 \text{ m s}^{-1}$ | $0.08° \text{ m}^{-1}$ | $0.12 \text{ s}^{-1}$ | **CCR** |
| **CR_NV** | $10 \text{ m s}^{-1}$ | $0° \text{ m}^{-1}$ | $0.12 \text{ s}^{-1}$ | **CCR_NV** |
| **CR_b** | $10 \text{ m s}^{-1}$ | $-0.08° \text{ m}^{-1}$ | $0.12 \text{ s}^{-1}$ | **CCR_b** |
| **CR_ds4** | $10 \text{ m s}^{-1}$ | $0.04° \text{ m}^{-1}$ | $0.12 \text{ s}^{-1}$ | **CCR_ds4** |
| **CR_ds12** | $10 \text{ m s}^{-1}$ | $0.12° \text{ m}^{-1}$ | $0.12 \text{ s}^{-1}$ | **CCR_ds12** |
| **CR_ds16** | $10 \text{ m s}^{-1}$ | $0.16° \text{ m}^{-1}$ | $0.12 \text{ s}^{-1}$ | **CCR_ds16** |
| **CR_ds20** | $10 \text{ m s}^{-1}$ | $0.20° \text{ m}^{-1}$ | $0.12 \text{ s}^{-1}$ | **CCR_ds20** |
| **CR_u6** | $6 \text{ m s}^{-1}$ | $0.08° \text{ m}^{-1}$ | $0.12 \text{ s}^{-1}$ | **CCR_u6** |
| **CR_u14** | $14 \text{ m s}^{-1}$ | $0.08° \text{ m}^{-1}$ | $0.12 \text{ s}^{-1}$ | **CCR_u14** |
| **CR_$\Omega$l** | $10 \text{ m s}^{-1}$ | $0.08° \text{ m}^{-1}$ | $0.058 \text{ s}^{-1}$ | **CCR_$\Omega$l** |
| **CR_$\Omega$h** | $10 \text{ m s}^{-1}$ | $0.08° \text{ m}^{-1}$ | $0.175 \text{ s}^{-1}$ | **CCR_$\Omega$h** |
| **CR_$\Omega$vh** | $10 \text{ m s}^{-1}$ | $0.08° \text{ m}^{-1}$ | $0.23 \text{ s}^{-1}$ | **CCR_$\Omega$vh** |

simulation CR_NV, the wake is only slightly deflected towards the left in the top-tip sector (Fig. 4(c)) and towards the right in the bottom-tip sector (Fig. 4(k)). This effect is caused by the rotation of the rotor, which transports higher momentum air counterclockwise, resulting in a wake deflection to the left at $z = 125$ m (Fig. 4(c)). Consequently, the opposite situation prevails at $z = 75$ m (Fig. 4(k)). As the inflow veer contribution to wake deflection is much larger compared to the effect of a clockwise

5    rotating rotor, the wake deflection changes from left in CR_NV (Fig. 4(c)) to the right in CR (Fig. 4(a)) in the upper rotor half and vice versa in the lower rotor half.

As a next step, the rotational direction impact in the non-veered simulations CR_NV and CCR_NV is investigated (Figs. 4(c) vs. (d), (g) vs. (h), and (k) vs. (l)). The impact of the rotational direction on the wake is limited to the wake deflection differences at the upper (Fig. 4(c), (d)) and the lower (Fig. 4(k), (l)) rotor height, which are nearly axis-symmetric to $y = 0$ D and result

10    from the rotational direction of the rotor. These differences in the non-veered simulations agree with results of Vermeer et al. (2003), Shen et al. (2007), Sanderse (2009), Kumar et al. (2013), Hu et al. (2013), Yuan et al. (2014), Mühle et al. (2017), and Englberger et al. (2019).

The rotational direction impact on the wake structure under veering inflow is investigated by a comparison of CCR to CR (Fig. 4(b) vs. (a), (f) vs. (e), and (j) vs. (i)). In CCR, the wake recovers more rapidly (Fig. 4(f) vs. (e)) and the wake deflection

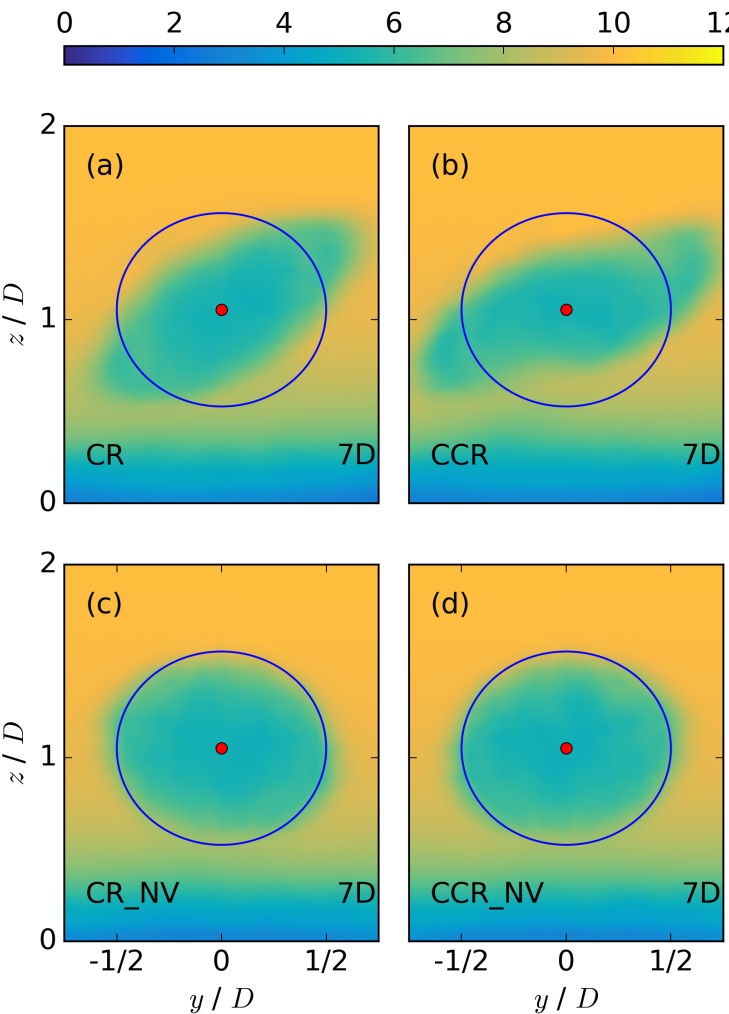

**Figure 5.** Contours of the streamwise velocity $\overline{u_{i,j,k_*}}$ in m s$^{-1}$ at a downward position of $x = 3$ D behind the rotor of for CR in (a), CCR in (b), CR_NV in (c), and CCR_NV in (d). The blue circle represents the circumference of the actuator disc.

angle is larger (Figs. 4(b) vs. (a) and (j) vs. (i)) in comparison to CR. Further, the wake width is larger in the spanwise direction in CCR in comparison to CR (Fig. 4(b) vs. (a), (f) vs. (e), and (j) vs. (i)).

The differences in the spanwise wake width and the wake deflection angle are investigated in more detail with the $y$-$z$ crosssections at $x = 7$ D in Fig. 5 for veering inflow (CR in (a), CCR in (b)) and no wind veer (CR_NV in (c), CCR_NV in (d)) with both rotational directions of the actuator. In the case of no veering inflow, the simulated wake at $x = 7$ D retains the shape of the rotor (Fig. 5(c)). In the case of a veering inflow, however, the wake in the lower rotor half is shifted to the left and in the upper rotor half to the right (Fig. 5(a)). The striking difference between veering and non-veering inflow simulations

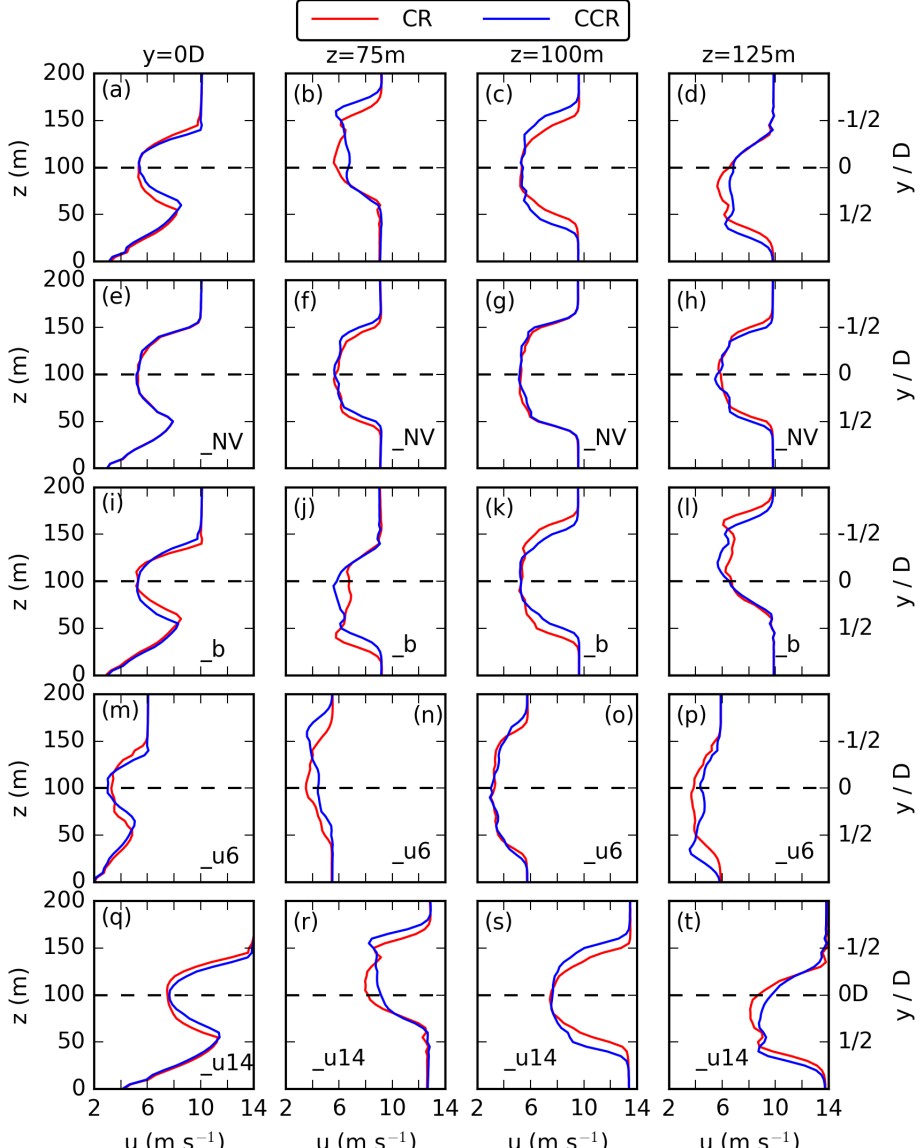

**Figure 6.** Vertical (first column) and horizontal profiles at $z = 75$ m (second column), $z = 100$ m (third column), and $z = 125$ m (fourth column) of the 30 min averaged streamwise velocity at $x = 7$ D downwind of the actuator for CR and CCR in (a) - (d), CR_NV and CCR_NV in (e) - (h), CR_b and CCR_b in (i) - (l), CR_u6 and CCR_u6 in (m) - (p), and CR_u14 and CCR_u14 in (q) - (t).

in combination with a clockwise rotating actuator corresponds to the inflow profile (Eqs. 6, 8), where a veering inflow is characterized by a wind component from right to left for $z < 100$ m and from left to right for $z > 100$ m, whereas the spanwise inflow velocity is zero in the case of no veer in all rotor heights. The skewed wake structure under veering inflow resembles

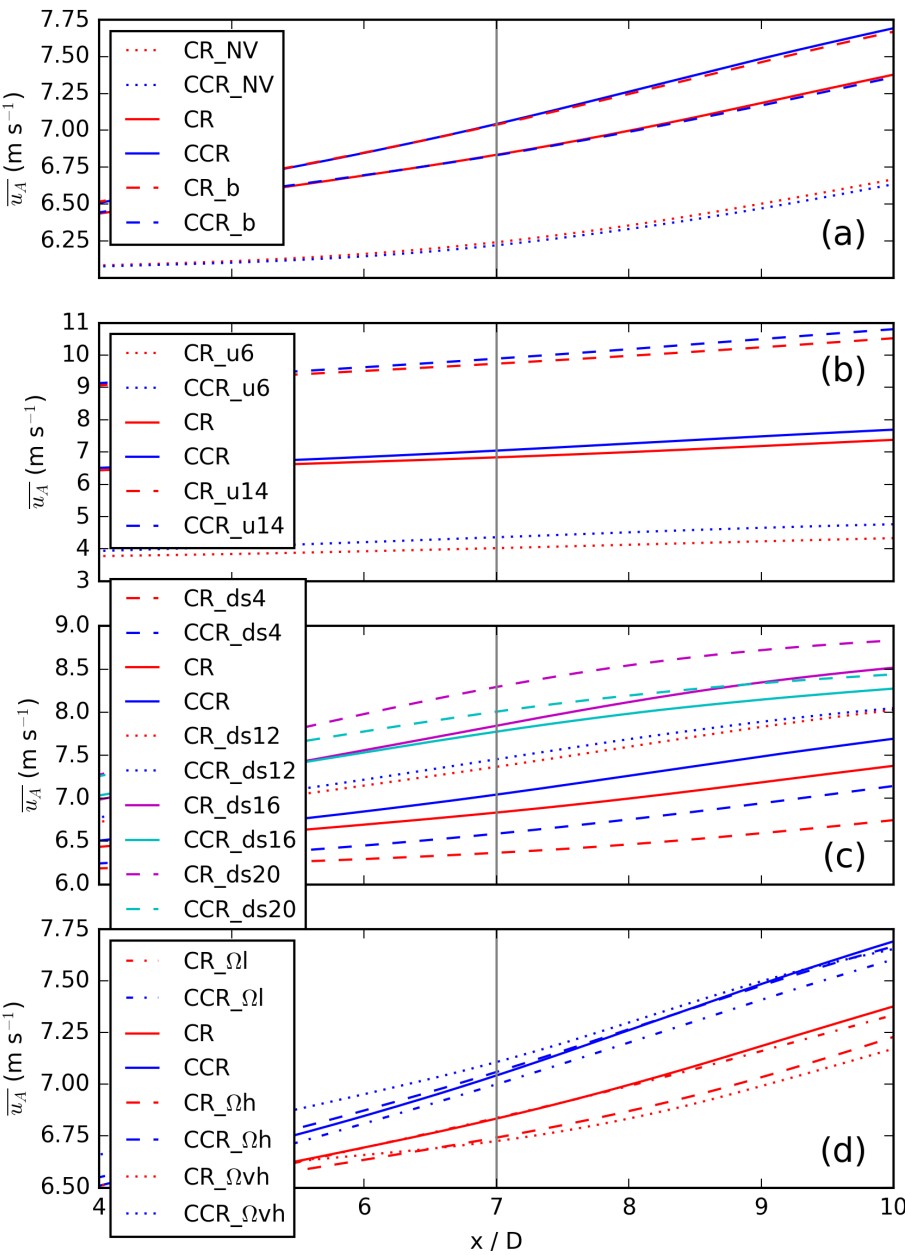

**Figure 7.** The rotor and time averaged streamwise velocity $\overline{u_A}$ presented for a downwind region of $[4\,D;10\,D]$ with special emphasis at $x = 7\,\text{D}$ for the simulations CR_NV, CCR_NV, CR, CCR, CR_b and CCR_b in (a), for different geostrophic wind values in (b), for different directional shears in (c), and for different rotational frequencies in (d).

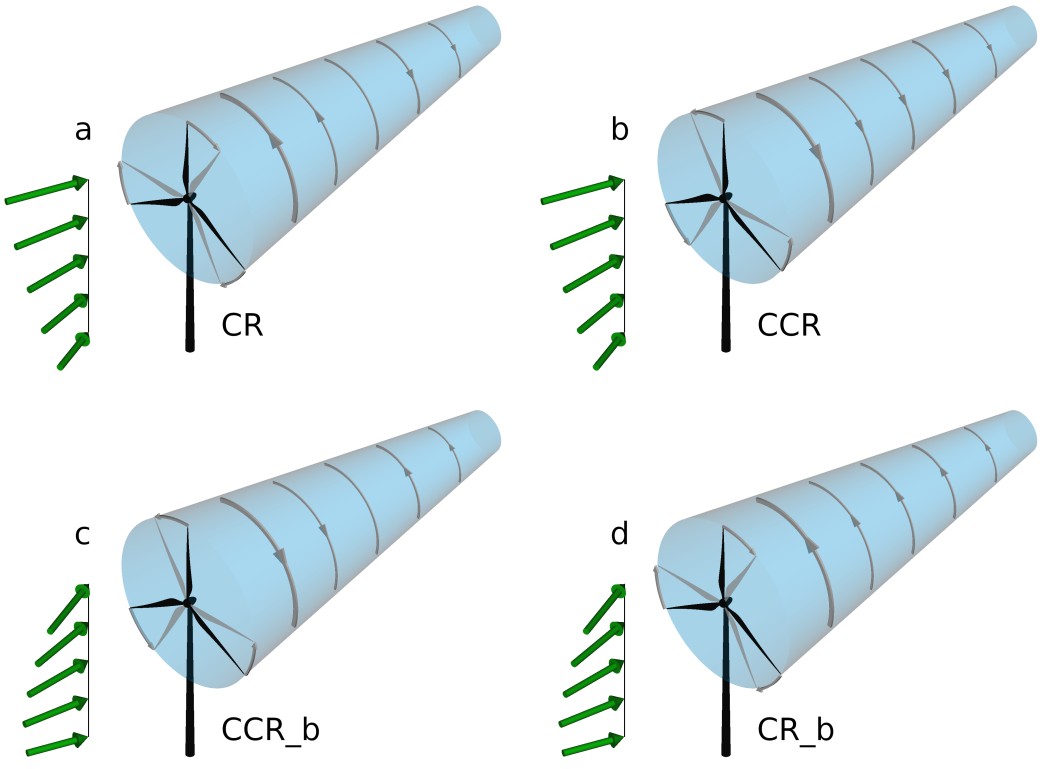

**Figure 8.** Schematic illustration of the rotational direction of the wake for the cases: Clockwise blade rotation CR with veering wind in NH (corresponding to backing wind in SH) in (a), counterclockwise blade rotation CCR with veering wind in NH in (b), counterclockwise blade rotation with backing wind CCR_b in NH (corresponding to veering wind in SH) in (c), and clockwise blade rotation with backing wind CR_b in NH in (d).

those of the simulations of Abkar and Porté-Agel (2016), Vollmer et al. (2017), Bromm et al. (2017), Churchfield and Sirnivas (2018), and Englberger and Dörnbrack (2018a).

Further, we compare the differences between a clockwise and a counterclockwise rotating actuator for non-veering and veering inflow. In the case of no wind veer, the simulated wake structures of CCR_NV (Fig. 5(d)) and CR_NV (Fig. 5(c)) show
5  no striking difference. In the case of veering inflow, however, the skewed wake structure differs in CR and CCR (Fig. 5(a), (b)). Whereas the wake is elliptical in CR, this shape is stretched in the rotor region in CCR. This difference in shape explains the difference in the spanwise wake width at hub height (Fig. 4(f)) and also in the lower (Fig. 4(j)) and the upper (Fig. 4(b)) rotor part. The wake structure outside the rotor region also differs between Fig. 5(a) and (b). Due to the elongation of the elliptical structure in CCR in the rotor region (Fig. 4(b)), and approximately the same vertical wake extension in CCR and CR, the wake
10  deflection angle increases in the case of CCR (Fig. 5(b) vs. (a)), as shown in the lower rotor half in Fig. 4(j) vs. (i) and also in the upper rotor half in Fig. 4(b) vs. (a).

A quantitative description of the streamwise velocity differences is presented in Fig. 6 at the downwind position $x = 7$ D. Figure 6 represents the vertical profiles at $y = 0$ D in (a), and spanwise profiles of $u$ at $z = 75$ m in (b), at $z = 100$ m in (c), and at $z = 125$ m in (d) for both rotational directions CR and CCR. The heights correspond to Fig. 4. Figure 6(e) - (h) represent the non-veering inflow simulations CR_NV and CCR_NV. Whereas the vertical and spanwise profiles of CR and CCR in the case of no inflow veer (_NV) are almost overlapping (Fig. 6(e) - (h)), there is a difference in the case of veering inflow (Fig. 6(a) - (d)). Firstly, the streamwise wake elongation difference of Figs. 4(f) vs. (e) is represented by larger $u$-values in the lower and the upper rotor half in the case of CCR in Fig. 6(b) and (d). The larger wake deflection angle in CCR in comparison to CR (Fig. 4 (b) vs. (a) and (j) vs. (i)) is represented by a larger spanwise distance of the minimum of $u$ from $y = 0$ D in case of CCR in the lower (Fig. 6(b)) and the upper (Fig. 6(d)) rotor half. This spanwise difference of $u_{min}$ is accompanied by larger $u$-values in the case of CCR for $y < $ -1/2 D in the lower rotor part (Fig. 6(b)) and for $y < 1/2$ D in the upper rotor part (Fig. 6(d)). Secondly, the difference in the spanwise wake width is represented in all three heights by a larger $\Delta L_y$ with smaller $u$-values in CCR in the outermost region of the left and the right sectors in comparison to CR (Fig. 6(b) - (d)).

As final step, the difference in the wake is summarized by the 30-min time and rotor area averaged streamwise velocity $\overline{u_A}$. Figure 7(a) represents the difference between clockwise and counterclockwise rotating rotors for a veering inflow and in the case of no wind veer from $x = 4$ D to 10 D. At $x = 7$ D, $\overline{u_A}$ is 0.24 m s$^{-1}$ larger in the counterclockwise rotating rotor simulation CCR in comparison to CR, whereas there is no difference between CCR_NV and CR_NV. According to Fig. 6(a) - (d), these larger $\overline{u_A}$-values in the case of CCR result from larger $u$-values in the upper and lower sector related to the larger wake deflection angle in the case of CCR, which compensates for the larger $u$-values in the outer region of the left and right sectors resulting from a larger spanwise wake width in the case of CCR.

The previous investigations show a striking dependence of the rotational direction of the rotor on the wake under veering inflow, which is qualitatively well explained by the analysis. A schematic illustration of the deceleration or even reversion of the spanwise flow if a clockwise rotating rotor CR interacts with a veering wind is presented in Fig. 8(a). The amplification of the spanwise flow in the case of a counterclockwise rotating rotor CCR interacting with veering inflow is presented in Fig. 8(b).

## 4.2  Veering Wind vs. Backing Wind

According to the analytical results (Fig. 2(e) vs. (h)), the spanwise component $v$ in the wake is expected to be comparable for a clockwise rotating rotor in veering inflow and a counterclockwise rotating rotor in backing inflow, as well as a clockwise rotating rotor in backing inflow and a counterclockwise rotating rotor in veering inflow. The wake characteristics resulting from a backing wind with both rotational directions are investigated in the simulations CR_b and CCR_b and compared to the veering wind cases CR and CCR in Fig. 9. The parameters applied in the corresponding simulations are listed in Table 1.

The behaviour in the upper and the lower rotor part in Fig. 9 can directly be compared after mirroring at $y = 0$ D, an effect resulting from the opposite sign of the directional shear and $\Delta\phi$ in Eqs. 9 and 10. A strong similarity is prevalent in the streamwise velocity component at hub height (Fig. 9(f), (g) and (e), (h)), in the lower rotor half (Fig. 9(i), (l) and (j), (k)) as well as in the upper rotor half (Fig. 9(a), (d) and (b), (c)). The more rapid wake recovery and the larger spanwise wake width for CCR and CR_b in comparison to CR and CCR_b are present in all rotor heights. The larger wake deflection angle in the

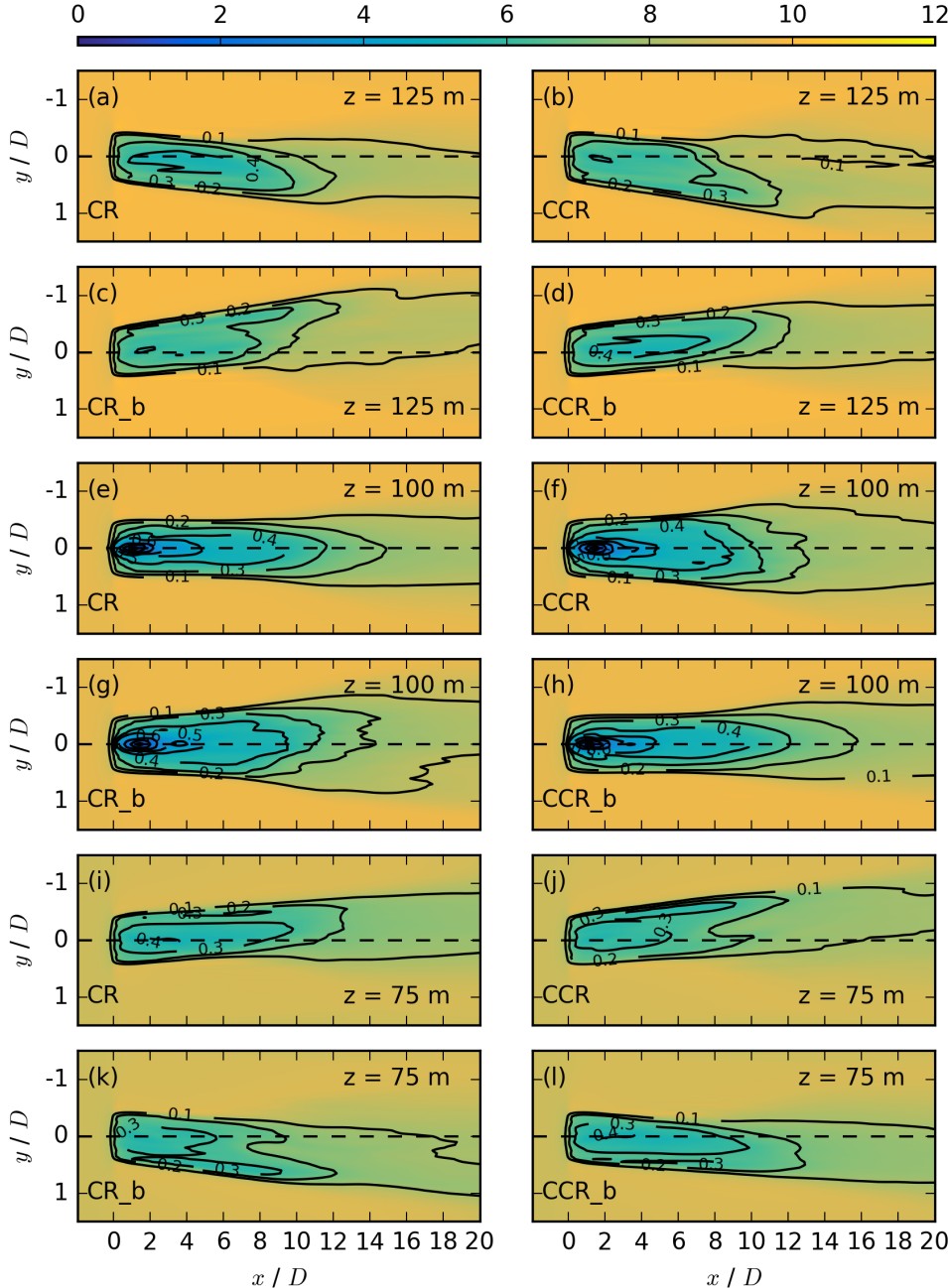

**Figure 9.** Contours of the streamwise velocity $\overline{u_{i,j,k_*}}$ in m s$^{-1}$ at $z = 125$ m in the first two rows, at $z = 100$ m the third and fourth row, and at $z = 75$ m in the last two rows for the simulations CR, CCR, CR_b and CCR_b, each averaged over 30 min. The black contours represent the velocity deficit $VD_{i,j,k_*}$ at the same vertical location.

upper and the lower rotor half in CCR (Fig. 9(b), (j)) is also comparable to CR_b (Fig. 9(c), (k)), whereas the smaller wake deflection angle in CR (Fig. 9(a), (i)) is comparable to CCR_b (Fig. 9(d), (l)).

The qualitative comparison in Fig. 6(i)‑(l) shows the differences from Fig. 9 between CR_b and CCR_b in the case of streamwise wake elongation, spanwise wake width and the wake deflection angle. Further, comparing the backing wind situation (Fig. 6(i)‑(l)) to the veering wind situation (Fig. 6(a)‑(d)), CR_b corresponds to CCR and CCR_b to CR in the vertical (Fig.6(i) vs. (a)) and at hub height (Fig.6(k) vs. (c)). After mirroring at $y = 0\,\mathrm{D}$ CR_b corresponds to CCR and likewise CCR_b to CR in the lower and the upper rotor part (compare Fig. 6(j) to (b) and (l) to (d)).

Expressing the differences between a backing and a veering wind from both rotational directions of the rotor by the quantity $\overline{u_A}$ in Fig. 7(a), the $\overline{u_A}$-values are 0.24 m s$^{-1}$ larger if a backing wind (CR_b) interacts with a clockwise rotating rotor in comparison to a veering wind (CR). Similarly, the $\overline{u_A}$ values are larger if a backing wind interacts with a counterclockwise rotating rotor (CCR_b). Therefore, $\Delta\overline{u_A}$ is the same for CR and CCR_b and likewise for CCR and CR_b.

The northern hemispheric results of CR and CCR are comparable to southern hemispheric CR_b and CCR_b situations, whereas the northern hemispheric results of CR_b and CCR_b correspond to CR and CCR on the SH. The schematic illustration of a backing wind interacting with both rotational directions is presented in Fig. 8(c), (d) with an amplification of the spanwise wind component in the case of a backing wind and a clockwise rotating rotor CR_b (Fig. 8(d)) and a weakening/reversion in the case of a counterclockwise rotating rotor CCR_b (Fig. 8(c)).

### 4.3 Wind Speed

Wind speed may also affect the veering inflow (Eq. 8 via Eq. 6), modifying the spanwise velocity component. There is no significant impact of $u_g = (6, 10, 14)$ m s$^{-1}$ on the wake elongation, the spanwise wake width, and the wake deflection angle between clockwise and counterclockwise rotating actuators. Therefore, the contour plots are not shown. Only a qualitative comparison is presented in Fig. 6(m)-(p) for $u_g = 6$ m s$^{-1}$ and in Fig. 6(q)-(t) for $u_g = 14$ m s$^{-1}$. The occurrence of a wake width as well as the wake deflection angle difference between clockwise and counterclockwise rotating actuators from the reference case $u_g = 10$ m s$^{-1}$ (Fig. 6(b)-(d)) is independent of $u_g$. Only for smaller velocity values ($u_g = 6$ m s$^{-1}$ (Fig. 6(o)), the hub height differences are less pronounced.

The similarity of the vertical and spanwise profiles for all geostrophic wind values in Fig. 6 results in no remarkable difference in $\overline{u_A}$ in Fig. 7(b) between clockwise and counterclockwise rotating simulations. Independent of $u_g$, the values of $\overline{u_A}$ are slightly larger in the case of counterclockwise rotating simulations in compar´ison to clockwise ones, only the difference in $\overline{u_A}$ between clockwise and counterclockwise rotating simulations increases for decreasing $u_g$.

### 4.4 Directional Shear

The directional shear is the second contributing parameter to the veering inflow (Eq. 8), modifying the spanwise velocity component resulting from analysis. The impact of all five directional shear values from Table 1 on the wake is investigated at hub height (Fig. 10), in the upper (Fig. 11), and in the lower (Fig. 12) rotor half. In the clockwise as well as the counterclockwise rotating actuator simulations (Figs. 10‑12) the wake recovers more rapidly if directional shear increases. A larger directional

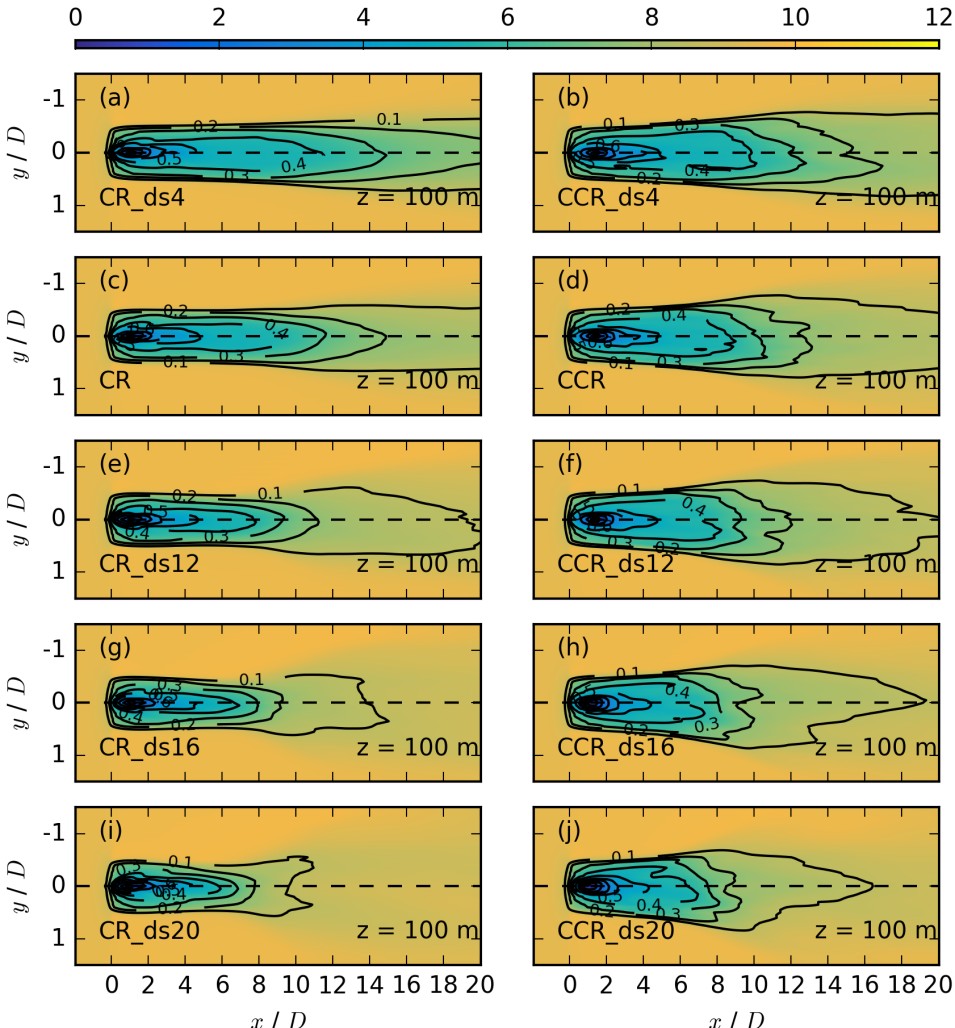

**Figure 10.** Contours of the streamwise velocity $\overline{u_{i,j,k_*}}$ in m s$^{-1}$ for different directional shears at $z = 100$ m for CR_ds4 in (a), CCR_ds4 in (b), CR in (c), CCR in (d), CR_ds12 in (e), CCR_ds12 in (f), CR_ds16 in (g), CCR_ds16 in (h), CR_ds20 in (i), and CCR_ds20 in (j), each averaged over 30 min. The black contours represent the velocity deficit $VD_{i,j,k_*}$ at the same vertical location.

shear represents a larger turbulence source due to an increase of $\frac{\partial v_f}{\partial z}$. Therefore, the simulations with larger directional shear values result in higher entrainment rates and a more rapid wake recovery. Our simulated dependence of the wake recovery on the amount of wind veer for clockwise rotating simulations is comparable to the numerical results in Fig. 11 of Bhaganagar and Debnath (2014).

5     The magnitude of directional shear affects the wake elongation in dependence of the rotor direction, but, not to the same extent in clockwise rotating simulations in comparison to counterclockwise rotating ones. The wake elongation in CR_ds4

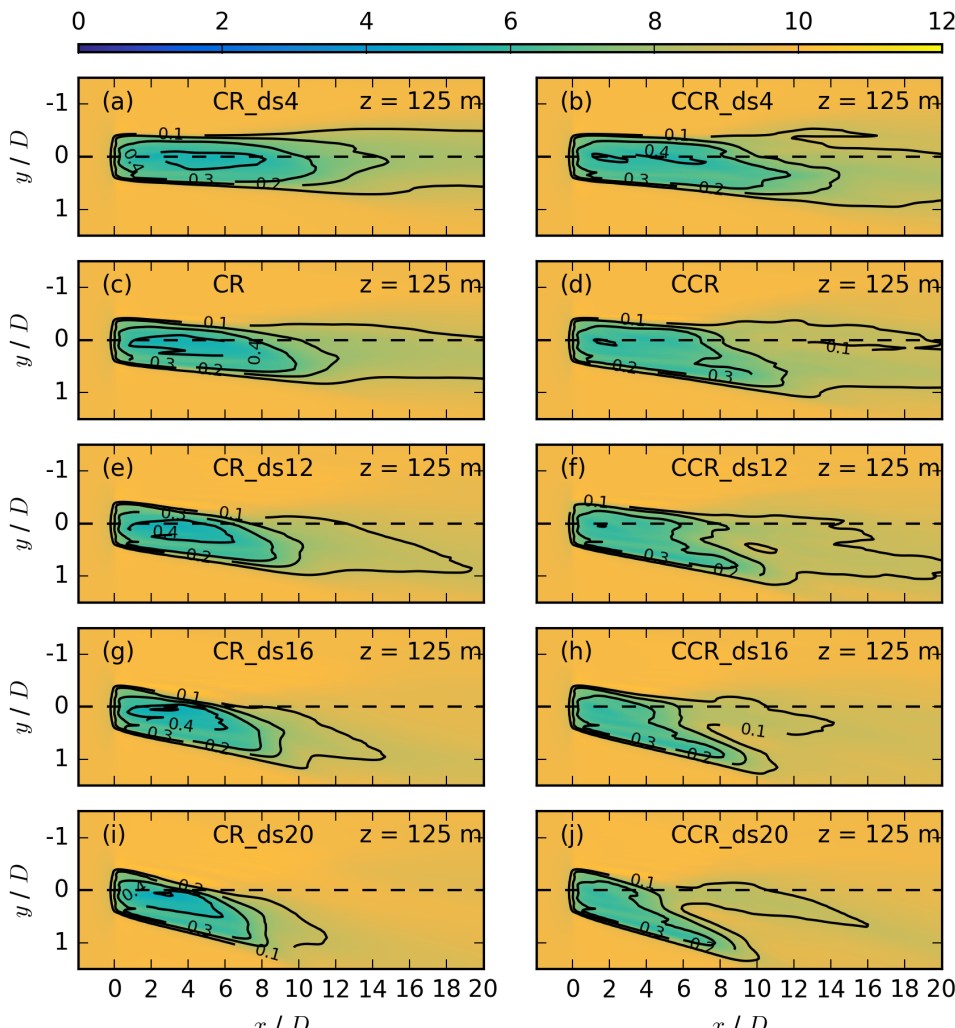

**Figure 11.** Contours of the streamwise velocity $\overline{u_{i,j,k_*}}$ in m s$^{-1}$ for different directional shears at $z = 125$ m for the same simulations as in Fig. 10. The black contours represent the velocity deficit $VD_{i,j,k_*}$ at the same vertical location.

is much longer in comparison to CCR_ds4 (Figs. 10-12(a) vs. (b)). It is still larger in CR in comparison to CCR (Figs. 10-12(c) vs. (d)). A further increase of the directional shear finally results in a similar wake recovery of CR_ds12 and CCR_ds12 (Figs. 10-12(e) vs. (f)) and a slightly more rapid wake recovery of CR_ds16 in comparison to CCR_ds16 (Figs. 10-12(g) vs. (h)). Comparing the very strong directional shear cases CR_ds20 and CCR_ds20 (Figs. 10-12(i) vs. (j)), the wake recovery is significantly faster in CR_ds20. Further, the difference between CR_ds4 and CR is larger in comparison to CCR_ds4 and CCR (Figs. 10-12(a), (c) and (b), (d)). This trend continues for increasing directional shear.

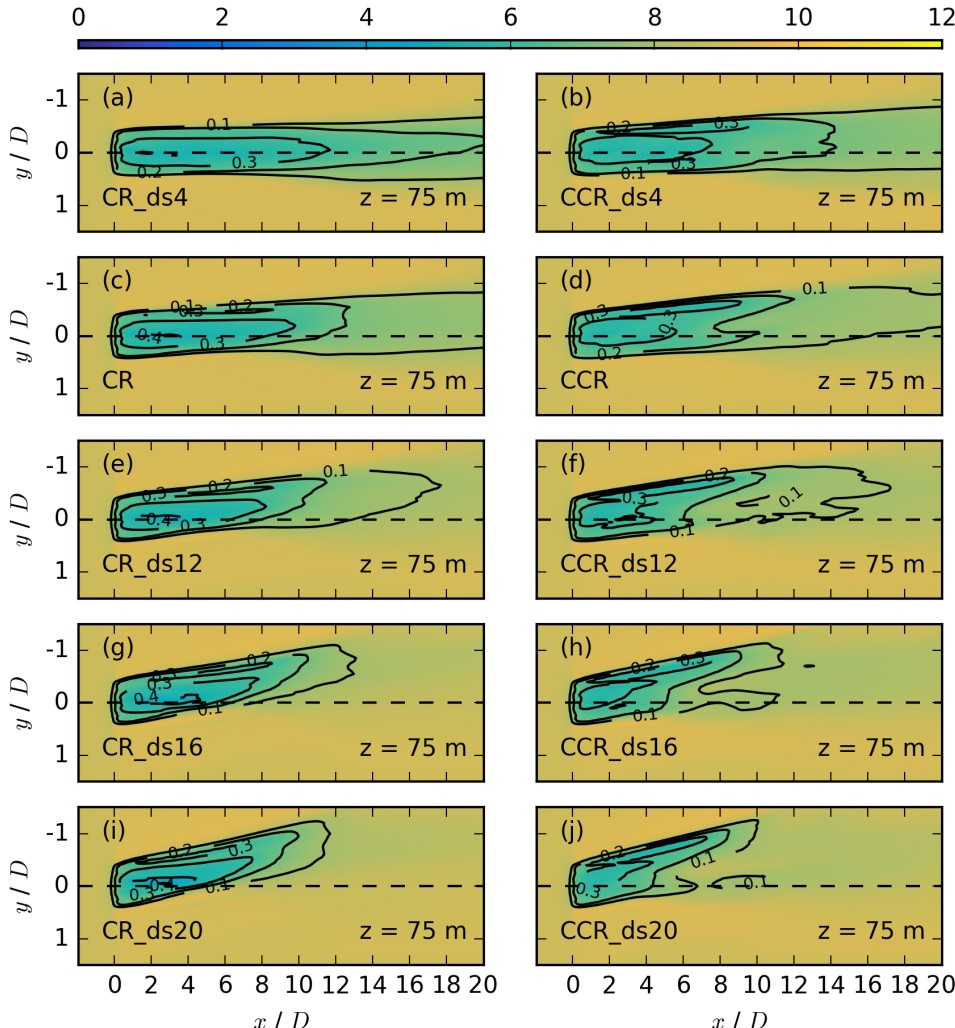

**Figure 12.** Contours of the streamwise velocity $\overline{u_{i,j,k_*}}$ in m s$^{-1}$ for different directional shears at $z = 75$ m for the same simulations as in Fig. 10. The black contours represent the velocity deficit $VD_{i,j,k_*}$ at the same vertical location.

Another difference between clockwise and counterclockwise rotating actuators is the spanwise wake width (Fig. 10(c) vs. (d)). The impact of the directional shear on the spanwise wake width at hub height results in an increase of the difference of the spanwise wake width between a clockwise and a counterclockwise rotating simulation (Fig. 10(a), (c), (e), (g), (i) vs. (b), (d), (f), (h), (j)). In addition, the wake deflection angle increases for increasing values of the directional shear. The
5  difference of larger wake deflection angles in the case of a counterclockwise rotating actuator in the upper and the lower rotor part is also prevalent for all directional shear values (right column of Figs. 11, 12(b), (d), (f), (h), (j)). Further, large values of the directional shear in combination with a counterclockwise rotating actuator leads to a break up of the wake (Fig. 11, 12(h), (j)).

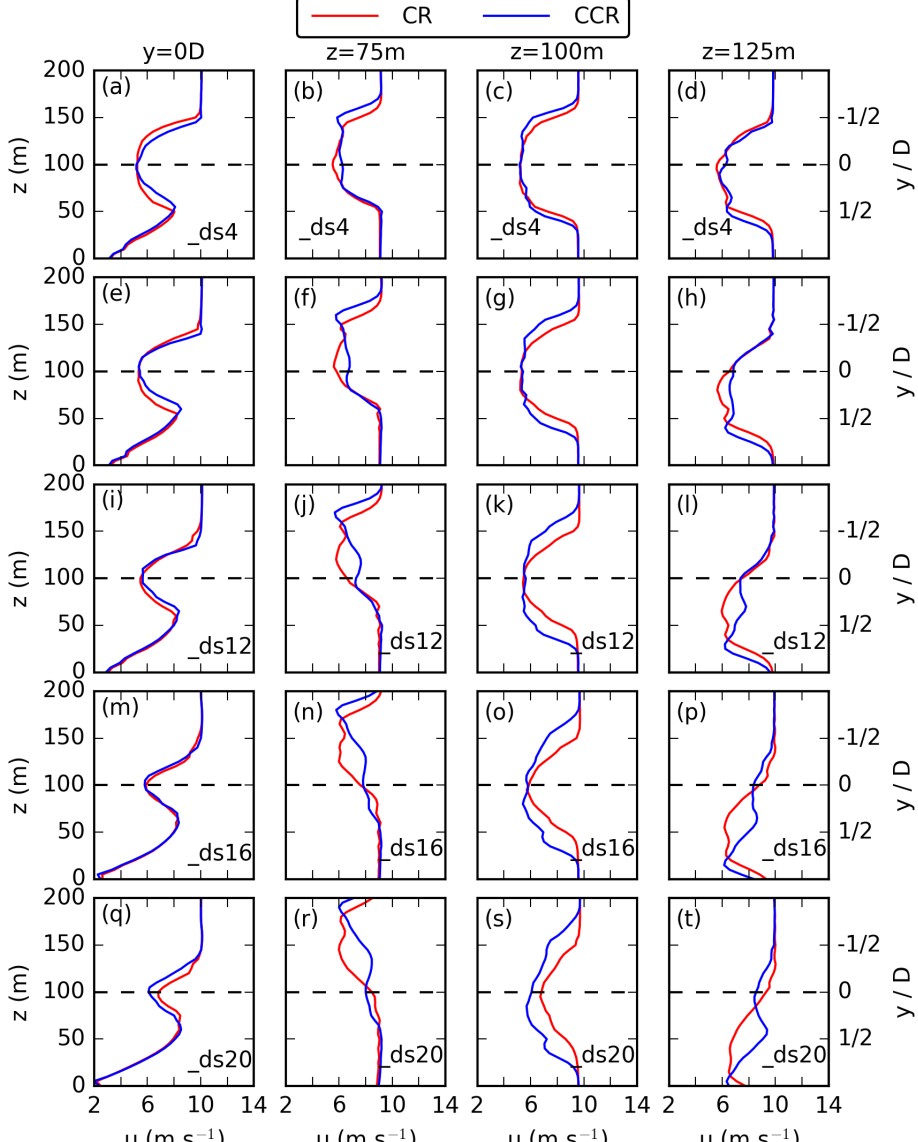

**Figure 13.** Vertical (first column) and horizontal profiles at $z = 75$ m (second column), $z = 100$ m (third column), and $z = 125$ m (fourth column) of the 30 min averaged streamwise velocity at $x = 7$ D downwind of the actuator for a directional shear of $0.04°$ m$^{-1}$ in (a)-(d), $0.08°$ m$^{-1}$ in (e)-(h), $0.12°$ m$^{-1}$ in (i)-(l), $0.16°$ m$^{-1}$ in (m)-(p), and $0.20°$ m$^{-1}$ in (q)-(t).

This erosion could be related to high spanwise velocity values in the case of CCR_ds16 and CCR_ds20 due to amplification of the inflow and the vortex spanwise component, which is not the case in CR_ds16 and CR_ds20 (Fig. 12, 11(g), (i)).

For a quantitative investigation of the directional shear impact on the differences in the wake between clockwise and counterclockwise rotating actuators, vertical and horizontal profiles at $x = 7\,\mathrm{D}$ for all five cases of different directional shear values are presented in Fig. 13. Considering the vertical profile through $y = 0\,\mathrm{D}$ (left column of Fig. 13), the vertical wake extension decreases if the directional shear increases, as the wake deflection is influenced by the incoming wind direction at each

height (Churchfield and Sirnivas, 2018; Tomaszewski et al., 2018; Bodini et al., 2017; Englberger and Lundquist, 2020). This dependency of wake veer on wind veer is also represented at $z = 75$ m (Fig. 13(b), (f), (j), (n), (r)) and at $z = 125$ m (Fig. 13(d), (h), (l), (p), (t)), where the wake deflection angle is additionally influenced by the rotational direction of the actuator. The wake deflection angle is larger if the actuator rotates counterclockwise, independent of the values of directional shear.

The directional shear impact on the spanwise wake width is investigated via the profiles of Fig. 13. Especially at hub height (Fig. 13(c), (g), (k), (o), (s)), the wake width decreases if the directional shear increases. This effect can be related to the increase in skewness in the wake for an increasing directional shear. Comparing clockwise and counterclockwise rotating actuators, the spanwise wake width is larger in the case of a counterclockwise rotating actuator, independent of the directional shear value.

Considering the rotor-averaged values $\overline{u_A}$ in Fig. 7(c), the rotor-averaged wind speeds are larger for a weak wind veer in the counterclockwise rotating actuator simulations (CCR_ds4) in comparison to the clockwise rotating ones (CR_ds4). As the wind veer increases, the difference in $\overline{u_A}$ between clockwise and counterclockwise rotating disc simulations decreases. In the case of a moderate to strong wind veer, $\overline{u_A}$ is only slightly larger for the clockwise rotating rotor CR_ds12. Approaching an even higher directional shear in the strong and very strong wind shear cases, this difference between CR_ds16 and CCR_ds16

and likewise between CR_ds20 and CCR_ds20 increases, whereas now the rotor-averaged wind speeds are larger for clockwise rotating actuators in comparison to counterclockwise ones.

Independent of the directional shear, the streamwise velocity values at $x = 7\,\mathrm{D}$ are larger in the case of a counterclockwise rotating actuator in the top and the bottom sector (Fig. 13 second and fourth column). The difference between counterclockwise and clockwise rotation increases in the radial direction away from the nacelle (not shown). This is related to the larger wake

deflection angle in the counterclockwise rotating case in comparison to the clockwise case. Also independent of the directional shear, the streamwise velocity values are larger in the case of a clockwise rotating actuator in the left and right sectors (Fig. 13 third row). This is an effect of the narrower wake width in the case of a clockwise rotating actuator. If the directional shear is small, the larger $u$-values of counterclockwise rotating actuators in the top and bottom sectors are compensating for the larger $u$-values in the case of a clockwise rotating actuator in the right and left sectors. If the rotational direction is very high, the

opposite is the case.

## 4.5   Rotational Frequency

The rotational frequency contributes to the wind-turbine forces in Eq. 1 and modifies the spanwise velocity component (Eq. 17). The wake impact of the four rotational frequency values from Table 1 is presented at hub height (Fig. 14), at $z = 125$ m (Fig. 15), and at $z = 75$ m (Fig. 16). In comparison to the impact of a change of the atmospheric parameters, which was mainly limited

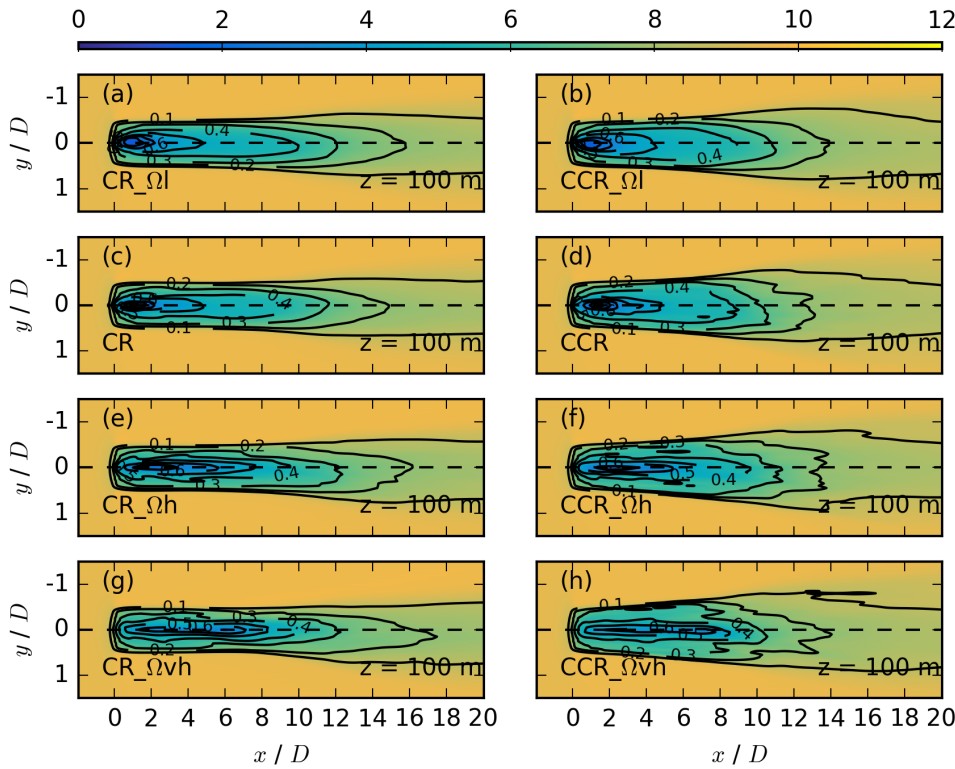

**Figure 14.** Contours of the streamwise velocity $\overline{u_{i,j,k_*}}$ in m s$^{-1}$ for different rotational frequencies at $z = 100$ m for CR_$\Omega$l in (a), CCR_$\Omega$l in (b), CR in (c), CCR in (d), CR_$\Omega$h in (e), CCR_$\Omega$h in (f), CR_$\Omega$vh in (g), and CCR_$\Omega$vh in (h), each averaged over 30 min. The black contours represent the velocity deficit $VD_{i,j,k_*}$ at the same vertical location.

to the far wake, the rotational frequency also significantly impacts the near wake. An increase of the rotational frequency results in a larger minimum value of the velocity deficit (Figs. 15 and 16(g), (h) vs. (a), (b)) and a less rapid wake recovery (Figs. 15 and 16(g), (h) vs. (a), (b)). As the rotational frequency increases, the wake structure differs more between clockwise and counterclockwise rotating actuators. The difference in the spanwise wake width increases for an increasing rotational frequency at all heights. Further, an increase in the rotational frequency results in a slightly larger downwind wake extension in the case of a clockwise rotating actuator at all heights. In the case of a counterclockwise rotating actuator, however, an increase in the rotational frequency results in a similar downwind wake extension of $VD$ (Eq. 14).

A wake splitting pattern exists in the upper (Fig. 15(f), (h)), as well as in the lower rotor part (Fig. 16(f), (h)) for large rotational frequency values. The pattern is similar to the break up of the wake for large directional shear values interacting with a counterclockwise rotating actuator in CCR_ds16 and CCR_ds20 (Fig. 11, 12(h), (j)). The occurrence of the pattern in combination with high rotational frequency values could also be related to a very large spanwise flow component, now resulting from a large contribution of the vortex. An additional simulation (not shown) with $u_g = 10$ m s$^{-1}$, $ds = 0.20°$ m$^{-1}$, and

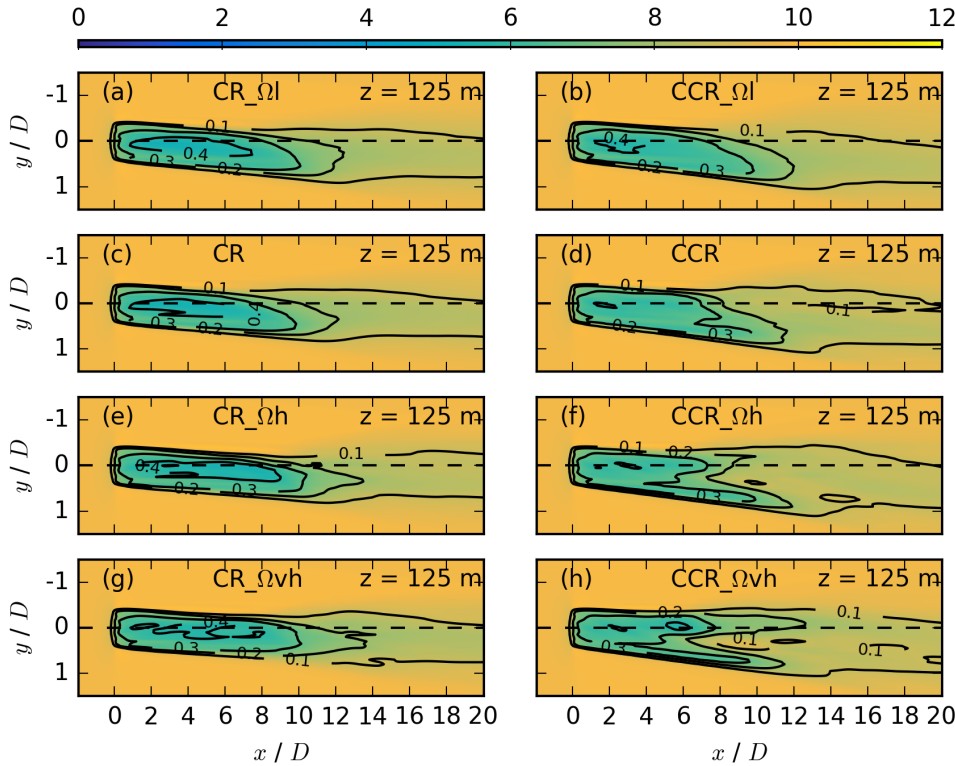

**Figure 15.** Contours of the streamwise velocity $\overline{u_{i,j,k_*}}$ in m s$^{-1}$ for different rotational frequencies at $z = 125$ m for the same simulations as in Fig. 14. The black contours represent the velocity deficit $VD_{i,j,k_*}$ at the same vertical location.

$\Omega = 0.23$ s$^{-1}$ reinforces the slitting pattern of Figs. 12 and 11(h), (j), supporting our assumption as to why the splitting occurs only for counterclockwise rotating actuators. Further, a similar but less distinctive wake splitting pattern for a counterclockwise rotating actuator was observed in the veer affected lower rotor half with $ds = 0.28°$ m$^{-1}$ and $\Omega = 0.12$ s$^{-1}$, see Fig. 10(f) by Englberger et al. (2019).

5  For a quantitative investigation of the rotational frequency impact on the wake differences between clockwise and counterclockwise rotating rotors, the vertical and horizontal profiles at $x = 7$ D are presented for all four cases in Fig. 17. Considering the vertical and spanwise profiles at $z = 100$ m (Fig. 17 first and third column), the rotational direction decrease of $u$ results from larger wind-turbine forces due to an increase of $\Omega$. The difference in the wake defection angle (Fig. 17 second and fourth column) and in the spanwise wake width (Fig. 17 second, third, and fourth column) between clockwise and counterclockwise rotating discs increases for increasing $\Omega$. An increase of $\Omega$ further results in two $u$-minima in the lower (Fig. 17(j), (n))

10 and the upper (Fig. 17(l), (p)) rotor half and a larger decrease of $u$ approaching $r = R$ at hub height (Fig. 17(k), (o)) in the counterclockwise rotating simulations.

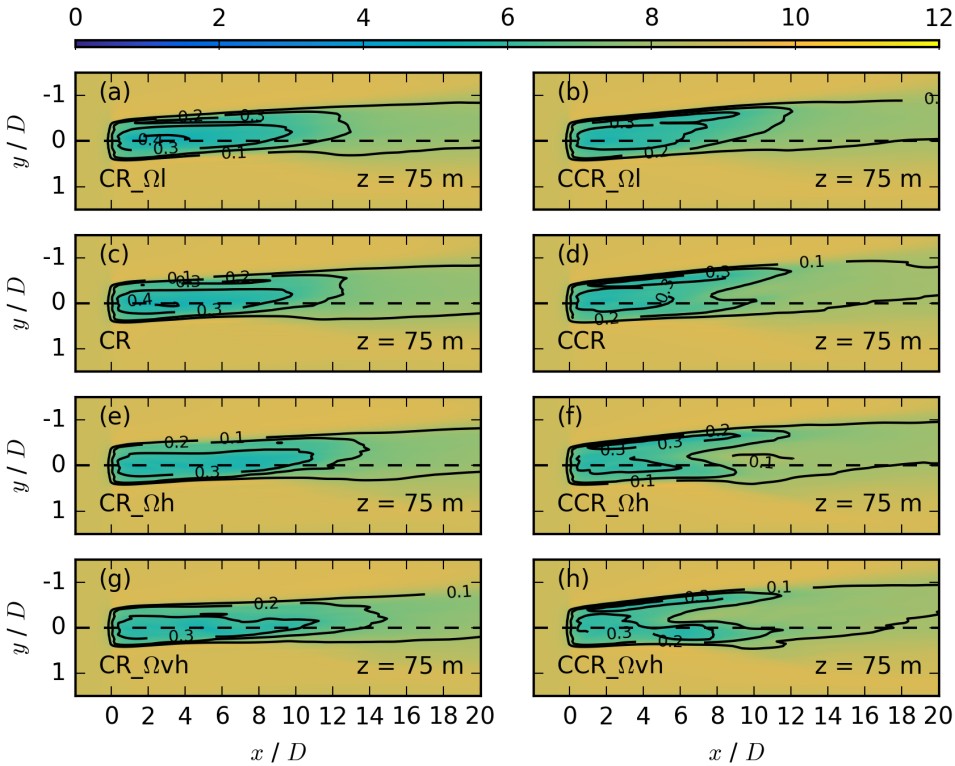

**Figure 16.** Contours of the streamwise velocity $\overline{u_{i,j,k_*}}$ in m s$^{-1}$ for different rotational frequencies at $z = 75$ m for the same simulations as in Fig. 14. The black contours represent the velocity deficit $VD_{i,j,k_*}$ at the same vertical location.

The increase of $u$ in the lower and upper sector compensates for the decrease in the left and right sector for increasing $\Omega$ in the case of counterclockwise rotating discs (Fig. 17), resulting in larger values of $\overline{u_A}$ in Fig. 7(d) for all counterclockwise rotating simulations. The $\overline{u_A}$-difference between clockwise and counterclockwise rotating actuators increases for an increasing rotational frequency, which is related to the splitting of the wake.

## 5 Comparison to Analytic Model

The idealized numerical simulations investigated the impact of the rotational direction of the actuator in combination with veering inflow, no wind veer, and backing inflow on the wake of a single wind turbine. The parameter study investigated the streamwise dependency of the wake on wind speed, directional shear, and rotational frequency. For a comparison of the simulated results with the expected results from analysis in Fig. 2, Fig. 18 is plotted for 90° bottom and top sectors ranging from $0$ m $< $ r $\leq 50$ m (Fig. 1). Figure 2 represents the results at $y = 0$ D and $z = 75$ m as the bottom part of the rotor disk and at $y = 0$ D and $z = 125$ m as the top of the rotor disk. The general structure (slope, sign-changing point), however, is independent

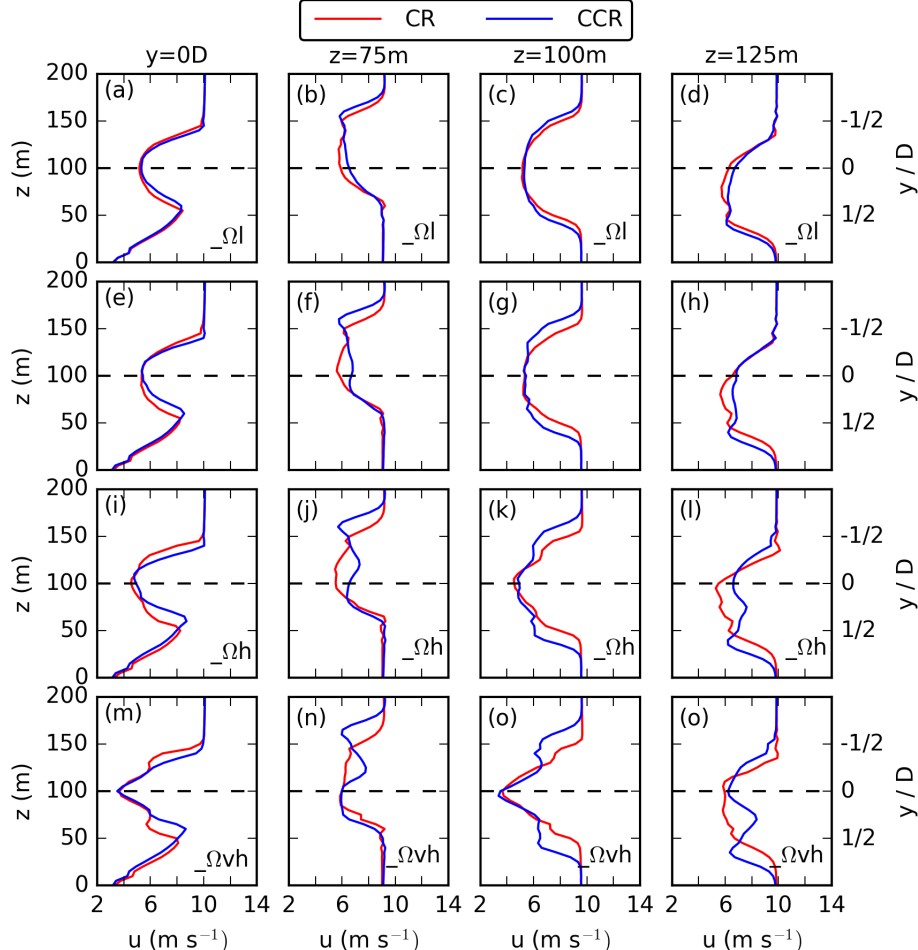

**Figure 17.** Vertical (first column) and horizontal profiles at $z = 75$ m (second column), $z = 100$ m (third column), and $z = 125$ m (fourth column) of the 30 min averaged streamwise velocity at $x = 7$ D downwind of the actuator for a rotational frequency of $\Omega = 0.058°$ s$^{-1}$ in (a) - (d), $\Omega = 0.12°$ s$^{-1}$ in (e) - (g), $\Omega = 0.175°$ s$^{-1}$ in (i) - (l), and $\Omega = 0.23°$ s$^{-1}$ in (m) - (o).

of the vertical location in the analysis. Only the magnitude of the spanwise inflow $v_f$ at $x_{down} > x_\xi$ and of the spanwise vortex component $v_v$ at $x_{down} < x_\xi$ are affected in Fig. 2, as $v_f$ is height dependent and asymmetric to the rotor center and $v_v$ has a radial dependency. Therefore, the panels of Fig. 18 are directly comparable to those of Fig. 2 regarding the difference in $\Delta u_g$, $\Delta ds$, and $\Delta \Omega$ between low, moderate, and high value cases.

5    Comparing the non-veering simulations CR_NV and CCR_NV (Fig. 18(b)) to the analysis prediction (Fig. 2(b)), $v > 0$ in the top sector and $v < 0$ in the bottom sector both with a clockwise rotating rotor. In the case of a counterclockwise rotating rotor, $v$ in the top sector corresponds to $v$ in the bottom sector of a clockwise rotating simulation and vice versa. Only the downwind slope for $x_{down} < x_\xi$ is much smaller. This results from a different radial distribution and a smaller absolute value of the

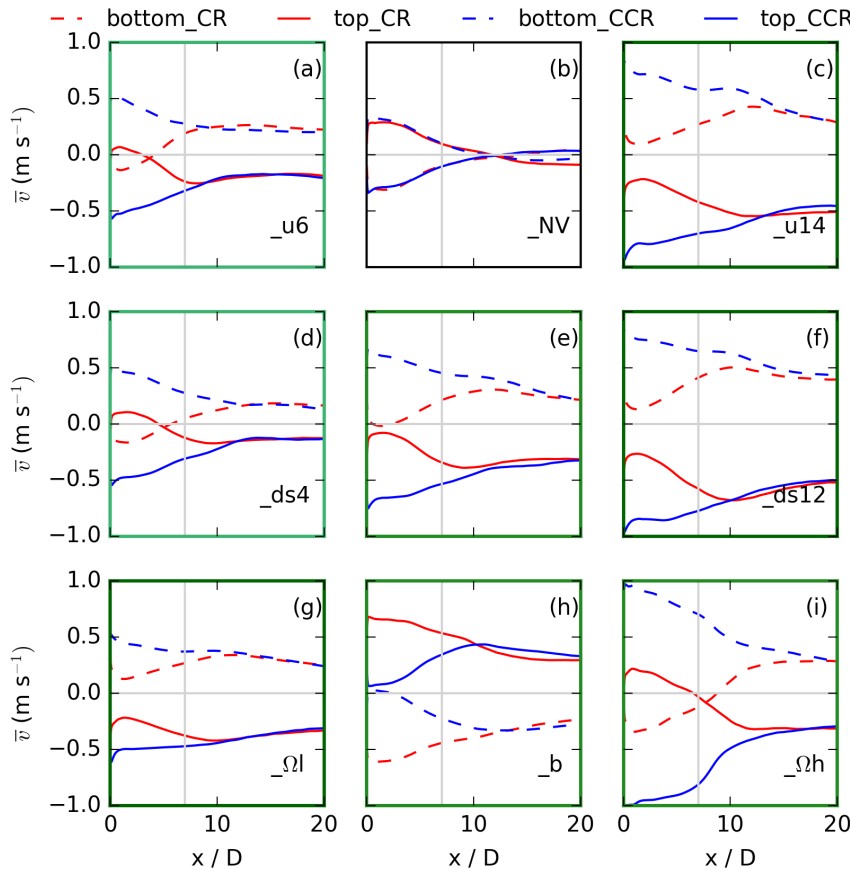

**Figure 18.** Sector averages of $\overline{v}$ representing the top and bottom $90°$-sectors for $0\,\text{m} < r \le 50\,\text{m}$ for clockwise and counterclockwise rotating actuators in the corresponding simulation of no veer in (b), a veering wind in (e) and a backing wind in (h). In the case of a veering wind in (e), moderate parameters of $u_g = 10\,\text{m s}^{-1}$, $ds = 0.08°\,\text{m}^{-1}$, and $\Omega = 0.12°\,\text{s}^{-1}$ are applied. In the left and right column, only one parameter is changes compared to the veering wind situation in (e). Applying low parameters $u_g = 6\,\text{m s}^{-1}$ in (a), $ds = 0.04°\,\text{m}^{-1}$ in (d), and $\Omega = 0.058°\,\text{s}^{-1}$ in (g), and applying high parameters $u_g = 14\,\text{m s}^{-1}$ in (c), $ds = 0.12°\,\text{m}^{-1}$ in (f), and $\Omega = 0.175°\,\text{s}^{-1}$ in (i). The plot is directly comparable to Fig. 2 considering the configurations.

wind-turbine forces applied in the numerical simulations (Eq. 1) in comparison to the Rankine vortex applied in the theoretical analysis (Eqs. 15, 16). Further, the smaller slope in the numerical simulations can be related to the resolved turbulence and the resulting wake recovery.

Comparing the simulation with moderate-veering inflow CR and CCR (Fig. 18(e)) to the analysis predictions (Fig. 2(e)), the acceleration in the case of a counterclockwise rotating rotor and the weakening in the case of a clockwise rotating rotor up to $x_{down} \approx 10\,\text{D}$ are prevalent. The smaller slope-values has the same reason as in the non-veering case. The different slope-

values between the simulations and the analysis predictions simply results in an upwind shift of the sign-changing location in the wake. An increase of $\Omega$ (Fig. 18(i)) approaches the structure predicted by analysis (steeper slope, flow reversal behind the rotor, downwind shift of the sign-changing point) of Fig. 2(b). The values of $\Omega$ applied in the BEM method to calculate the wind-turbine forces (Eq. 1) are the same as the values of $\omega$ applied in the Rankine vortex (Eq. 15). Due to the differences in the calculation of the spanwise flow field between the BEM method and the Rankine vortex, the near wake absolute values are not comparable in Fig. 2 and Fig. 18. Considering the backing inflow simulation (Fig. 18(h)) and exchanging clockwise and counterclockwise and likewise top and bottom, it corresponds to the veering inflow in Fig. 18(e), which is predicted by Fig. 2(h) and (e).

The analytic model predicts the same impact of the geostrophic wind and the directional shear on the spanwise wake structure (Fig. 2(a) vs. (d) and (c) vs. (f)). The general structure at $x_{down} < x_\xi$ is comparable in the numerical simulations with $u_g = 6$ m s$^{-1}$ and $ds = 0.08°$ m$^{-1}$ (Fig. 18(a)) and $u_g = 10$ m s$^{-1}$ and $ds = 0.04°$ m$^{-1}$ (Fig. 18(d)) and likewise with $u_g = 14$ m s$^{-1}$ and $ds = 0.08°$ m$^{-1}$ (Fig. 18(c)) and $u_g = 10$ m s$^{-1}$ and $ds = 0.12°$ m$^{-1}$ (Fig. 18(f)). Minor differences exist e.g., compare the difference between clockwise and counterclockwise rotating simulations at $x = 7$ D in Fig. 18(a) and (d). The larger difference between CR_ds4 and CCR_ds4 in Fig. 18(d) can be related to a decrease of $\frac{\partial v_f}{\partial z}$ and a smaller amount of resolved turbulence generated by the inflow with $ds = 0.04°$ m$^{-1}$ (Fig. 18(d)) in comparison to $ds = 0.08°$ m$^{-1}$ (Fig. 18(a)), whereas the change in $u_g$ has no influence on $\frac{\partial v_f}{\partial z}$. Changing the wind speed and the directional shear has a significant impact on $|v|$ further downwind at $x_{down} > x_\xi$, e.g. $v_{20D}$ in Fig. 18(f) $\approx 2 \cdot v_{20D}$ in Fig. 18(d). This corresponds to the differences between Fig. 2(f) and (d) at $x_{down} > x_\xi$.

Changing the rotational frequency of the vortex has its largest impact on the spanwise wake velocity directly behind the rotor, whereas an increase of the spanwise vortex component $v_v$ results in a larger amplification of $|v|$ in the case of a counterclockwise rotating rotor and a larger weakening of $|v|$ in the case of a clockwise rotating rotor (Fig. 18(g), (e), (i)). This behaviour corresponds to the near wake differences in Fig. 2(g), (e), (i). For large enough values of the rotational frequency, the spanwise wake component reverses sign in the simulation CR_$\Omega$h directly behind the rotor (Fig. 18(i)).

The comparison of the simulation results with the analysis predictions can be summarized as follows:

- The simulated amplification or weakening/reversion of the spanwise inflow wind component in the wake follows the theoretical analysis in the case of no wind veer, a veering inflow, and a backing inflow for clockwise and counterclockwise rotating discs. It can be understood and described by the superposition of the rotational flow induced by the disk and the vertical shear of the incoming wind.

- The agreement of the simulation results with the analysis predictions proves that the impact of the rotational direction on the spanwise wake field is determined by the mean values of the inflow wind field and influenced by the resulting turbulence.

- The inflow parameters (wind speed and directional shear) and the rotation rate of the rotor are two counteracting processes. The individual magnitudes determine the differences in the spanwise wake component between clockwise and

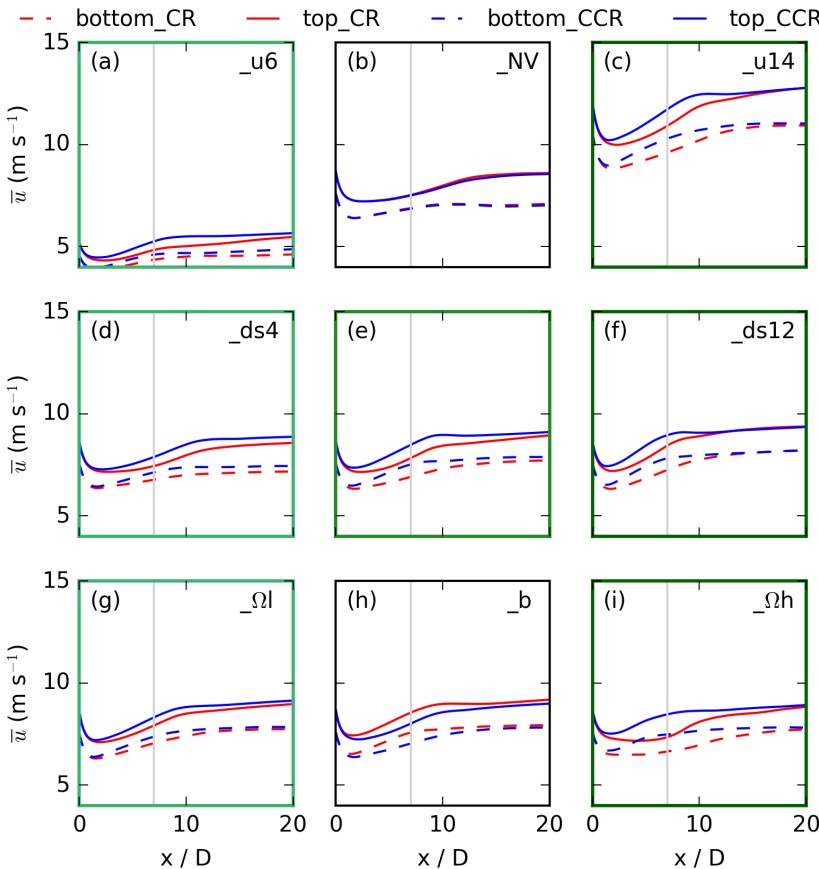

**Figure 19.** Sector averages of $\overline{u}$ representing the top and bottom $90°$-sectors for $0\,\text{m} < \text{r} \le 50\,\text{m}$ for clockwise and counterclockwise rotating actuators for the same simulations as in Fig. 18.

counterclockwise rotating actuators. The difference increases (decreases) for increasing (decreasing) values of $u_g$, $ds$, and $\Omega$.

The rotational direction impact on the spanwise velocity component in the wake also modifies the streamwise flow component. The streamwise velocity components in the wake are shown in Fig. 19 for the same sectors and simulations as in
5  Fig. 18. The larger values in the top sector in comparison to the bottom sector result from the height-dependent streamwise velocity (Eq. 6) with larger values in the upper rotor half. In the case of no wind direction change with height (_NV, Fig. 19(b)), the $\overline{u}$ values are independent of the rotational direction of the actuator. In the case of a veering (backing) inflow (Fig. 19(e) (Fig. 19(b))), the streamwise wake velocity is larger in case of CCR (CR_b) in comparison to CR (CCR_b) in both sectors. The parameters under veering inflow impact the difference $\Delta\overline{u}$ between counterclockwise and clockwise rotating actuators

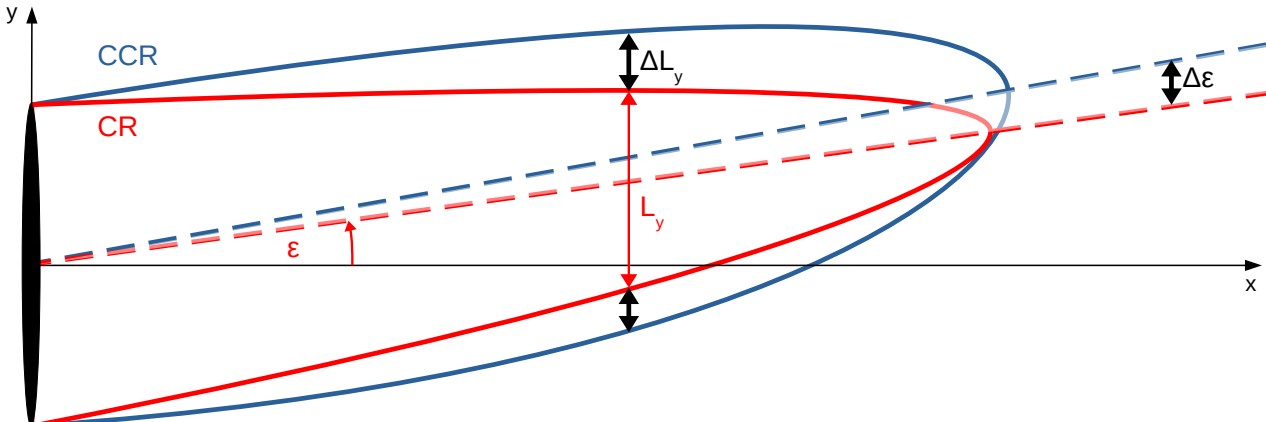

**Figure 20.** Schematic illustration of the difference in the spanwise wake width ($L_y$) and the wake deflection angle ($\epsilon$) between clockwise CR and counterclockwise CCR rotating actuators. The parameter impact on these wake differences is represented by $\Delta L_y$ and $\Delta \epsilon$. The differences in $\Delta L_y$ are valid over the whole rotor, whereas $\epsilon = \Delta \epsilon = 0$ at hub height as $v_f(z_h) = 0$ (Eq. 8).

in the corresponding sector (Fig. 19 left and right column). An increase of $u_g$, $ds$, or $\Omega$ result in larger $\Delta \overline{u}$-values (Fig. 19 right column), whereas smaller parameter values decrease $\Delta \overline{u}$ (Fig. 19(b) left column). The difference $\Delta \overline{u}$ in the left and right sectors is much less distinct in comparison to the top and bottom sectors (not shown).

The rotational direction impact on the wake can be summarized:

– A rotational direction impact on the streamwise velocity components in the wake exists only in the case of a veering (or backing) inflow.

– In the case of veering inflow in the NH the spanwise wake width ($\Delta L_y$) as well as the wake deflection angle ($\Delta \epsilon$) are larger in case of a counterclockwise rotating actuator CCR (Fig. 20). This behaviour is independent of the magnitude of the parameters.

– Increasing the magnitudes of the directional shear $ds$ or the rotation rate $\Omega$ increase the spanwise wake width difference ($\Delta L_y$) and the wake deflection angle difference ($\Delta \epsilon$) between a counterclockwise and a clockwise rotating actuator. The impact of the geostrophic wind $u_g$ is much less pronounced.

– An increase of $u_g$ or likewise a decrease of $\Omega$ result in a more rapid wake recovery. Increasing $ds$, there is no wake difference between a clockwise and a counterclockwise rotating actuator for a specific directional shear $ds_c$. If $ds < ds_c$,
15
the streamwise velocity is larger in the case of a counterclockwise rotating rotor, whereas for $ds > ds_c$, the streamwise velocity is larger in the case of a clockwise rotating rotor. Approaching smaller or larger values of the directional shear, $\Delta \overline{u}$ increases between a clockwise and a counterclockwise rotating actuator.

## 6    Conclusions

We investigate the impact of the rotational direction on the wake of a wind turbine for veering and backing inflow conditions, as well as in the case of no wind veer in both hemispheres, using idealized LES in comparison to a simple analytic model. In addition, the impact of the geostrophic wind and the directional shear as well as the impact of the rotational frequency on the wake differences between clockwise and counterclockwise rotating wind turbines was investigated in the case of veering inflow.

The rotational direction of a wind turbine has only a minor impact on the wake in the case of no wind veer. This result from the numerical experiments is consistent with previous investigations by Vermeer et al. (2003), Shen et al. (2007), Sanderse (2009), Kumar et al. (2013), Hu et al. (2013), Yuan et al. (2014), Mühle et al. (2017), and Englberger et al. (2019). An inflow without wind veer is a typical daytime situation and also occurs in the evening transition of the diurnal boundary layer evolution, were the flow is still influenced by daytime turbulence.

In the case of veering or backing inflow, however, the wake characteristics (streamwise wake elongation, spanwise wake width, wade deflection angle) depend significantly on the rotational direction. Veering and backing inflow are characteristic nighttime situations of the boundary layer flow if no other processes as topographically induced circulations or large scale weather systems prevent the establishment of an SBL regime. Veer within the wind turbine rotor layer has been observed with several field campaigns with towers and lidars (Walter et al., 2009; Sanchez Gomez and Lundquist, 2020; Bodini et al., 2019, 2020), and veer throughout the boundary layer has been observed globally using radiosonde datasets (Lindvall and Svensson, 2019).

Under veering inflow in the NH (backing inflow in the SH), the spanwise wake width and the wake deflection angle are larger for a counterclockwise (clockwise) rotating actuator in comparison to a clockwise rotating one. An increase (decrease) of the directional shear in the atmospheric flow or of the rotational frequency of the rotor increases (decreases) the differences in the spanwise wake width and the wake deflection angle. The wind speed does not impact these wake characteristics significantly. In locations with veering inflow in the NH (backing inflow in the SH) and directional shear values $ds < ds_c$ with $0.12° \text{ m}^{-1} < ds_c < 0.16° \text{ m}^{-1}$, the streamwise velocity is larger in the case of a counterclockwise (clockwise) rotating rotor. These differences apply to the wake ranging from $x = 4\,\text{D}$ to at least $x = 10\,\text{D}$ downwind. For less common higher values of the directional shear $ds > ds_c$, the streamwise velocity is larger in the case of a clockwise (counterclockwise) rotating rotor in the NH (SH).

Different operating conditions (e.g. yaw control) of upwind turbines are already applied to mitigate downwind impacts in wind parks (Fleming et al., 2019). This work suggests that counterclockwise rotating blades in the case of veering inflow and clockwise rotating blades in the case of backing inflow in the NH (and vice versa in the SH) could have benefits as well. The wake deflection angle becomes larger if the spanwise flow component is amplified by the vortex induced by the rotating wind turbine. This process occurs independent of the magnitude of the parameter values applied in the numerical simulations.

As the numerical results of this study arise from an idealized parameter study employing specific assumptions, they have limitations. For example, the turbulent perturbations applied in the numerical simulations are incorporated via a simple tur-

bulence parametrization. The imposed turbulence parameters were retrieved from precursor LES and no real-time wind and potential temperature profiles are applied. Particularly, the impact of the rotational direction on a wind-turbine wake under veering (or backing) inflow results from basic analytical predictions and was compared with the numerical model. However, the impact of rotational directions has never been measured, as no counterclockwise rotating wind turbines currently exist.

Despite the limitations of this numerical study, the simple analysis as well as the idealized parameter study show a consistent and clear impact of the rotational direction of a wind turbine on the wake flow during conditions for which the wind direction turns with height.

To explore a more comprehensive assessment of the wake impact, further investigations would be interesting. The investigation of the non-linearity of the interaction process, numerical simulations applying the turbulence of a SBL precursor simulation

for different strengths of stratification and directional shears, or even considering a low-level jet at the rotor height. Topography could influence the wake dynamic explored here. We have assessed the wake of an individual turbine, but these results could be extended to a large farm in which the presence of upwind turbines could affect turbulence intensity, which probably affects the magnitude. However, an important point will be to prove the theoretically predicted effect resulting from superposition of inflow veer with the vortex component on the wake with measurements.

Finally, the overall assessment of the impact of these results depends on the frequency of occurrence of veering inflow. Only limited sets of long-term observations provide an assessment of the frequency of veering (Walter et al., 2009; Sanchez Gomez and Lundquist, 2020; Bodini et al., 2019, 2020) in the wind turbine rotor layer. The global climatology of veer throughout the atmospheric boundary layer based on radiosonde data (Lindvall and Svensson, 2019) suggest that veer occurs broadly in mid-latitudes and polar regions, but further investigation is required to assess if that boundary-layer veer broadly affects wind

energy generation.

## Appendix A: Turbulence parametrization

The turbulence parametrization of Englberger and Dörnbrack (2018a) is applied in the simulations of this work. The main part is conducted with $\alpha = 0.3$, $\alpha_u = 0.15$, $\alpha_v = 0.24$, and $\alpha_w = 0.13$. This values are nighttime representations following Table 1 of Englberger and Dörnbrack (2018b). Figure A1 presents the reference CR and CCR wind-turbine simulation applied in this

work at $z = 125$ m in (a) and (b), for $z = 100$ m in (e) and (f) and at $z = 125$ m at (i) and (j). Further, simulation results $\alpha = 0.3$ and $\boldsymbol{\alpha}_{i^*,j,k} = 0$ are presented at $z = 125$ m in (c) and (d), for $z = 100$ m in (g) and (h) and at $z = 125$ m at (k) and (l). Panels (a) and (b) are the reference simulations CR and CCR with veering inflow. Panel (c) and (d) correspond to $\alpha = 0.3$, panel (e) and (f) to $\alpha = 0.5$, and panel (g) and (h) to $\alpha = 0.7$, with $\boldsymbol{\alpha}_{i^*,j,k} = 0$ in all three cases. The streamwise velocity at hub height, as well as in the lower and the upper rotor half show similar characteristics of the near wake velocity deficit maximum, the

streamwise wake elongation, the spanwise wake width, and the wake deflection angle. Only the strength of occurrence of these wake characteristics depends on the turbulent intensity, which is larger in the case of $\boldsymbol{\alpha}_{i^*,j,k} = 0$. This reinforces the assumption that wake characteristic differences depend on the mean wind profile, which is the same in all simulations of Fig. A1, and is no effect of the applied turbulence parametrization.

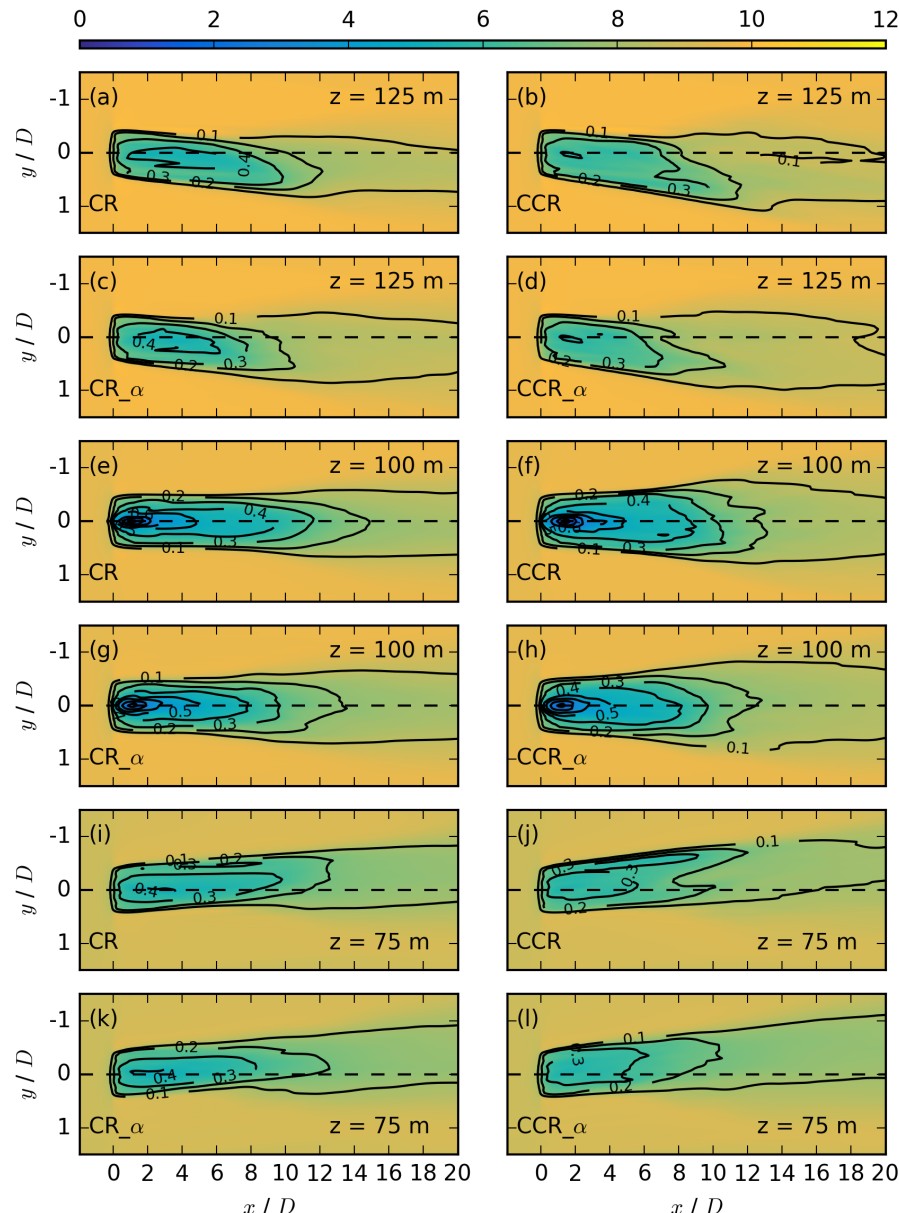

**Figure A1.** Contours of the streamwise velocity $\overline{u_{i,j,k_*}}$ in m s$^{-1}$ at $z = 125$ m in the first two rows, at $z = 100$ m the third and fourth row, and at $z = 75$ m in the last two rows for the simulations CR, CCR, CR_$\alpha$ and CCR_$\alpha$, each averaged over 30 min. The black contours represent the velocity deficit $VD_{i,j,k_*}$ at the same vertical location.

*Author contributions.* All authors designed the idea. A. Englberger performed the simulations and prepared the manuscript with contributions from both co-authors.

*Competing interests.* The authors declare that they have no conflict of interest.

*Acknowledgements.* The authors gratefully acknowledge the Gauss Centre for Supercomputing e.V. (www.gauss-centre.eu) for funding this project by providing computing time on the GCS Supercomputer SuperMUC at Leibniz Supercomputing Centre (LRZ, www.lrz.de).

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
