# Peer review of "Changing the rotational direction of a wind turbine under veering inflow: A parameter study"

_Wind Energy Science, 2019_

## Short Comment (SC1) · 29 Jan 2020

Amazing that this is could have a significant impact on efficiency. The Coriolis effect gets stronger the larger the spatial extent becomes and so in the differential approximation it disappears. I didn't see any indication that the model was evaluated with respect to radius of the turbine blade. Is that possible to add?

---

## Referee Comment (RC1) · Paul van der Laan (Referee) · 19 Feb 2020

**Review of *Should wind turbines rotate in the opposite direction?* by Antonia Englberger, Julie K. Lundquist, and Andreas Dörnbrack**

Reviewer: M. Paul van der Laan, DTU Wind Energy

The authors employ large-eddy simulations (LES) of a single actuator disk subjected to different stable atmospheric boundary layers to investigate the impact of the rotational direction on the potential downstream wind turbine power. The article has an interesting topic and I think it is worth publishing an in depth study about it. However, I have four main concerns with this work. First of all, the inflow is not a solution of the LES model, but simply set as an initial condition without any inflow turbulence. Secondly, the methodology of quantifying the impact of the rotational direction of the rotor on a downstream wind turbine is not sufficient and the reported gains in power are misleading because they only reflect a few specific cases that are rare with respect to all the flow cases that are typically present in an annual energy production calculation of a wind farm. Thirdly, the information provided in the article is not sufficient to redo the simulations and understand the presented results. Finally, I disagree with the main conclusion. I provided an Appendix where I have performed Reynolds-averaged Navier-Stokes simulations of two NREL-5MW wind turbines with 7D spacing subjected to an atmospheric inflow with a strong wind veer. I also see a relatively large impact of the rotation direction on the power output of the downstream wind turbine for a specific wind direction. However, my simulations suggest the opposite of the present paper, where a clockwise rotating wind turbine in the Northern Hemisphere performs better than a counter-clockwise rotating wind turbine (subjected to a strong wind veer). This is because the initial horizontal wake deflection for clockwise rotating wind turbine (without the effect of wind veer but including a wind shear) is clockwise (as seen from above). The counter-clockwise rotating wake brings fresh momentum from above towards the right side of the wind turbine, which results in a stronger deficit on the left side, and this causes the wake to deflect clockwise at seen from above, as shown by Zahle and Sørensen (2008). The addition of wind veer in the Northern Hemisphere deflects the wake even more clockwise, which is also shown in van der Laan and Sørensen (2017). Hence, I disagree with the authors conclusion. I have written a list of main and minor comments below. Since there are so many major concerns, I am afraid that I have to reject the article.

**Main comments**

1. Why do use a scaled down version of the DTU-10MW wind turbine? Wouldn't it be easier to either use the NREL-5MW wind turbine or the original DTU-10MW wind turbine (which is an upscaled version of the NREL-5MW wind turbine)? These reference wind turbines are made to make a comparison between scientific literature in wind energy more fair, and using a reference wind turbine allows other researchers to redo your simulations more easily.

2. Page 4, Line 26: Here you mention that you set different magnitudes of wind veer over the rotor area. How do you set these magnitudes? It seems that you specify them according to an initial profile from Eq. (6), without changing the physical parameters that actually influence the wind veer, i.e. the Coriolis parameter or geostrophic wind (for a constant inversion strength and atmospheric stability). If this is the case then all simulations will converge to the same wind veer if you run them long enough unless you have periodic conditions on all four lateral boundaries. You never mention the word *precursor* or which boundary conditions you use, so it is unclear to me how you make sure that inflow has reached a quasi steady-state before you apply the inflow to a wind turbine wake simulation. If you do not use a precursor simulation, then the inflow will develop downstream and you cannot isolate the wake effects from the imbalance of the inflow profile. If you do use a precursor simulation for each case, then please specify all the input parameters necessary to run each case. In addition, it would make sense to plot all inflow profiles and report the wind speed and turbulence intensity (for example based on the turbulent kinetic energy) at hub height. Furthermore, you seem to use a laminar

inflow (without a roughness length), which does not make sense when modeling a wind turbine wake subjected to an atmospheric inflow.

3. Eq. (9): What is $\eta_{\mathrm{mech}}$? If you intend to calculate the electric (hypothetical) power from the mechanical (hypothetical) power, one would expect to have a 6% loss for a modern wind turbine, not 36%. In addition, it is unclear where the power in the other two dimensions are evaluated ($y$ and $z$), you only mention the downstream distance. Furthermore, I would expect to the power to scale with $U^3$ and I would take the integral of $U^3$ over the rotor area. Please clarify. In addition, if you are not considering a second wind turbine, you are ignoring upstream effects of the downstream wind turbine. It is worth while to mention this simplification.

4. It would be nice to report the tip speed ratio, the thrust coefficient and the power coefficient, for each case. This information is necessary to replicate your simulations.

5. Table 1: This table is confusing. Why do you have both clockwise and counter-clockwise at the same row? I also get confused with the amount of cases and labeling. You also use these labels all the time in the article and it makes it hard to follow the text. Couldn't you simplify the cases and pick those that are really important for your conclusions?

6. Table 1: How is it possible to get a negative and positive wind veer for the same Coriolis parameter? This seems to me that your inflow profile is not in balance with your equations because these kind of effects are typically caused by (unsteady) meso-scale phenomena, which you are not modelling, as far I can understand.

7. I would expect the wake deflects differently for different rotor rotation directions or the wake deflects simply more for a certain rotor direction. Is this correct? It might be worthwhile to discuss and show this (for example with a wake center tracking method). In addition, if you had a downstream wind turbine in unfavourable staggered position, then the additional wake deflection could reduce the power of the downstream wind turbine.

8. Figure 1, why not just plot wake deficit profiles as function of the cross coordinate at different downstream distances? Now you are only looking directly behind the wind turbine, while the wake has deflected laterally (and possible vertically as well), so you are missing a lot of important information.

9. The stream-wise velocity contour plots presented of Figures 3, 5 and 6, do not seem to resemble converged statistics. If you have converged statistics, then I expect smooth plots, see for example the low turbulence intensity case in van der Laan and Andersen (2018), where 1 hour LES results are presented. This could indicate that your LES data set is not large or long enough, or your simulation has not converge to a (quasi) steady-state, but keeps changing instead.

10. Conclusion and abstract: You have to mention that the simulated power increase of 23% only reflects a specific wind direction. In other words, if you would consider multiple wind directions, then the impact of rotor rotation direction on the power (deficit) is much smaller than you report. In addition, if a full wind farm is considered, I expect that the effect of rotation direction is reduced further downstream in the wind farm because of an increase in turbulence level. Finally, if one would look at the effect of the rotor rotation on the wind farm annual energy production, which also consists of many flow cases, where rotor rotation has no influence, you might find that the effect of rotor rotation direction is far less than 1%. Such the study would be necessary in order to answer the question raised in the title.

If your title is *Should wind turbines rotate in the opposite direction?* then I expect to find a thorough answer in the article. The presented simulations cannot answer this question because we need an estimate of the rotor rotation direction on the annual energy production.

**Minor comments**

1. Page 2, Line 20. Remove (SH) at the end of the sentence.

**References**

Apsley, D. D. and Castro, I. P.: A limited-length-scale $k$-$\varepsilon$ model for the neutral and stably-stratified atmospheric boundary layer, Boundary-Layer Meteorology, 83, 75, 1997.

Jonkman, J., Butterfield, S., Musial, W., and Scott, G.: Definition of a 5-MW Reference Wind Turbine for Offshore System Development, Tech. rep., National Renewable Energy Laboratory, 2009.

van der Laan, M. P. and Andersen, S. J.: The turbulence scales of a wind turbine wake: A revisit of extended k-epsilon models, Journal of Physics: Conference Series, 1037, 1, https://doi.org/10.1088/1742-6596/1037/7/072001, 2018.

van der Laan, M. P. and Sørensen, N. N.: Why the Coriolis force turns a wind farm wake clockwise in the Northern Hemisphere, Wind Energy Science, 2, 285–294, https://doi.org/10.5194/wes-2-285-2017, https://www.wind-energ-sci.net/2/285/2017/, 2017.

van der Laan, M. P., Sørensen, N. N., Réthoré, P.-E., Mann, J., Kelly, M. C., and Schepers, J. G.: Nonlinear Eddy Viscosity Models applied to Wind Turbine Wakes, in: Proceedings of International Conference on Aerodynamics of Offshore Wind Energy Systems and Wakes, pp. 514–525, 2013.

van der Laan, M. P., Hansen, K. S., Sørensen, N. N., and Réthoré, P.-E.: Predicting wind farm wake interaction with RANS: an investigation of the Coriolis force, Journal of Physics: Conference Series, 524, 1, https://doi.org/10.1088/1742-6596/625/1/012026, 2015a.

van der Laan, M. P., Sørensen, N. N., Réthoré, P.-E., Mann, J., Kelly, M. C., and Troldborg, N.: The $k$-$\varepsilon$-$f_P$ model applied to double wind turbine wakes using different actuator disk force methods, Wind Energy, 18, 2223, https://doi.org/10.1002/we.1816, 2015b.

van der Laan, M. P., Sørensen, N. N., Réthoré, P.-E., Mann, J., Kelly, M. C., Troldborg, N., Schepers, J. G., and Machefaux, E.: An improved $k$-$\varepsilon$ model applied to a wind turbine wake in atmospheric turbulence, Wind Energy, 18, 889, https://doi.org/10.1002/we.1736, 2015c.

Zahle, F. and Sørensen, N. N.: Overset grid flow simulation on a modern wind turbine, in: AIAA-2008-6727, 2008.

**Appendix A: Reynolds-averaged Navier-Stokes simulations from reviewer**

I have performed Reynolds-averaged Navier-Stokes (RANS) simulations of two NREL-5MW wind turbines (Jonkman et al., 2009) with 7D spacing. The wind turbines are aligned for a wind direction of 270° and 90°. The wind turbines are represented by actuator disks, where the forces are based on airfoil data. The wind turbine force controller is described in van der Laan et al. (2015b). The turbulence is modeled by the limited-length scale $k$-$\varepsilon$ model of Apsley and Castro (1997) coupled with the $k$-$\varepsilon$-$f_P$ model of van der Laan et al. (2013), the latter has been developed for wind turbine wake simulations. The model setup is described in more detail in van der Laan et al. (2015a); van der Laan and Sørensen (2017). The inflow is generated by a precursor simulation and it represents a stable atmospheric boundary layer, where a maximum turbulence length scale of 5 m is set and geostrophic wind speed of 8.4 m/s is chosen. Buoyancy source terms and a temperature equation are not included, the effect of the stability is solely modeled by setting a maximum turbulence length scale, as described in detail in Apsley and Castro (1997). The Coriolis parameter is $10^{-4}$ 1/s and the roughness length is set to a typical offshore value of $10^{-4}$ m. This results in wind speed and turbulence intensity at hub height of 8 m/s and 3%, respectively. The wind veer over the rotor diameter is approximately 12°. The inflow is plotted in Figure A1.

I have run two cases where:

1. Both wind turbines rotate clockwise.

2. Both wind turbines rotate counter-clockwise.

Figure A2 show the power of the downstream wind turbine for the two cases, for a range of wind directions between 240° and 300° with an interval of 2°. Figure A3 shows the relative difference in power of the downstream wind turbine for the two cases. The RANS simulations indicate that the downstream wind turbine has more power when the upstream wind turbine is rotating in the clockwise direction for most wind directions, with a maximum difference of 26%, for a wind direction of 274°. A clockwise rotating wind turbine deflects the wake slightly clockwise (as seen from above) due to the interaction with the shear, as shown by Zahle and Sørensen (2008). The wind veer (in the Northern Hemisphere) also deflects a wind turbine clockwise (as seen from above), as shown in van der Laan and Sørensen (2017). Hence, a clockwise rotating wind turbine and the wind veer in the Northern Hemisphere compliment each other to increase the clockwise wake deflection, which result in an increased power output of the downstream wind turbine for a small range of wind directions. This is the opposite conclusion of the present article.

[Figure]

**Figure A1.** Inflow profiles of wind speed, wind veer, turbulence intensity ($I$) and turbulence length scale ($\ell$). Bottom plot is a zoom of the top plot. Dashed lines represents the boundaries of the rotor plane.

Please note that the RANS simulations presented here use a coupled turbulence model that has not yet been validated thoroughly and might need a re-calibration. However, I expect that the model can predict the qualitative difference between the two cases (clockwise and counter-clockwise rotation) accurately. I have checked that my simulations produce a counter-clockwise wake rotation for a clockwise rotating wind turbine and a clockwise wake rotation for a counter-clockwise rotating wind turbine. I have also checked that the inflow wind veer is rotating clockwise while going up in height. In addition, the RANS setup represent an idealized ABL, which means that the model cannot represent a wind profile with backing.

[Figure]

**Figure A2.** Power of downstream wind direction for clockwise and counter-clockwise rotor rotation using an atmospheric boundary inflow with a strong wind veer for the Northern Hemisphere.

[Figure]

**Figure A3.** Relative difference in power of downstream wind direction for clockwise and counter-clockwise rotor rotation using an atmospheric boundary layer inflow with a strong wind veer for the Northern Hemisphere. Negative difference means more power for the clockwise rotating wind turbine.

To show that a clockwise rotating wind turbine induces a clockwise wake deflection (as seen from above), and to show that a counter-clockwise rotating wind turbine induces a counter-clockwise wake deflection (as seen from above), I have performed additional RANS simulation for neutral surface layer inflow conditions (logarithmic law), using a wind speed and turbulence intensity of 8 m/s and 6%, respectively, at hub height. No Coriolis forces and wind veer are present and only the $k$-$\varepsilon$-$f_P$ model is used as described in van der Laan et al. (2015c). The results are presented in Figures A4 and A5.

[Figure]

**Figure A4.** Power of downstream wind direction for clockwise and counter-clockwise rotor rotation using an atmospheric surface layer inflow without wind veer.

[Figure]

**Figure A5.** Relative difference in power of downstream wind direction for clockwise and counter-clockwise rotor rotation using an atmospheric surface layer inflow without wind veer. Negative difference means more power for the clockwise rotating wind turbine.

---

## Referee Comment (RC2) · Anonymous Referee #2 · 20 Feb 2020

Review of "Should wind turbines rotate in the opposite direction?" (wes-2019-105)

**Summary**

The article investigates the impact of clockwise and counterclockwise rotating wind turbine rotors on the power of a waked downstream turbine in the presence of wind veer and backing. The simulation results show an increased power of the downwind turbine if the streamwise component of the vorticity of the wake is consistent with the one resulting from the wind direction change (i.e. both have the same sign). It is concluded that changing the rotational direction of wind turbines on the northern hemisphere from clockwise to counterclockwise could increase the power of waked downstream turbines. The influence of the stratification, the magnitude and structure of the wind veer, and the wind speed on this result is also investigated to some extent.

**General Comments**

The research question of the article is interesting and well-motivated. I cannot comment on the technical set-up of the LES and the turbine model, as I have no experience with modeling, but some of the chosen simulation parameters seem questionable to me. There are several issues with the results:

(i)     A presentation of the wake structure away from the hub height is missing.

(ii)    The exclusive focus on the mean streamwise velocity ignoring other quantities that affect a downstream turbine (I am not counting the power as a separate quantity here due to way it is computed).

(iii)   No physical explanation is given how the stronger rotation of the wake causes the higher entrainment, which is provided as reason for the main finding.

(iv)    I am not convinced that the increased entrainment is the sole reason for the higher streamwise mean velocity across rotor of the downstream turbine and a modification of the spanwise advection influencing the shape of the wake should be investigated, too.

The conclusions do not account for the limitations of the study and its applicability is overestimated. Therefore, the rather definitive answer to the research question provided here does not hold in my opinion (but there could be an argument to pursue the research question further).

**Language**

I am not a native English speaker, but the manuscript seems to be well written and I did not notice any spelling or grammar mistakes.

**Specific comments**

Page 2, lines 12-14: Sentence should be narrowed to the mixed layer in absence of synoptic or mesoscale forcing.

Page 2, lines 17-19: From the text, it could be misunderstood that the wind veer resulting from the influence of friction is directly connected to temperature advection and lifting. Therefore, I would propose to change the sentence ("This wind veer is associated with…") to something like "Besides the surface friction, temperature advection and dynamic lifting also influence the veering of the wind".

Page 3, lines 9-11: Vasel-Be-Hagh and Archer, 2017 (https://doi.org/10.1016/j.seta.2016.10.004) studied counter-rotating rows of wind turbines in a wind farm and mentions different wake characteristics for the counter rotating turbines.

Page 4, lines 9: The rotor diameter is a third of the height and the width of the simulation domain. Can this affect the wake development? Also the temperature inversion is 50 m above the top tip of the turbine, which corresponds to a very shallow boundary layer. Would a higher inversion layer have an influence on the results?

Page 4, lines 27-29: What is the reasoning for choosing the lower rotor area in contrast to the upper rotor area to modify the type of wind veer? While it is difficult to say anything general about a stable boundary layer, at least for textbook cases the wind veer is stronger in the upper part (opposed to convective boundary layer where wind veer stronger near the surface layer). In addition, the effect is presumably larger in the upper part, because the wind speeds are higher due to wind shear.

Page 5, Eq. 8: Is $\Theta_0$ changed for the very stable case? Otherwise, there is an unstable layer above the hub height, because the pot. temp is 306 K at 200 m 303 K above and that would influence the dynamics for this case.

Page 5, Eq. 9: That should be $\overline{u_A}^3$ instead of $\overline{u_A}$ and since all else is constant, the available power could be used instead.

Page 6, lines 9-10: In Engelberger et al. (2019) – Fig. 8 it is shown that the consistent wake cases have a stronger rotation of the wake compared to the contrasting wake cases at $x/D = 7$. This means that the downwind turbine is receiving a stronger wind veer for the consistent cases compared to contrasting wake cases (beside the higher $\overline{u_A}$ shown here). That stronger wind veer would presumably impact the power of a downwind turbine negatively. Maybe the downwind turbine could be viewed as a yawed turbine for the upper / lower rotor part and Eq. (9) modified to use an adapted power coefficient for each sections of the rotor.

Section 3 in general: Spanwise plots of the streamwise velocity at x/D=7 similar to Fig. 3, 5 and 6 should be shown and discussed. I understand that $\overline{u_A}$ is including values above and below the hub height, but in my opinion this is not sufficient to understand the effect of the direction of the rotor rotation and wind veer on the wake structure. Further insights into the mechanism might be gained by looking at turbulent momentum transport or turbulence production, if available from the LES. A few of the following comments reiterate this comment for the specific subsections.

Page 9, lines 12 – 15: I have three questions on this. First, after a look at the model from Engelberger et al. (2019), I do not yet understand the distinction between entrainment and wake recovery and why entrainment is considered as the explanation for the observations.
Second, do the authors have any notion why the entrainment is larger for the consistent wake case in a physical sense? For example, whether the consistent wake cases have larger gradients of the absolute value of the wind vector due to the rotation, which might facilitate a stronger turbulent momentum transport. Is the turbulent momentum transports from the LES available to investigate this?
Third, I wonder whether a more pronounced ellipsoidal wake cross-section might contribute to a higher $\overline{u_A}$ beside entrainment? Looking at Fig. 6 in Engelberger et al. 2019, an increase of the veer in the wake by the consistent wake cases could make the wake more ellipsoidal. This in turn could cause parts of the wake missing the rotor area of the downstream turbine and increase $\overline{u_A}$, too.

Page 9, lines 27-29: Linking stability directly to the time of day requires the assumption of a radiation driven diurnal cycle of the boundary layer with the absence strong synoptic or meso-scale forcing. The same for page 12, lines 8-11.

Fig. 4: Panel b is quite busy. Would it be possible to make this figure a four panel figure and separate the weak, moderate and strong wind veer cases in one panel and the cases with only the lower rotor area affected by wind veer in a second panel? That would be also more consistent with the subsection structure used in the text.

Page 12, lines 6-9: I believe the phrasing of this sentence is unfortunate, because it could be misunderstood that the power improvement of the downstream turbine itself becomes larger with longer duration (the percentage values from the previous sentence increase over time).

Page 12, lines 18-21: This sentence explains the difference between CR and CCR, but not the difference between CCR_th60 and CCR_th15/CCR. The faster wake recovery for more stable stratification (and presumably a subsequently lower turbulence intensity) for the consistent wake cases is still counter intuitive to me. Do the authors have any explanation what is causing that behavior?

Page 12, lines 25: As for the comment on page 9, lines 12-15, I believe it is possible that an increased ellipsoidal wake shape with increasing wind veer might have a pronounced effect on $\overline{u_A}$ beside entrainment. Vertical cross-sections of the streamwise velocity and plots of the momentum transport could be used to investigate. Maybe some insights into the curious decrease for the strong wind veer case might be gained from them, too.

Page 13, line 1-2: Is this amplified the turbulence production occurring at specific regions of the wake? Could the terms of the TKE budget provide any insights into the cause of the higher entrainment (if they can be computed from the LES)?

Page 15, lines 5-6: I would always expect a larger $\overline{u_A}$ for an increased inflow wind speed if the efficiency of the upwind turbine is not changing (as it is the case here) and I am not seeing where the entrainment is entering the picture from the results. Is that sentence referring to the relative difference between CR / CCR and CR_u14 / CCR_u14?

Section 3.6 and Fig. 7: I like this section bringing everything together and the figure is very informative, but I had a hard time reading the first two paragraphs of this section due to the amount of simulation abbreviations. Since the simulations can be deduced from the Fig. 7, perhaps the text could focus on the physical meanings. E.g. "The blue square shows a power increase by 4% for counterclockwise rotating turbines compared to a clockwise rotating ones for a weakly stable stratification." instead of "The point 'th15' represents a power increase by 4% at 7D for CCR_th15 in comparison to CR_th15".

Page 17, line 6-7: It should be specified that the power of the waked downstream turbines is considered here (it could be misunderstood that the power of the upwind turbine improves, too).

Page 17, lines 22-24: How much of that cumulative capacity is located in wind farms, where wake effects can occur? (in contrast to isolated turbines where it would not matter).

Page 17, lines 28-29: I believe there is a need for further studies on some aspects to this question:

1) This conclusion is based on numerical simulations with simplified a very simplified estimation of the downstream turbine power. A verification with experiments for real wind turbines would be a reasonable call.
2) Unstable and neutral stratification of the boundary layer is not regarded in this study, but can be subject to wind veer as well.
3) Real wind turbines have an induction zone that modify the flow further from the simulation results.
4) Besides the higher streamwise velocity investigated here, the wake structure could see further changes (turbulence intensity, veer, shear), which could impact a downstream turbine.
5) It is possible that two important categories of wind farm locations have a different veering / backing ratios then considered here. Offshore wind parks in proximity to a coast due to the baroclinicity between land and sea. Wind farms located on a ridge due to topography and baroclinicity.

---

## Short Comment (SC2) · 25 Mar 2020

**Comments on Coriolis Effect of Should wind turbines rotate in the opposite direction?**

Antonia Englberger[1], Julie K. Lundquist[2,3], and Andreas Dörnbrack[1]

[1]German Aerospace Center, Institute of Atmospheric Physics, Oberpfaffenhofen, Germany
[2]Department of Atmospheric and Oceanic Sciences, University of Colorado Boulder, Boulder, USA
[3]National Renewable Energy Laboratory, Golden, Colorado, USA

**Correspondence:** Antonia Englberger (antonia.englberger@dlr.de)

Comment: Amazing that this is could have a significant impact on efficiency. The Coriolis effect gets stronger the larger the spatial extent becomes and so in the differential approximation it disappears. I didn't see any indication that the model was evaluated with respect to radius of the turbine blade. Is that possible to add?

Dear Dr. Paul Pukite,

Thank you for your interest in our research.

With respect to your first comment, about the length scale of Coriolis effects, we would like to point out that the ambient or background flow is indeed affected by the Coriolis force. The effect of the Coriolis force is considered by the shape of the hemispheric-dependent Ekman spiral in Eqs. 5 and 6, and therefore determines the meridional velocity component $v$ of the inflow. The wake interacts with this inflow. Considering two simulations in the same hemisphere with both rotational directions (e.g. CR and CCR), the only difference between these simulations is the sign of the prefactor $\beta_v$ in Eq. 1. Therefore, the rotational direction difference on the power output $P$ only depends on the rotational direction.

The second question dealt with how these effects might vary with rotor size. An increase or a decrease of the rotor radius $r$ impacts the rotor area A (and the rotor averaged zonal velocity $\overline{u_A}$), and therefore $P$ in Eq. 9. As we think it is a very interesting question, we performed additional simulations. All simulations in the paper are calculated with $R = 50$ m. Now we added simulation with R = 40 m and 60 m for both rotational directions of the rotor and updated the cross-section plots (Fig. 1) (see Fig. 3 in the paper) and the plot representing the downstream development of $\overline{u_A}$ for both rotational directions (Fig. 2).

The increase of $\Delta \overline{u_A}$ for decreasing $R$ could be related to a larger impact of the veering wind in the rotor region, as a similar dependency arises for different strength of wind veer over the rotor.

[Figure]

**Figure 1.** Coloured contours of the streamwise velocity $\overline{u_{i,j,k_h}}$ in m s$^{-1}$ at hub height $k_h$, averaged over the last 10 min, for R = 40 m in $a$ and $b$, R = 50 m in $c$ and $d$, and R = 60 m in $e$ and $f$. The left column represent a common clockwise rotating rotor $CR$ or $CCW$ and the right column corresponds to the $CCR$ or $CW$ simulations. The black contours represent the velocity deficit $VD_{i,j,k_h}$ at the same vertical location.

[Figure]

**Figure 2.** The rotor averaged streamwise velocity $\overline{u_A}$ and the power $P$ of a hypothetical downwind turbine are presented for a downstream region of $[4D; 10D]$ for different R of 40 m, 50 m and 60 m for $CR$ clockwise rotating rotor and $CCR$ counter-clockwise rotating rotor.

---

## Referee Comment (RC3) · Anonymous Referee #3 · 26 Mar 2020

**Should wind turbines rotate in the opposite direction?**

Antonia Englberger et al.

Referee comments

The authors argue that counter-clockwise rotation wind turbines in northern hemisphere (as opposed to clockwise as is currently done) can lead to a power increase of 11% in the downwind turbine due to constructive interactions between the axial vorticity in the wake and veered Ekman layer, especially when strong stable stratification is present. While I am fascinated by the overall theme of this research, I do not feel that the authors have done a thorough investigation to corroborate their hypothesis. While the paper uses a provocative title and well written, I hesitant in recommending publication at this time since I have the following serious concerns regarding the quality of the numerical simulations performed.

1.  The Ekman layers being simulated are highly stratified with very high gradient Richardson numbers. TKE based eddy-viscosity SGS closures are notoriously terrible at stably stratified layers; see the work by Sullivan et. al, (JAS, 2016) where they show grid sensitivities up to 0.25m for similar states of stratification. You must show that the Ozmidov scale is larger than the grid scale, especially for your strongest stratification case for me to accept the accuracy of the SBL simulated using your SGS closure. This is not done in the current version of the manuscript.
2.  Since much of the argument made in the paper relies on axial vorticity, the authors need to present a strong case showing that the axial vorticity captured by the their grid resolution and actuator-line parameterization is correct. A grid convergence study might help, although I remain skeptical regarding whether actuator lines can correctly represent axial vorticity. There is substantial discussion on this topic in open-literature.
3.  There is new evidence that suggests that ignoring the horizontal component of Earth's rotation (as the authors have done) has a significant quantitative impact on wakes of large turbines representing small Rossby numbers. See the recent work by Howland et al. (2020, JFM) on this topic. Even at approx.. 45deg. Latitudes, I would speculate the direction of wind (Westerly vs Easterly) would affect the power of the downwind turbine by similar order of magnitude as shown by the authors for CW vs CCW rotation.

At this time, I do not feel comfortable recommending publication. However, if the authors systematically address the concerns outlined above and show rigorous grid convergence, I would be happy consider the revised manuscript.

---

## Author Response (AR1)

**Comments on the Review of Should wind turbines rotate in the opposite direction? - Reviewer 1**

Antonia Englberger[1], Julie K. Lundquist[2,3], and Andreas Dörnbrack[1]

[1]German Aerospace Center, Institute of Atmospheric Physics, Oberpfaffenhofen, Germany
[2]Department of Atmospheric and Oceanic Sciences, University of Colorado Boulder, Boulder, USA
[3]National Renewable Energy Laboratory, Golden, Colorado, USA

**Correspondence:** Antonia Englberger (antonia.englberger@dlr.de)

Dear Dr. M. Paul van der Laan,

Thank you for taking the time to carefully review our paper. We read your review in detail and appreciate you sharing your own simulation results. Regarding your comments, we think there are several misunderstandings with the first version of the paper. Therefore, with the help of your comments, we performed some far-reaching changes to the manuscript. Here is a list of the major changes.

– We changed the title.

– We explained in detail the turbulence generation method we applied in the simulations.

– We included a section, introducing a simple analytical model predicting the expected changes in the spanwise velocity field in the wake by a superposition of a veering inflow with a Rankine vortex. (New section 3)

– We added additional simulations with different directional shears (including the $0.12°\ m^{-1}$ value you applied in your simulation).

– We investigated the impact of the rotational frequency on the wake differences.

– We added additional plots, explaining the wake differences and its occurrence for different rotational direction of the actuator.

– We added a section comparing the numerical results predictions of the analytical model. This section explains in detail the source of the difference in the wakes between a clockwise and a counterclockwise rotating rotor in case of a veering inflow.

– We added an Appendix, verifying the application of the turbulence preserving method for this theoretical and idealized parameter study.

In the following we respond in detail to each of your comment/question.

The authors employ large-eddy simulations (LES) of a single actuator disk subjected to different stable atmospheric boundary layers to investigate the impact of the rotational direction on the potential downstream wind turbine power. The article has an interesting topic and I think it is worth publishing an in depth study about it.

We are pleased you think it is an interesting topic and it is worth publishing a detailed study about it. When we received the replies from the reviewers, we realized that we had insufficiently introduced the fairly new topic. The differences were only described and not thoroughly explained. We have corrected this oversight in the revised version.

However, I have four main concerns with this work.

First of all, the inflow is not a solution of the LES model, but simply set as an initial condition without any inflow turbulence.

This was a misunderstanding: In the paper we stated:'A turbulent stably stratified regime in our wind-turbine simulations performed with open horizontal boundary conditions is verified by applying the parametrization of Englberger and Dörnbrack (2018). All parameters required to apply the parametrization are described in detail in Englberger and Dörnbrack (2018).' Instead, we should have explained the turbulent inflow in more detail rather than simply referring to a previous paper where this LES spin-up and inflow turbulence has been developed and successfully validated.

In the modified version we emphasized more clearly that inflow turbulence is applied on the leftmost boundary as a 2D slice at every time step. The inflow turbulence results from the turbulence parametrization of Englberger and Dörnbrack (2018), including turbulent fluctuations retrieved from a neutral boundary layer precursor simulation (Englberger and Dörnbrack, 2017) in combination with adjustable stratification-dependent parameters. In the modified version we explained this impression of turbulence on the inflow in detail (also adding the corresponding equation). Further we verify its applicability for this investigation. In the appendix we further show that the occurrence of a difference between a clockwise and a counterclockwise rotating actuator does not depend on the applied turbulence intensity, only that the degree of the differences is modified by the turbulent intensity. The difference in the flow pattern (amplification of spanwise flow in case of counterclockwise rotating rotor and weakening/reversion in case of clockwise rotating rotor under veering inflow in the NH) only depends on the mean inflow profile and the vortex component of the wind turbine.

In addition, the applied inflow turbulence impacts the wake recovery and the resulting velocity deficit of ≈0.45 at a downstream distance of 7 D. If these simulations had laminar inflow, as we think your comment suggests, the wake would persist much longer.

Secondly, the methodology of quantifying the impact of the rotational direction of the rotor on a downstream wind turbine is not sufficient and the reported gains in power are misleading because they only reflect a few specific cases that are rare with respect to all the flow cases that are typically present in an annual energy production calculation of a wind farm.

The reviewer identifies two issues: the first concerning the methodology and second that these impacts are rare. Regarding the first issue, we changed the selection of considered cases in comparison to the previous version. Further we introduce the

expected results by simple analysis (superposition of spanwise component of veering inflow with Rankine vortex) in a new Section 3 and compared the simulation results to the analytical expectations (new Section 5), to make the manuscript more consistent. In the revised version, we considered cases of changing the atmospheric inflow (geostrophic wind, directional shear) and the rotational frequency of the rotor. All other simulation results from the previous manuscript version are eliminated to be more consistent. With including the expected results in the analysis section (3) and comparing the simulation results to them (section 5), we hope we have proven that our methodology is now sufficient and consistent.

Regarding the second issue, we agree with the reviewer that we specifically only considered an inflow from west to east in the northern hemispheric mid-latitudes 270° at hub height. Because this is an idealized study attempting to understand if this effect is significant in any case, we focused on this idealized inflow scenario and varied the impact factors (geostrophic wind, directional shear, rotational frequency) to investigate the impact of each of them on the difference in the wake structure between clockwise and counterclockwise rotating actuators.

It is quite common for idealized studies to focus on specific wind conditions to understand specific phenomena, and here we explore the difference of the rotational direction impact on the wake under veering inflow conditions. We certainly do not claim to address all relevant flow cases for the annual energy production calculation in a wind farm. The considered cases in our study (regarding the geostrophic wind speed and the directional shear) are chosen from measurement papers like Walter et al. (2009), Sanchez Gomez and Lundquist (2020), **?**, and Bodini et al. (2020). Of course, the measurement results are location specific. We considered three different measurement campaigns, including offshore measurements e.g. 13 months of lidar measurements in Massachusetts in (Bodini et al., 2020), as well as onshore measurements e.g. covering 3 months of lidar observations in north-central Iowa in Sanchez Gomez and Lundquist (2020) and two years of meteorological tower observations in Lubbock (Texas) in Walter et al. (2009). From these measurements we extracted the frequency of occurrence of veering vs. backing and likewise the frequency of occurrence of specific wind speeds and directional shears. In the introduction, we added a paragraph pointing out that our values are chosen in relation to these three measurement campaigns and that the percentage of occurrence of veering or specific wind speed or directional shear values is location dependent.

Maybe the reviewers reaction is related to the simple title of the study, 'Should wind turbines rotate in the opposite direction?'. This question was chosen as title for the paper as it is simple and interesting and for motivation to consider this issue. But we agree with the reviewer that our paper cannot give an answer to this question considering all relevant cases in a wind farm over a year or at any location on earth. Therefore, we change the title of our manuscript to 'Changing the rotational direction of a wind turbine under veering inflow: A parameter study.'

Thirdly, the information provided in the article is not sufficient to redo the simulations and understand the presented results.

Thank you for the comment. We listed all data to the best of our knowledge in the previous manuscript:

- $512 \times 64 \times 64$ grid points
- horizontal and vertical resolution of 5 m
- open horizontal boundaries

- 40 min simulation time
- $D = z_h = 100$ m
- inflow profiles of $u$, $v$, $w$, $\theta$ in Eqs. 4, 5, 7, 8
- wind veer profile in Eq. 6
- BEM with scaled wind turbine (here we refer to Englberger and Dörnbrack (2017, parametrization B))

A detailed listing of all main properties of the simulations are given in Table 1 (wind speed, directional shear both determining the mean inflow wind field and the rotational frequency determining the strength of the vortex)
For more complex simulation inputs (wind-turbine parametrization, turbulence preserving method), we referred to previous papers where all details are listed and the method are validated and explained:

- the wind-turbine parametrization including rotor properties
  'A detailed description of the wind-turbine parametrization and the applied smearing of the forces, as well as all values used in the blade parametrization are given in (Englberger and Dörnbrack, 2017, parametrization B).' (See Table 5 of Englberger and Dörnbrack (2017))
- the turbulence preserving method
  'A turbulent stably stratified regime in our wind-turbine simulations performed with open horizontal boundary conditions is verified by applying the parametrization of Englberger and Dörnbrack (2018). All parameters required to apply the parametrization are described in detail in Englberger and Dörnbrack (2018).' (See Table 1 of Englberger and Dörnbrack (2018))

In the revised version we included the following additional information:

Regarding the wind-turbine parametrization we extend the explanation, but only concisely. For more details about the parameters and applied calculation of $F_{WT}$ from Eq. 1 we refer to Englberger and Dörnbrack (2017, parametrization B). However, we added now the very simple analytical equation showing the same effect of amplification or reduction/reversion of the spanwise wake component. Therefore, the occurrence of the effect does not depend on the turbine type, rotor diameter, radial distribution of the forces. But of course they impact the strength of the effect. As this is a parameter study, not referring to a specific wind turbine, location etc. we did not change our wind-turbine. However, we include the turbine impact by changing the rotational frequency of the rotor to show the sensitivity to the strength of the vortex in the analytical section 3 and also the numerical simulation section 4.

Regarding the turbulence preserving method, we added a much more detailed description (see comment above).

Finally, I disagree with the main conclusion. I provided an Appendix where I have performed Reynolds-averaged Navier-Stokes simulations of two NREL-5MW wind turbines with 7D spacing subjected to an atmospheric inflow with a strong wind veer. I also see a relatively large impact of the rotation direction on the power output of the downstream wind turbine for a

specific wind direction. However, my simulations suggest the opposite of the present paper, where a clockwise rotating wind turbine in the Northern Hemisphere performs better than a counter-clockwise rotating wind turbine (subjected to a strong wind veer).

We really appreciate your effort of performing the simulations and your generosity in sharing your results. However, we do not agree with your statement that your results disagree with ours. We performed simulations with a veering over the rotor of $0.04^\circ\ \mathrm{m}^{-1}$, $0.08^\circ\ \mathrm{m}^{-1}$ and $0.16^\circ\ \mathrm{m}^{-1}$. According to our simulations, a counterclockwise rotating rotor results in a higher downstream velocity at 7 D in case of $0.04^\circ\ \mathrm{m}^{-1}$ and $0.08^\circ\ \mathrm{m}^{-1}$. In case of $0.16^\circ\ \mathrm{m}^{-1}$ a clockwise rotating wake has a higher downstream velocity at 7 D in comparison to a counterclockwise rotating one. Your simulation investigated it for a directional shear of $0.12^\circ\ \mathrm{m}^{-1}$ over the rotor with the same result as our $0.16^\circ\ \mathrm{m}^{-1}$ simulation. Therefore, we think your simulation results did not disagree with our results for the strong wind veer case.

To focus on the impact of the directional shear, we added a simulation with a directional shear of $0.12^\circ\ \mathrm{m}^{-1}$, corresponding to the directional shear you applied in your simulation, and likewise a simulation with a very high directional shear of $0.20^\circ\ \mathrm{m}^{-1}$ to point out the impact of the directional shear on the results. Please see Fig. 7, 10, 11, 12, and 13 in the revised version representing the results. According to our results, for low values of the directional shear ($0.04^\circ\ \mathrm{m}^{-1}$) the rotor and time averaged downwind velocity $\overline{u_A}$ at 7 D is larger for a counterclockwise rotational direction in comparison to a clockwise one. For very high values of the directional shear ($0.20^\circ\ \mathrm{m}^{-1}$) the opposite is the case. In between, there is a directional shear values with no difference in $\overline{u_A}$ between clockwise and counterclockwise rotating actuators. According to our results (and thanks to the added simulations in the new manuscript), this is the case for a critical directional shear value $ds_c$ with $0.12^\circ\ \mathrm{m}^{-1} < ds_c < 0.16^\circ\ \mathrm{m}^{-1}$. The specific value of $ds_c$ of course depends on the turbulent intensity, the rotor diameter, the radial distribution of the forces, the wind-turbine type, the resolution, etc. But regarding the result of $ds_c$ very close to your result with a directional shear of $0.12^\circ\ \mathrm{m}^{-1}$, the deviation is not unexpected for us. Especially regarding the main difference, you considered two wind turbines and we consider the available power in the wind. But also further differences, smaller geostrophic wind, different size of the WT, different radial distribution of the forces, different turbulence applied as inflow condition. Therefore, your results did not show a different result of the complete rotational direction under veering inflow topic. On the contrary, we think it supports our results. It also supports our assumption that the difference is related to the mean inflow fields as predicted by the analysis, as your result is very similar to ours despite all the differences in the atmospheric conditions (different geostrophic wind, turbulence method) and the wind turbine (rotor size, radial distribution of the forces, different rotational frequency).

This is because the initial horizontal wake deflection for clockwise rotating wind turbine (without the effect of wind veer but including a wind shear) is clockwise (as seen from above). The counter-clockwise rotating wake brings fresh momentum from above towards the right side of the wind turbine, which results in a stronger deficit on the left side, and this causes the wake to deflect clockwise at seen from above, as shown by Zahle and Sørensen (2008). The addition of wind veer in the Northern Hemisphere deflects the wake even more clockwise, which is also shown in van der Laan and Sørensen (2017).

We agree with your comments here, but they are valid in case of no veering wind, as you stated. We also apply this explanation in the work Englberger et al. (2019), explaining the difference between clockwise and counterclockwise rotating actuators in the evening boundary layer in case of no wind veer. Regarding the wake deflection in dependence of the rotational direction of

the rotor in combination with veering inflow, our results show that the wake deflection is larger in case of a counterclockwise rotating disc interacting with veering inflow, independent of the atmospheric parameters directional shear and wind speed. We explain this with the amplification of the spanwise flow component in the wake in case of a counterclockwise rotating rotor, resulting in a larger wake deflection in comparison to the weakening or even reversion which occurs in case of a clockwise rotating actuator.

Hence, I disagree with the authors conclusion. I have written a list of main and minor comments below. Since there are so many major concerns, I am afraid that I have to reject the article.

Regarding all the misunderstandings (insufficient description of turbulence generation method instead only referring to the corresponding paper, misleading title in a way we did not anticipate, misinterpretation of your simulation results with our results due to application of a directional shear value with is close to the critical value we detect in our results) we understand your recommendation to reject the previous version of this article. Your comments were really helpful to us to eliminate the misunderstandings via including a detailed description of the turbulence generation method applied, changing the title of the manuscript, adding additional simulations helping to narrow down the critical values of the directional shear, including the simple analytical equation which explained the differences seen in the simulation. Considering the extensive revisions in the presentation of the results we performed in this revised version, we hope we have addressed your concerns.

In the following we refer to your main comments:

1. Why do use a scaled down version of the DTU-10MW wind turbine? Wouldn't it be easier to either use the NREL-5MW wind turbine or the original DTU-10MW wind turbine (which is an upscaled version of the NREL-5MW wind turbine)? These reference wind turbines are made to make a comparison between scientific literature in wind energy more fair, and using a reference wind turbine allows other researchers to redo your simulations more easily.

We understand your point here. Our attempt was to apply the flow field modifications of a generalized wind turbine with a rotor diameter as well as a hub height of 100 m. As it is a parameter study, it is not related to a specific turbine, location, etc. In the revised version we also add the analytical model, which explains the difference and also that they are not turbine dependent (also occurring for a rather simple Rankine vortex).

Page 4, Line 26: Here you mention that you set different magnitudes of wind veer over the rotor area. How do you set these magnitudes? It seems that you specify them according to an initial profile from Eq. (6), without changing the physical parameters that actually influence the wind veer, i.e. the Coriolis parameter or geostrophic wind (for a constant inversion strength and atmospheric stability). If this is the case then all simulations will converge to the same wind veer if you run them long enough unless you have periodic conditions on all four lateral boundaries.

Here is a misunderstanding. We do not prescribe specific inflow profiles in a precursor simulation extending it until it reaches an equilibrium state, therefore, our results will not converge to the same wind veer. In EULAG, we apply the background/environmental wind profiles $u_e(z)$ and $v_e(z)$, without Coriolis force in the simulation, and superposed the turbulence on the inflow. This turbulent inflow wind field interacts with the actuator. The chosen background wind profiles determine if there is a veering or a backing wind or no wind veer at all or in case of a veering inflow the also determine the wind speed and directional shear in the simulations.

You never mention the word precursor or which boundary conditions you use, so it is unclear to me how you make sure that inflow has reached a quasi steady-state before you apply the inflow to a wind turbine wake simulation. If you do not use a precursor simulation, then the inflow will develop downstream and you cannot isolate the wake effects from the imbalance of the inflow profile. If you do use a precursor simulation for each case, then please specify all the input parameters necessary to run each case.

We should have explained the turbulence generation method in more detail. We changed this part of the paper (see general statement). We also added the stratification-dependent parameters applied in the simulations. In the revised version it says ... 'of a neutral boundary layer precursor simulation ...'. To make sure that our simulations reach steady state, we extended the simulation to 40 min, now averaging over 30 min. For the reference simulation we tested a simulation time of 1.5 h with the same result as averaging over the last 30 min.

In addition, it would make sense to plot all inflow profiles and report the wind speed and turbulence intensity (for example based on the turbulent kinetic energy) at hub height.

We thought about the comment of plotting the profiles, however, we did not include the plots basically due to three reasons: Firstly, they are idealized profiles following Eqs. 6 (streamwise component) and 8 (spanwise component). Secondly, the profiles referring to the turbulence generation method are already discussed and shown in Englberger and Dörnbrack (2018). Thirdly, the modified version of the manuscript already includes 20 figures considering the discussion of the results (as requested by the reviewers), therefore, this plot was eliminated in the end.

Furthermore, you seem to use a laminar inflow (without a roughness length), which does not make sense when modeling a wind turbine wake subjected to an atmospheric inflow.

If we understand this comment correctly, you refer to a roughness length $z_0$ in the inflow profile? In EULAG, we do not apply a MOST surface layer parametrization.

3. Eq. (9): What is $\eta_{mech}$ ? If you intend to calculate the electric (hypothetical) power from the mechanical (hypothetical) power, one would expect to have a 6% loss for a modern wind turbine, not 36%. In addition, it is unclear where the power in the other two dimensions are evaluated (y and z), you only mention the downstream distance. Furthermore, I would expect to the power to scale with U 3 and I would take the integral of U 3 over the rotor area. Please clarify. In addition, if you are not considering a second wind turbine, you are ignoring upstream effects of the downstream wind turbine. It is worth while to mention this simplification.

Our very simple power calculation is basically proportional to $u^3$ (sorry we missed the 3 in Eq. 9) should represent the power

available in the flow which could be extracted from a downwind turbine. And the important information is only the difference of $P$ in percent, as it is a parameter study. Our intent with this was only to give a comparison also in case of power not only of m/s, however, it was very missleading instead of helpful and the information in % could likewise be calculated from $u$ directly. Therefore, we excluded the power completely from our revised manuscript. In the revised version we only refer to a spanwise and streamwise velocity difference between clockwise and counterclockwise rotating simulations.

4. It would be nice to report the tip speed ratio, the thrust coefficient and the power coefficient, for each case. This information is necessary to replicate your simulations.

Our representation of the wind turbine parametrization does not rely on thrust or power coefficients but rather lift and drag coefficients are applied in the calculation of the wind-turbine forces, as explained in Englberger and Dörnbrack (2017). We recognize the importance a reader is able to replicate the simulations. For reasons of space and because it is a very long manuscript anyway, we do not include all wind-turbine values applied in the BEM method. Instead, for all other wind-turbine parameters we refer to Englberger and Dörnbrack (2017). In addition, in the revised version we included the rotational frequency of each individual simulation in Table 1.

5. Table 1: This table is confusing. Why do you have both clockwise and counter-clockwise at the same row? I also get confused with the amount of cases and labeling. You also use these labels all the time in the article and it makes it hard to follow the text. Couldn't you simplify the cases and pick those that are really important for your conclusions? The intention was to read it as 'Simulation with different rotational directions of the rotor' 'clockwise' vs. 'counterclockwise' (all in capital letters and bold). This should save us one additional column stating the rotational direction and we think it is applicable as all other parameters are the same in the clockwise as well as the counterclockwise rotational direction simulations referring to one case. In the modified version we explained this in the table caption.

Regarding your comment with the labelling, we changed it to make it more intuitive with _ds for directional shear, _u for geostrophic wind speed, and _Ω for rotational frequency with the corresponding figures following for $ds$ and $u$ and values for low, high and very high in case of the rotational frequency.

Following your comment, we simplified the cases and now we only consider three parameters: $u_g$, $ds$, $\Omega$

6. Table 1: How is it possible to get a negative and positive wind veer for the same Coriolis parameter? This seems to me that your inflow profile is not in balance with your equations because these kind of effects are typically caused by (unsteady) meso-scale phenomena, which you are not modelling, as far I can understand.

Veering or backing is defined by the inflow profile. To make it more clear that veering or backing is prescribed by the background flow field, not resulting from a precursor simulation, we excluded the Coriolis force from the simulations. The Coriolis force is only relevant for determining the mean inflow profiles (veering or backing inflow) but not its interaction with the wake is leading to the differences (this was another misinterpretation of the results).

7. I would expect the wake deflects differently for different rotor rotation directions or the wake deflects simply more for a certain rotor direction. Is this correct? It might be worthwhile to discuss and show this (for example with a wake center tracking method). In addition, if you had a downstream wind turbine in unfavourable staggered position, then the additional wake deflection could reduce the power of the downstream wind turbine.

Yes exactly. Our simulation results show that the wake deflects more in case of a counterclockwise rotating rotor operating in veering inflow (or a clockwise rotating one in backing inflow). We added horizontal lines of the streamwise velocity in the lower and the upper rotor part as well as at hub height. See Figs. 6, 13 and 17. They allow a quantitative evaluation of the differences in the wake deflection angle between counterclockwise and clockwise as it was possible from the contour plots in the original manuscript version.

We thought about your comment of unfavourable positions of a downwind turbine. In the revised version of the manuscript we elevated the 90° sector and time averaged grid points with and 0 m$<$ r $\leq$ R for the top and the bottom sector directly behind the wind turbine. Than we extracted the information at 7 D and added the same information at a spanwise distance of $y$ = 1/2 D and -1/2 D at $x$ = 7 D. The results are not presented in the new manuscript (basically as it is an extension and not directly related to the analysis section), however, here we would like to show you the plot in Fig. 1. Considering the horizontal profiles in the lower and the upper rotor half at $z$ = 75 m and at $z$ = 125 m in Figs. 5, 6, and 7, the wake is deflected in the lower rotor part towards the left (right) and in the upper rotor part towards the right (left) in case of veering (backing) inflow. As the lateral wake position depends on the inflow wind angle, the spanwise wake position approaches away from $y$ = 0 D for increasing directional shear. This is presented in Fig. 1 (here). In case of no wind veer (Fig. 1(b)), there is no difference. In case of veering inflow (Fig. 1(e)), at $y$ = -1/2 D, there is a small rotational direction difference in the bottom rotor part, and at $y$ = 1/2 D, there is a difference in the top rotor part. The top left ($y < 0$) and the bottom right rotor parts are unaffected by the rotational direction, as there is no wake in these sectors. In case of a backing wind (Fig. 1(h)) the situation is the opposite. In case of a veering inflow, increasing the geostropic wind (Fig. 1(c)) or the directional shear (Fig. 1(f)) increases the difference in $\overline{u}$, especially in the top right rotor part. The same is valid for an increase of the rotational frequency (Fig. 1(i)). Decreasing the atmospheric or vortex strength, the difference decreases. Therefore, there is an impact at $y$ = 0 D and likewise in the wake affected sectors to the right or the left. In the considered idealized simulations of this work, the impact on $\overline{u}$ has therefore the same tendency in case of staggered or unstaggered arrangements of the hypothetical downwind turbines.

8. Figure 1, why not just plot wake deficit profiles as function of the cross coordinate at different downstream distances? Now you are only looking directly behind the wind turbine, while the wake has deflected laterally (and possible vertically as well), so you are missing a lot of important information.

This is answered in point 7. in detail.

9. The streamwise velocity contour plots presented of Figures 3, 5 and 6, do not seem to resemble converged statistics. If you have converged statistics, then I expect smooth plots, see for example the low turbulence intensity case in van der Laan

[Figure]

**Figure 1.** Sector averages of $\overline{u}$ representing the top and bottom 90°-sectors for 0 m < r ≤ 50 m for clockwise and counterclockwise rotating actuators for the same simulations as in Fig. 19 of the manuscript at $y = 0$ D and in addition shifted by D/2 in both lateral directions. The indices 'b' and 't' at the top x-axis represent the corresponding bottom or top sectors.

and Andersen (2018), where 1 hour LES results are presented. This could indicate that your LES data set is not large or long enough, or your simulation has not converge to a (quasi) steady-state, but keeps changing instead.

- We extended the reference simulation to 1.5 h and the wake structure did not change.
- In Englberger et al. (2019), simulations with both rotational directions are conducted with the inflow resulting from the stable regime of a diurnal cycle precursor simulation. In these simulations the spanwise flow component was 8 times it is chosen for this parameter study and the effect did not occur. Therefore, we refer the wake structure to the domain size. We cannot reproduce the simulations in this work on a larger spanwise domain as the applied NBL precursor simulation in the turbulence generation method limits the domain size.

– The effect not only occurs for counterclockwise rotating rotors. This is supposed to be the case as the spanwise inflow velocity is amplified in the wake. Considering a geostrophic wind of 6 m $s^{-1}$ in CR_u6 it also occurs. Here, the spanwise flow component decreases in comparison to the reference case, and likewise the streamwise component. Therefore, the effect seems to be additionally influenced by the streamwise wind speed.

10. Conclusion and abstract: You have to mention that the simulated power increase of 23% only reflects a specific wind direction. In other words, if you would consider multiple wind directions, then the impact of rotor rotation direction on the power (deficit) is much smaller than you report.
We modified the introduction and listed in detail that

– This difference occurs only at night.

– There are seasonal differences.

– The percentage of occurrence is location dependent

– The occurrence of specific directional shears and wind speeds is also location and also seasonal dependent.

we modified the conclusion by: This is only an idealized parameter study (turbulence is not location sensitive or result from an SBL precursor simulation). The results are not valid everywhere.

In addition, if a full wind farm is considered, I expect that the effect of rotation direction is reduced further downstream in the wind farm because of an increase in turbulence level.
Further we added a common on wind farms: 'We have assessed the wake of an individual turbine, but these results could be extended to a large farm in which the presence of upwind turbines could affect turbulence intensity, which probably affects the magnitude.'

Finally, if one would look at the effect of the rotor rotation on the wind farm annual energy production, which also consists of many flow cases, where rotor rotation has no influence, you might find that the effect of rotor rotation direction is far less than 1%. Such the study would be necessary in order to answer the question raised in the title. If your title is Should wind turbines rotate in the opposite direction? then I expect to find a thorough answer in the article. The presented simulations cannot answer this question because we need an estimate of the rotor rotation direction on the annual energy production. Regarding your comment on wind direction, there is a misunderstanding. We do not simulate a specific wind direction (for a specific location). We only simulate different directional shear values in an idealized simulation set-up. But we agree that a wind direction change occurs mainly at night and a veering wind only represents a certain precentage of this nights. (See listed modifications of the introduction.) Further, we agree with your comment on the title and changed it as explained in the general comments.

**Correspondence:** Antonia Englberger (antonia.englberger@dlr.de)

Dear Reviewer 2,

Thank you for taking the time to carefully review our paper. We read your review in detail and appreciate you sharing your own simulation results. Regarding your comments (especially your comments asking what exactly contributes to higher/smaller $\overline{u_A}$ values), we think there are several misunderstandings with the first version of the paper. Therefore, with the help of your comments, we performed some far-reaching changes to the manuscript. Here is a list of the major changes.

- We changed the title.
- We explained in detail the turbulence generation method we applied in the simulations.
- We included a section, introducing a simple analytical model predicting the expected changes in the spanwise velocity field in the wake by a superposition of a veering inflow with a Rankine vortex. (New section 3)
- We added additional simulations with different directional shears.
- We investigated the impact of the rotational frequency on the wake differences.
- We added additional plots, explaining the wake differences and its occurrence for different rotational direction of the actuator.
- We added a section comparing the numerical results predictions of the analytical model. This section explains in detail the source of the difference in the wakes between a clockwise and a counterclockwise rotating rotor in case of a veering inflow.
- We added an Appendix, verifying the application of the turbulence preserving method for this theoretical and idealized parameter study.

In the following we respond in detail to each of your comment/question.

**General Comments**

The research question of the article is interesting and well-motivated. I cannot comment on the technical set-up of the LES and the turbine model, as I have no experience with modeling, but some of the chosen simulation parameters seem questionable to me. There are several issues with the results:

(i) A presentation of the wake structure away from the hub height is missing.

We added several figures covering this. Contour plots representing the top rotor half at $z = 125$ m and also the bottom rotor half at $z = 75$ m (Fig. 4, 9, 11, 12, 15, 16). We also included a y-z contour plot representing the difference in the wake skewing between clockwise and counterclockwise rotating actuators (Fig. 5). Further, for a quantitative comparison we added vertical and spanwise profiles of the streamwise velocity (Fig. 6, 13, 17).

(ii) The exclusive focus on the mean streamwise velocity ignoring other quantities that affect a downstream turbine (I am not counting the power as a separate quantity here due to way it is computed).

We eliminated the power from the paper, as is leads to many misunderstandings.

In a recently submitted revised version of a previous paper Englberger et al. (2019) (attached) we apply the inflow conditions from a stable regime from a diurnal cycle precursor simulation. In that work we also focus on the turbulence in addition to the velocity components.

This work, however, is a parameter study with a very simplified setup of the numerical simulations. The applied turbulence is based on a turbulence generation method from Englberger and Dörnbrack (2018b), applying the turbulent perturbations of a neutral boundary layer precursor simulation (Englberger and Dörnbrack, 2017) in combination with adjustable-stratification dependent parameters resulting from this stable regime of Englberger and Dörnbrack (2018a), which is applied directly in Englberger et al. (2019). We apply this turbulence generation method as it provides a computationally fast testbed for wind-turbine simulations with open horizontal boundary conditions on a small domain and it also includes atmospheric characteristics in the inflow (not only random perturbation). This allows us to produces the large number of simulations in this work. We consider this method appropriate, as the occurrence of the differences between clockwise and counterclockwise rotating turbines results from the veering inflow and only the degree of the differences is modified by the turbulent intensity applied (see Appendix). Therefore, we only show velocities in the manuscript.

In the revised version, however, we also included the spanwise and vertical velocity. Further, we show vertical and horizontal profiles at different heights over the rotor of the streamwise velocity, not only the rotor averaged value as in the original manuscript version.

(iii) No physical explanation is given how the stronger rotation of the wake causes the higher entrainment, which is provided as reason for the main finding.

This is given in Englberger et al. (2019), where the turbulence profiles are presented. Here, due to the limitations of this work as listed above, it is not shown.

(iv) I am not convinced that the increased entrainment is the sole reason for the higher streamwise mean velocity across rotor of the downstream turbine and a modification of the spanwise advection influencing the shape of the wake should be investigated, too.

The main reason for the striking difference between clockwise and counterclockwise rotating rotors under veering or backing inflow presents the amplification or reduction/reversion of the spanwise flow field. To present and discuss this, we added Fig. 2. A $y$-$z$-cross section plot for veering and no veering inflow simulations at $x = 3\,\mathrm{D}$ for clockwise and counterclockwise rotating simulations. The first row presents the $(v,\, w)$ vectors in the $y$-$z$-plane, the second row the spanwise wake velocity $v$, and the third row the vertical wake velocity $w$. The figure shows a striking difference in the spanwise flow field between clockwise and counterclockwise rotating rotors and also in comparison to the difference between both rotational directions in case of $\frac{\partial v_f}{\partial z} = 0$.

The conclusions do not account for the limitations of the study and its applicability is overestimated. Therefore, the rather definitive answer to the research question provided here does not hold in my opinion (but there could be an argument to pursue the research question further).

We agree with your comment. Therefore, we added the limitations of this work to the introduction:

- Veering tends to occur only at night.
- Veer shows seasonal variability.
- The frequency of occurrence is location dependent
- The occurrence of specific directional shears and wind speeds is also location and also seasonal dependent.

and in the conclusion:

- This work is an idealized parameter study (turbulence is not location sensitive or results from an SBL precursor simulation). The results are not valid everywhere.
- Transferring the results of this study to a wind farm, the presence of upwind turbines has an effect on the turbulence intensity, which did not affect the occurrence of the difference, but its magnitude (see Appendix). Therefore, the rotational direction impact on the power production of a wind farm is another open research topic.

We also excluded any referring to a preferential rotational direction. We only stated that there are differences in the wake in case of $\frac{\partial v_f}{\partial z} \neq 0$. And added the limitations of this work, as it is not valid for every location etc. (see above)

Further we changed the title, excluding the question at all. The question was chosen as title for the paper as it is simple and interesting and for motivation to consider this issue. But we agree with the reviewer that our paper cannot answer this question as it is only a simplified parameter study. Therefore, we change the title of our manuscript to 'Changing the rotational direction of a wind turbine under veering inflow: A parameter study'

**Specific comments**

Page 2, lines 12-14: Sentence should be narrowed to the mixed layer in absence of synoptic or mesoscale forcing.

This is no longer included in the manuscript.

Page 2, lines 17-19: From the text, it could be misunderstood that the wind veer resulting from the influence of friction is directly connected to temperature advection and lifting. Therefore, I would propose to change the sentence ("This wind veer is associated with...") to something like ”Besides the surface friction, temperature advection and dynamic lifting also influence the veering of the wind".

Thank you, we changed it according to your suggestion.

Page 3, lines 9-11: Vasel-Be-Hagh and Archer, 2017 (https://doi.org/10.1016/j.seta.2016.10.004) studied counter-rotating rows of wind turbines in a wind farm and mentions different wake characteristics for the counter rotating turbines.

Thank you, we included the paper together with the 1.4% power increase of a wind farm with clockwise and counterclockwise rotating wind turbine rows in case of no wind veer.

Page 4, lines 9: The rotor diameter is a third ofthe height and the width ofthe simulation domain. Can this affect the wake development? Also the temperature inversion is 50 m above the top tip of the turbine, which corresponds to a very shallow boundary layer. Would a higher inversion layer have an influence on the results?

The spanwise extension of the wake probably has an influence on the streamwise velocity. The averaged x-z contour plots are not smooth for the counterclockwise rotating simulations. This is supposed to be the case as the spanwise inflow velocity is amplified in the wake. The effect not only occurs for counterclockwise rotating rotors. Considering a geostrophic wind of 6 m $s^{-1}$ in CR_u6 it also occurs. Here, the spanwise the the streamwise flow component did change size in comparison to the reference case. Therefore, the effect seems to be additionally influenced by the streamwise wind speed.

In Englberger et al. (2019), simulations with both rotational directions are conducted with the inflow resulting from the stable regime of a diurnal cycle precursor simulation. In these simulations the spanwise flow component was 8 times it is chosen for this parameter study and the effect did not occur. Therefore, we refer it to the domain size. We cannot reproduce the simulations in this work on a larger spanwise domain as the applied NBL recursor simulation in the turbulence generation method limits the domain size.

The shallow boundary layer is also related to the domain size of the simulations. In Englberger et al. (2019) the inversion layer starts higher above, but the impact of the rotational direction is still present.

As it is a parameter study which requires a computationally faster method in comparison to the simulations in Englberger et al. (2019) in order to run all various simulations and as the occurring difference is in agreement with the analysis predictions, the spanwise and vertical domain size limitations are not responsible for the rotational difference in the wake.

Page 4, lines 27-29: What is the reasoning for choosing the lower rotor area in contrast to the upper rotor area to modify

the type of wind veer? While it is difficult to say anything general about a stable boundary layer, at least for textbook cases the wind veer is stronger in the upper part (opposed to convective boundary layer where wind veer stronger near the surface layer). In addition, the effect is presumably larger in the upper part, because the wind speeds are higher due to wind shear.

The reason was the Ekman spiral in case it is only affecting the lower rotor region with no significant veer in the upper rotor region. See modified attached version of Englberger et al. (2019) in Fig. 3. However, we excluded the simulations with veer limited to the lower rotor area.

As you mentioned in your summary, we investigate 'the influence of the stratification, the magnitude and structure of the wind veer, and the wind speed on this result is also investigated to some extent'. To make this study more consistent, we included the analytical predictions and prepared the numerical simulations only for the corresponding cases. In the revised version, we considered cases of changing the atmospheric inflow conditions (geostrophic wind, directional shear) and the rotational frequency. All other simulation results from the previous manuscript version are eliminated (including veer limited to the lower rotor part) to be more consistent. With including the expected results in the analysis section (3) and comparing the simulation results to them (section 5), our methodology is now more consistent.

Page 5, Eq. 8: Is 00 changed for the very stable case? Otherwise, there is an unstable layer above the hub height, because the pot. temp is 306 K at 200 m 303 K above and that would influence the dynamics for this case.

Yes, sorry this was a typo. It is no longer included in the manuscript as these simulations are eliminated.

Page 5, Eq. 9: That should be $\overline{u_A}^3$ instead of $\overline{u_A}$ and since all else is constant, the available power could be used instead.

Our very simple power calculation is basically proportional to $u^3$ (sorry we missed the 3 in Eq. 9) should represent the power available in the flow which could be extracted from a downwind turbine. Our intent with this was only to give a comparison also in case of power differences in % not only of m/s, however, it was very missleading instead of helpful for reviewers. Therefore, we excluded the power completely from our manuscript. In the revised version we only refer to a spanwise and streamwise velocity difference between clockwise and counterclockwise rotating simulations. The difference in % can also be calculated from $\overline{u_A}$.

Page 6, lines 9-10: In Engelberger et al. (2019) — Fig. 8 it is shown that the consistent wake cases have a stronger rotation of the wake compared to the contrasting wake cases at x/D = 7. This means that the downwind turbine is receiving a stronger wind veer for the consistent cases compared to contrasting wake cases (beside the higher $\overline{u_A}$ shown here). That stronger wind veer would presumably impact the power of a downwind turbine negatively. Maybe the downwind turbine could be viewed as a yawed turbine for the upper / lower rotor part and Eq. (9) modified to use an adapted power coefficient for each sections of the rotor.

You are right, please see Fig. 11 and 12 of the revised manuscript version of Englberger et al. (2019) (attached). The turbulent intensity is slightly larger in case of a counterclockwise rotating rotor at 7 D in all rotor heights. His would impact the hypothetical downwind turbine.

Section 3 in general: Spanwise plots of the streamwise velocity at x/D=7 similar to Fig. 3, 5 and 6 should be shown and discussed. I understand that $\overline{u_A}$ is including values above and below the hub height, but in my opinion this is not sufficient to understand the effect of the direction of the rotor rotation and wind veer on the wake structure. Further insights into the mechanism might be gained by looking at turbulent momentum transport or turbulence production, if available from the LES. A few of the following comments reiterate this comment for the specific subsections.

We included a y-z plot of the spanwise and vertical velocity (Fig. 3) and also of the streamwise velocity (Fig. 5). We also include x-y plots also in the upper and the lower rotor half in Figs. 4, 9, 10, 12, 15, 16. Further, we included vertical and horizontal profiles of the streamwise velocity at three specific rotor heights in Figs. 6, 13, and 17. The turbulence profiles are shown in the previous study of Englberger et al. (2019), where the 2 D slices of all three wind components as well as the potential temperature are applied as upstream inflow condition at each time step. As this is a very simplified parameter study and due to the applied turbulence generation method, we decided showing the turbulence profiles is not helpful.

Page 9, lines 12 — 15: I have three questions on this. First, after a look at the model from Engelberger et al. (2019), I do not yet understand the distinction between entrainment and wake recovery and why entrainment is considered as the explanation for the observations. Second, do the authors have any notion why the entrainment is larger for the consistent wake case in a physical sense? For example, whether the consistent wake cases have larger gradients of the absolute value of the wind vector due to the rotation, which might facilitate a stronger turbulent momentum transport. Is the turbulent momentum transports from the LES available to investigate this?

To give an explanation for the difference we updated both manuscript versions (also Englberger et al. (2019)) including a very simple analytical equation, which is the superposition of the spanwise veering inflow equation with the spanwise component of a Rankine vortex (Eq. 17). This shows that the amplification of the spanwise flow component in case of a counterclockwise rotating rotor in case of veering inflow in the NH is responsible for the difference in comparison to a clockwise rotating rotor in which the spanwise flow component is weakened/reversed due to the superposition of the vortex component. This different behaviour of the spanwise flow component impacts the streamwise flow and results in larger turbulent intensity values in case of a counterclockwise rotating rotor. The larger turbulence resulting from the amplification of the wake in case of a counterclockwise rotating rotor results in a larger entrainment rate and therefore in a more rapid wake recovery in comparison to the clockwise rotating case.

Third, I wonder whether a more pronounced ellipsoidal wake cross-section might contribute to a higher $\overline{u_A}$ beside entrainment? Looking at Fig. 6 in Engelberger et al. 2019, an increase of the veer in the wake by the consistent wake cases could make the wake more ellipsoidal. This in turn could cause parts of the wake missing the rotor area of the downstream turbine and increase $\overline{u_A}$, too.

In fact there are two differences contributing. The larger wake deflection angle and the larger spanwise wake width in case of a counterclockwise rotating rotor result in larger $\overline{u_A}$ values.

[Figure]

**Figure 1.** Vertical (first column) and horizontal profiles at different heights for the CR and CCR reference cases.

To show this in the paper, we added a similar figure as in Englberger et al. (2019) for both rotational directions in this work (see Fig. 5). The different lateral elongation of the wake can lead to this assumption for the outer part of the top and bottom sectors. To investigate it in more detail, we also include vertical and horizontal profiles of $\overline{u}$ at specific heights. According to the vertical profile (Fig. 6a), this can be assumed. Looking at the upper and lower rotor half profiles (Fig. 6b, d), the streamwise velocity is larger in case of a counterclockwise rotating rotor. Looking at the same plot in Fig. 1 (only added in this response, not in the paper) at $z = 55$ m or 145 m in the first row, at $z = 65$ m or 135 m in the second row and at $z = 85$ m or 115 m in the fourth row, the difference increases for increasing the radial distance to the nacelle. Therefore, in the top and bottom sector this will certainly contribute to $\overline{u_A}$. This increase in $\overline{u_A}$ in case of a counterclockwise rotating rotor is related to the larger wake deflection angle in case of a counterclockwise rotating wake.

Further, looking at the profile at $z = 100$ m, and also at 85 m and 115 m, the spanwise wake width is larger in case of a counterclockwise rotating rotor. This difference, which is especially pronounced in the right and left sectors, also contribute to the larger $\overline{u_A}$-values in case of counterclockwise rotating actuators.

The larger $\overline{u_A}$(a) values in Fig. 7 are therefore a result of the larger wake deflection angle and the larger spanwise wake width in case of a counterclockwise rotating simulation comparing the reference case CR and CCR.

Page 9, lines 27-29: Linking stability directly to the time of day requires the assumption of a radiation driven diurnal cycle of the boundary layer with the absence strong synoptic or meso-scale forcing. The same for page 12, lines 8-11.

We agree. As including different levels of the atmospheric stability is rather complex and it is not explained by the simple analytical equation, we postponed this results and will investigate them in more detail in the future.

Fig. 4: Panel b is quite busy. Would it be possible to make this figure a four panel figure and separate the weak, moderate and strong wind veer cases in one panel and the cases with only the lower rotor area affected by wind veer in a second panel? That would be also more consistent with the subsection structure used in the text.

We agree. As we eliminated a few of the simulation, we only result with one figure showing $\overline{u_A}$. The corresponding figure 7(e) includes the same amount of profiles as old figure 4(b), however, now the only consider a different amount of directional shear and therefore the lines are not crossing etc. as before. Due to these changings, we leave the result for all simulations with varying the directional shear in one panel as it makes it easier for the reader to see the difference in the wake if the directional shear is changed.

Page 12, lines 6-9: I believe the phrasing of this sentence is unfortunate, because it could be misunderstood that the power improvement of the downstream turbine itself becomes larger with longer duration (the percentage values from the previous sentence increase over time).

We agree that it could be misunderstood. We eliminate this sentence as the potential temperature varying simulations are not longer included.

Page 12, lines 18-21: This sentence explains the difference between CR and CCR, but not the difference between CCR_th60 and CCR_th15/CCR. The faster wake recovery for more stable stratification (and presumably a subsequently lower turbulence intensity) for the consistent wake cases is still counter intuitive to me. Do the authors have any explanation what is causing that behavior?

As the investigation with different background potential turbulent profiles is rather complex and cannot explained with the simply analytic equation, it is excluded from this paper and we will investigate it in more detail in the future.

Page 12, lines 25: As for the comment on page 9, lines 12-15, I believe it is possible that an increased ellipsoidal wake shape with increasing wind veer might have a pronounced effect on $\overline{u_A}$ beside entrainment. Vertical cross-sections of the streamwise

velocity and plots of the momentum transport could be used to investigate. Maybe some insights into the curious decrease for the strong wind veer case might be gained from them, too.

To investigate this in more detail, we perform two additional simulations with a directional shear of 0.12° $\mathrm{m}^{-1}$ and 0.20° $\mathrm{m}^{-1}$. According to our results there is a critical directional shear value $ds_c$ with 0.12° $\mathrm{m}^{-1} < ds_c <$0.16° $\mathrm{m}^{-1}$. Below this critical values, the $\overline{u_A}$-value is larger for a counterclockwise rotating rotor, whereas above it is larger in case of a clockwise rotating one.

The new figure 13 gives some insight into this. An increase of the directional shear increases the wake deflection angle. However, increasing the directional shear to high values of 0.16° $\mathrm{m}^{-1}$ and even very high values of 0.20° $\mathrm{m}^{-1}$ results in a larger streamwise velocity close to the nacelle, contributing to larger values especially in the left and the right 90° sectors (last two rows of Fig. 13). This overcomes the larger streamwise velocity values in the top and bottom sectors in Fig. 13 at 75 m and 125 m respectively. Therefore, the more rapid wake recovery for large directional shear values results in larger $\overline{u_A}$-values in case of clockwise rotating discs.

In the paper it is explained with: 'In the clockwise as well as the counterclockwise rotating actuator simulations (Figs. 10 - 12) the wake recovers more rapidly if directional shear increases. A larger directional shear represents a larger resolved turbulence source due to an increase of $\frac{\partial v_f}{\partial z}$, and, therefore, the simulations with higher directional shear values result in higher entrainment rates and a more rapid wake recovery.'

Page 13, line 1-2: Is this amplified the turbulence production occurring at specific regions of the wake? Could the terms of the TKE budget provide any insights into the cause of the higher entrainment (if they can be computed from the LES)?

This has to be tested in LESs applying precursor simulations of the SBL for different directional shears. This is one of our planned next steps.

Page 15, lines 5-6: I would always expect a larger $\overline{u_A}$ for an increased inflow wind speed if the efficiency of the upwind turbine is not changing (as it is the case here) and I am not seeing where the entrainment is entering the picture from the results. Is that sentence referring to the relative difference between CR / CCR and CR_u14 / CCR_u14?

Yes we agree with your expectation. A larger wind speed has an effect on the streamwise wake elongation. Yes, the specific sentence is referring to the relative difference between CR/CCR and CR_u14/CCR_u14.

Section 3.6 and Fig. 7: I like this section bringing everything together and the figure is very informative, but I had a hard time reading the first two paragraphs of this section due to the amount of simulation abbreviations. Since the simulations can be deduced from the Fig. 7, perhaps the text could focus on the physical meanings. E.g. "The blue square shows a power increase by 4% for counterclockwise rotating turbines compared to a clockwise rotating ones for a weakly stable stratification." instead of “The point 'th15' represents a power increase by 4% at 7D for CCR_th15 in comparison to CR_th15".

Thank you. Actually, due to all new figures we extracted this figure and also the text. See Fig. 2 (only here) as updated version.

[Figure]

**Figure 2.** Coloured contours of the streamwise velocity $\overline{u_{i,j,k_*}}$ in m s$^{-1}$ for different geostrophic winds at $z = 75$ m. The black contours represent the velocity deficit $VD_{i,j,k_*}$ at the same vertical location.

Page 17, line 6-7: It should be specified that the power of the waked downstream turbines is considered here (it could be misunderstood that the power of the upwind turbine improves, too).

Thank your for this hint. As we excluded power and only discuss the velocity of one wind turbine, no misunderstandings like that should be possible.

Page 17, lines 22-24: How much of that cumulative capacity is located in wind farms, where wake effects can occur? (in contrast to isolated turbines where it would not matter).

This is an interesting question. In the GEWC there is no distinction between offshore and onshore. As we extracted the power and the discussion about any preferential rotational direction, we also extracted the NH and SH comparison of installed capacity.

Page 17, lines 28-29: I believe there is a need for further studies on some aspects to this question:

We agree and added are complete paragraph about this:

’To explore a more comprehensive assessment of the wake impact, further investigations would be interesting. The investigation of the non-linearity of the interaction process, numerical simulations applying the turbulence of a SBL precursor

simulation for different strengths of stratification and directional shears, or even considering a low-level jet at the rotor height. Topography could influence the wake dynamic explored here. We have assessed the wake of an individual turbine, but these results could be extended to a large farm in which the presence of upwind turbines could affect turbulence intensity, which probably affects the magnitude. However, an important point will be to prove the theoretically predicted effect resulting from superposition of inflow veer with the vortex component on the wake with measurements. '

1: This conclusion is based on numerical simulations with simplified a very simplified estimation of the downstream turbine power. A verification with experiments for real wind turbines would be a reasonable call. 2: Unstable and neutral stratification of the boundary layer is not regarded in this study, but can be subject to wind veer as well. 3: Real wind turbines have an induction zone that modify the flow further from the simulation results. 4: Besides the higher streamwise velocity investigated here, the wake structure could see further changes (turbulence intensity, veer, shear), which could impact a downstream turbine. 5: It is possible that two important categories of wind farm locations have a different veering/backing ratios then considered here. Offshore wind parks in proximity to a coast due to the baroclinicity between land and sea. Wind farms located on a ridge due to topography and baroclinicity.

1: The power is eliminated and we now stated in the conclusion: 'However, an important point will be to prove the theoretical effect resulting from superposition of inflow veer with the vortex component on the wake with measurements.'

2: We only focus on veering and backing in nighttime situations following Walter et al. (2009).

3: We agree, this will modify the streamwise velocity at the downwind turbines location. Here, however, we compare the streamwise velocities for both rotational directions with no downwind turbine in both cases. Therefore, the difference between clockwise and counterclockwise is comparable.

4: We investigated more aspects in the revised version of the paper with including the horizontal and vertical profiles of $u$.

5: We listed possible differences related to topography and location in the conclusion in the revised version of the manuscript.

**Correspondence:** Antonia Englberger (antonia.englberger@dlr.de)

Dear Reviewer 3,

Thank you for taking the time to carefully review our paper. We read your review in detail and appreciate you sharing your own simulation results. Regarding your comments, we think there are several misunderstandings with the first version of the paper. Therefore, with the help of your comments, we performed some far-reaching changes to the manuscript. Here is a list of the major changes.

– We changed the title.

– We explained in detail the turbulence generation method we applied in the simulations.

– We included a section, introducing a simple analytical model predicting the expected changes in the spanwise velocity field in the wake by a superposition of a veering inflow with a Rankine vortex. (New section 3)

– We added additional simulations with different directional shears.

– We investigated the impact of the rotational frequency on the wake differences.

– We added additional plots, explaining the wake differences and its occurrence for different rotational direction of the actuator.

– We added a section comparing the numerical results predictions of the analytical model. This section explains in detail the source of the difference in the wakes between a clockwise and a counterclockwise rotating rotor in case of a veering inflow.

– We added an Appendix, verifying the application of the turbulence preserving method for this theoretical and idealized parameter study.

**Referee comments**

The authors argue that counter-clockwise rotation wind turbines in northern hemisphere (as opposed to clockwise as is currently done) can lead to a power increase of 11% in the downwind turbine due to constructive interactions between the axial vorticity in the wake and veered Ekman layer, especially when strong stable stratification is present. While I am fascinated by the overall theme of this research, I do not feel that the authors have done a thorough investigation to corroborate their hypothesis. While the paper uses a provocative title and well written, I hesitant in recommending publication at this time since I have the following serious concerns regarding the quality of the numerical simulations performed.

The intent of the manuscript was not to be provocative. The question was chosen as title for the paper as it is simple and interesting and for motivation to consider this issue. But we agree with the reviewer that is could lead to misunderstandings. Therefore, in the revised version, we changed the title to 'Changing the rotational direction of a wind turbine under veering inflow: A parameter study'

Further, there seem to be some misunderstandings with the simulations. The simulations in this manuscript are wind turbine simulations performed under prescribed wind and turbulence conditions. In the parameter study presented in this work, we applying a very simplified set-up with a turbulence generation method. This is not a stable boundary layer input applied in the wind-turbine simulations. But we agree the manuscript could give the impression as we talk about veering wind in a stably stratified regime. In the revised version of the manuscript we only talk about a veering inflow or a backing inflow or no veer at all. Further, we added a detailed explanation of the turbulence generation method, instead of only referring to the corresponding paper. We also added the basic equation for this. The modification should make it clear that no SBL LES is performed. Further, we rerun all simulations as implicit LES also excluding the Coriolis force. With that we would like to make clear that it is only an idealized parameter study, and the Coriolis force has only an effect on the prescribed inflow wind field whether the resulting differences between clockwise and counterclockwise rotating turbines not results from an interaction effect of the vortex with the wake. It is not affected by the Coriolis force interacting with the wake and delecting it.

We apply the turbulence parametrization instead of the SBL precursor simulation as it provides a computationally fast testbed for wind-turbine simulations on a small domain. Regarding the large number of performed simulations, it would be computationally very expensive running them all as SBL simulation and a resolution refinement down to 0.25 m is not possible with the current supercomputer resources we can use. Especially considering the effect that the simulations with varying wind speed and directional shear would require different precursor simulations to conduct the SBL wind-turbine simulations.

This parameter study with a very simplified numerical setup was the first attempt to investigate the impact of the atmospheric parameters (geostrophic wind speed and directional shear) and the impact of the vortex paramter (rotational frequency) on the wake differences between clockwise and counterclockwise rotating actuators. The results allow us to identify which SBL precursor simulations are required to investigate the interesting cases in detail in future simulations.

1. The Ekman layers being simulated are highly stratified with very high gradient Richardson numbers. TKE based eddy-viscosity SGS closures are notoriously terrible at stably stratified layers; see the work by Sullivan et. al, (JAS, 2016) where they show grid sensitivities up to 0.25m for similar states of stratification. You must show that the Ozmidov scale is larger than the grid scale, especially for your strongest stratification case for me to accept the accuracy of the SBL simulated using your SGS closure. This is not done in the current version of the manuscript.

The presented simulations represent a wind-turbine simulation with prescribed wind and turbulence conditions. It is not an SBL simulation with a rather fine resolution close to the ground. To make this clear, we rerun all simulations as ILES excluding the SGS closure. The applied resolved turbulence develops from small fluctuations impressed on the flow field by our 'turbulence preserving method'.

2. Since much of the argument made in the paper relies on axial vorticity, the authors need to present a strong case showing that the axial vorticity captured by the their grid resolution and actuator-line parameterization is correct. A grid convergence study might help, although I remain skeptical regarding whether actuator lines can correctly represent axial vorticity. There is substantial discussion on this topic in open-literature.

There is a misunderstanding, we did not apply an actuator line technique, we run the simulations with an actuator disc approach. The disc is resolved with 21 grid points. Following Ivanell et al. (2008), Wu and Porté-Agel (2011), and Gomes et al. (2014), the minimum number of grid points to result in the same resolution independent wake structure for acutator disc models is 10 grid points in vertical and spanwise direction.

3. There is new evidence that suggests that ignoring the horizontal component of Earth's rotation (as the authors have done) has a significant quantitative impact on wakes of large turbines representing small Rossby numbers. See the recent work by Howland et al. (2020, JFM) on this topic. Even at approx.. 45deg. Latitudes, I would speculate the direction of wind (Westerly vs Easterly) would affect the power of the downwind turbine by similar order of magnitude as shown by the authors for CW vs CCW rotation.

In the present study, we only consider the Coriolis force as cause for the inflow profiles. To make this clear, we rerun all simulations without a Coriolis force.

These only recently published results, however, are rather interesting and we will include the horizontal component of the Coriolis force in the fine resolved SBL WT simulations we plan to perform next.

. In this work, a common clockwise rotor rotation 'cr' is defined as $\beta_v$ = 1 and $\beta_w$ = -1 and a counterclockwise rotor rotation 'ccr' as $\beta_v$ = -1 and $\beta_w$ = 1, with $\beta_u$ = 1 in both cases.

**2.2 Setup of the Wind-Turbine Simulations**

Wind-turbine simulations on 512 × 64 × 64 grid points with a horizontal and vertical resolution of 5 m and open  boundaries are performed for veering and non-veering inflow lasting 40 min. The rotor of the wind turbine has a diameter $D$ as well as a hub height $z_h$ of 100 m and is located at 300 m  downwind from the inflow boundary and centred in the spanwise $y$-direction.

$$u_{BL}(z) = u_g * \left(1 - exp\left(-\frac{z\sqrt{f/\kappa}}{\sqrt{2}}\right)\right),$$

A veering wind profile can be described by the Ekman spiral 2

$$u_{Ekman}(z) = u_g \cdot (1 - \exp(-z\gamma)\cos(z\gamma)), \quad (4)$$

$$v_{Ekman}(z) = u_g \cdot (\exp(-z\gamma)\sin(z\gamma)), \quad (5)$$

following Stull (1988), with a geostrophic wind $u_g$  and

$$\gamma = \sqrt{\frac{f}{2\kappa}}$$

representing a Coriolis parameter f = 1.0 × 10$^{-4}$ s$^{-1}$  and an eddy viscosity coefficient $\kappa$.

Wind direction change between two heights is defined as directional shear. As we assume the directional shear to be an impact factor for the interaction process of a rotating system with veering inflow, a modified version of the Ekman spiral is applied as $v_f$ in the simulations in this work. Further, the negative vertical gradient of the streamwise velocity in the supergeostropic component of the Ekman spiral is not considered in $u_f$.

The simulations are initialized with the streamwise velocity profile

$$u_f(z) = u_g * (1 - exp(-z\gamma)), \quad (6)$$

with an eddy viscosity coefficient $\kappa$ = 0.06 m$^2$ s$^{-1}$. The corresponding  spanwise velocity profile is

$$v_{BL\,f}(z) = 0 \quad (7)$$

in the case of no veering inflow with $\frac{\partial v_f}{\partial z}$ = 0, and

$$v_f(z) = u_{BL\,f}(z) * tan \cdot tan(\phi_{wind}(z)) \quad (8)$$

in the case of veering inflow with $\frac{\partial v_f}{\partial z} \neq 0$ with a given directional shear

$$ds = \frac{\Delta\phi}{100\ m}, \tag{9}$$

with $\cdot - \frac{z}{D}\phi_{50\ m}$ and $\phi(z) = \pm 2\Delta\phi\left(1-\frac{z}{D}\right)$

(10)

in the lowest 200 m and constant above. The influence of the Coriolis force on the flow field is only included in the simulations via Eqs. 6, 8. Note that no Coriolis force is applied in the numerical model (Eq. 1).

In the incoming wind conditions we consider on For $u_f$ and $v_f$, we consider the NH ($f>0$) and the SH ($f<0$), a veering ($\frac{\partial\phi_{wind}}{\partial z}\Delta\phi <> 0$ in NH, $\frac{\partial\phi_{wind}}{\partial z}\Delta\phi >< 0$ in SH), and a backing ($\frac{\partial\phi_{wind}}{\partial z}\Delta\phi >< 0$ in NH, $\frac{\partial\phi_{wind}}{\partial z}\Delta\phi <> 0$ in SH) wind. In the reference simulation (with a veering wind on in the NH, the wind direction change over the rotor radius is $\Delta\phi = 4°$ with $v_{BL}(z_h)$), the directional shear is 0.08° m$^{-1}$ with $v_f(z_h) = 0$. The The initial vertical velocity is

$$w_{BL\,f}(z) = 0 \tag{11}$$

in all simulations. The flow components $u_f$, $v_f$, and $w_f$ are used for specifying the initial conditions. The pressure solver in EULAG further applies $u_f$ and $v_f$ as boundary conditions. The potential temperature is

$$\Theta_e(z) = \Theta_0 + \frac{3\,K}{200\,m}z \tag{12}$$

in the lowest 200 m and 303 K above.

For a veering wind in the NH, we further modify the strength of $\Delta\phi$ over the rotorwith 2°, 4°, and 8° corresponding to a weak ($w$modify the geostrophic wind, the directional shear and the rotational frequency of the rotor. We sample winds with a geostrophic wind component of $u_g = 6$ m s$^{-1}$, $u_g = 10$ m s$^{-1}$ (reference simulation), and $u_g = 14$ m s$^{-1}$, referred to with the acronyms u6 and u14 in the simulation nomenclature. Further, we apply a directional shear of 0.04° m$^{-1}$, 0.08° m$^{-1}$, 0.12 ° m$^{-1}$, 0.16 ° m$^{-1}$, and 0.20 ° m$^{-1}$ corresponding to weak (ds4), moderate ($m$), or strong ($s$)veer. We also test the rotational direction sensitivity towards the rotor section interacting with $\Delta\phi$reference simulation), moderate to strong (ds12), strong (ds16) and very strong (ds20) shear. As additional parameter, the rotational frequency ranges from $\Omega \neq$=0.058° s$^{-1}$, $\Omega$ 0, limited to the lower rotor part ($l$)or extended over the entire rotor ($e$=0.12° s$^{-1}$, $\Omega$ =0.175° s$^{-1}$, to $\Omega$ =0.23° s$^{-1}$, corresponding to low (Ωl), moderate (reference simulation), high (Ωh), and very high (Ωvh) in the simulation nomenclature.

This work is a parameter study investigating the impacts of the inflow (directional shear, wind speed) and the geostrophic wind $u_g$ with 10 m s$^{-1}$ and 14 m s$^{-1}$.

[revised manuscript text omitted]

$$v(z,x_{down}) = \left\{ \begin{array}{ll} \text{no veer:} & v_v \cdot (1-\frac{x_{down}}{x_\zeta}) \\ & = \pm\omega r sin(\vartheta)(1-\frac{x_{down}}{x_\zeta}) \\ \text{veer:} & v_f + v_v \cdot (1-\frac{x_{down}}{x_\zeta}) \\ & = u_g \cdot exp(-z\gamma)\cdot tan\left(2\Delta\phi\left(1-\frac{z}{D}\right)\right) \pm\omega r sin(\vartheta)(1-\frac{x_{down}}{x_\zeta}) \end{array} \right\} x_{WT} \leq x_{down} \leq x_\xi \quad (17)$$

In Eq. 17, a linear decrease of $v_v(z,x_{down})$ is assumed for a given downwind distance $x_{down}$ from the rotating system. In this work, we only consider the spanwise flow component, as $w_f$ = -1 and $\beta_w$0 (Eq. 11).

Figure 2 represents the spanwise velocity component $v$ resulting from Eq. 17 at $z$ = 1 and a counterclockwise wake rotation by $\beta_v$125 m and at $z$ = 1 and $\beta_w$75 m at the rotor center in lateral direction. The rotating system has a rotor center $z_h$ = -1, with $\beta_u$100 m and a rotor radius $R$ = 1 in each simulation.A clockwise wake rotation is initiated by a counterclockwise blade rotation, due to conservation of angular momentum (e. g.described in Zhang et al. (2012))and can be linked directly towards the opposite rotation of the rotor. Therefore, our simulations represent 50 m. The vertical positions are centered in the top and the bottom sectors of Fig. 1.

[revised manuscript text omitted]

List of all performed simulations in this study for a clockwise and a counterclockwise rotor rotation. Here, $b$ represents a backing wind and $v$ a veering wind. $th15$ a low stably stratified regime and $th60$ a strongly stably stratified one. $e$ and $l$ correspond to the rotor position, which is affected by wind veer (entire ($e$) rotor or lower ($l$) rotor half). $s$ and $w$ corresponds to a strong or a weak wind veer and u14 to a higher geostrophic wind speed of 14 m s$^{-1}$. $\phi_{wind}$ results from Eq. 10. The simulations $CR$ and $CCR$ correspond to both reference simulations with opposite rotational direction with $CR_v_NH_th30_em_u10$ and $CCR_v_NH_th30_em_u10$. All _ to $CR$ and $CCR$ correspond to the differences between the corresponding simulation and its reference simulations $CR$ and $CCR$.

**5 Rotational Direction Impact on the Wake**

In the following, the impact of the rotational direction of the rotor towards different atmospheric conditions is systematically investigated. Here, we consider the 10-min time and rotor area averaged streamwise velocity $\overline{u_A}$. It is further used to calculate

[Figure]

**Figure 3.** $y$-$z$-cross sections for veering and no veering (NV) inflow simulations at $x = 3$ D for CR (first column), CCR (second column), CR_NV (third column), and CCR_NV (last column). The first row ((a) - (d)) presents the ($v$, $w$) vectors in the $y$-$z$-plane, the second row ((e) - (h)) the spanwise wake velocity $v$, and the third row ((i) - (j)) the vertical wake velocity $w$. The blue circle represents the circumference of the actuator disc. This picture is looking downwind on the wake (Fig. 1, corresponding to the left sector for $y < 0$ D and the right sector for $y > 0$ D.

the power produced by a hypothetical downwind turbine up to 10 D downstream with

$$P = \frac{1}{2}\,\rho_0\,c_p\,\eta_{mech}\,A\,\overline{u_A}$$

where $\rho_0$ is the density of the air, $c_p$

A downwind distance of $x$ = 0.5, $\eta_{mech}$ =3 0.64 and $A$ the area of the rotor (Manwell et al., 2002)D is visualized in Fig. 3 because a significant spanwise vortex impact on the spanwise flow component in the wake can be expected. In the following, $\overline{u_A}$ and P are evaluated and discussed at all downwind positions from 4special emphasis is placed at $x$ D to 10= D, with special emphasis at 7 D, as which is often considered a typical downwind distance for a hypothetical waked wind turbine in numerical

[Figure]

**Figure 4.** Contours of the streamwise velocity $\overline{u_{i,j,k_*}}$ in m s$^{-1}$ at $z = 125$ m in the first two rows, at $z = 100$ m the the third and fourth row, and at $z = 75$ m in the last two rows for the simulations CR, CCR, CR_NV and CCR_NV, each averaged over 30 min.The black contours represent the velocity deficit $VD_{i,j,k_*}$ at the same vertical location.

simulation studies (e.g. Gaumond et al. (2014); Abkar et al. (2016)).

$$VD_{i,j,k} = \frac{\overline{u_{i_1,j,k}} - \overline{u_{i,j,k}}}{\overline{u_{i_1,j,k}}},$$

**Table 1.** List of all performed simulations in this study for a clockwise (leftmost column) and a counterclockwise (rightmost column) rotor rotation. The parameters $u_g$, $ds$, and $\Omega$ refer to both rotational directions, whereas the only difference e.g. between CR and CCR in the first line is the rotational direction. Further, _NV represents no wind veer, _b a backing wind, _ds refers to varying the directional shear, _u refers to varying the geostrophic wind, and _$\Omega$ refers to varying the rotational frequency in the corresponding simulations.

| | SIMULATIONS WITH DIFFERENT ROTATIONAL DIRECTIONS OF THE ROTOR | | |
|---|---|---|---|
| CLOCKWISE | $\phi_{wind}$(150 m)$\phi_{wind}$(50 m)$\frac{\partial\Theta}{\partial z}$ u$_g$ | f $ds$ | $\Omega$ |
| CR | -4° 4° 3 K / 200 m 10 m s$^{-1}$ | 10 m s$^{-1}$ 0.08° m$^{-1}$ | > 0 0.12° s$^{-1}$ |
| CR_bNV | 4° -4° 3 K / 200 m 10 m s$^{-1}$ | > 0° m$^{-1}$ | CCR_b CR_v_SH 4° -4° 3 K / 200 m10 m0.12° s |
| CR_SHb | -4° 4° 10 m s$^{-1}$ | 3 K / 200 m 0.08° m$^{-1}$ | 10 m0.12° s$^{-1}$ |
| CR_th15ds4 | -4° 4° 1.5 K / 200 m 10 m s$^{-1}$ | > 0 CCR_th15CR_th60 -4° 0.04° m$^{-1}$ | 4° 0.12° s$^{-1}$ |
| CR_ds12 | 10 m s$^{-1}$ | > 0 0.12° m$^{-1}$ | CCR_th60CR_es 0.12° s$^{-1}$ |
| CR_ds16 | 8° 10 m s$^{-1}$ | 3 K / 200 m0.16° m$^{-1}$ | 10 m0.12° s$^{-1}$ |
| CR_ewds20 | -2° 2° 10 m s$^{-1}$ | 3 K / 200 m0.20° m$^{-1}$ | 10 m0.12° s$^{-1}$ |
| CR_lsu6 | -8° 0° 6 m s$^{-1}$ | 3 K / 200 m0.08° m$^{-1}$ | 10 m0.12° s$^{-1}$ |
| CR_lmu14 | -4° 0° 14 m s$^{-1}$ | 3 K / 200 m0.08° m$^{-1}$ | 10 m0.12° s$^{-1}$ |
| CR_u14Ωl | -4° 4° 3 K / 200 m10 m s$^{-1}$ | 14 m s$^{-1}$ 0.08° m$^{-1}$ | > 0 0.058° s$^{-1}$ |
| CR_es_u14Ωh | -8° 10 m s$^{-1}$ | 8° 0.08° m$^{-1}$ | 3 K / 200 m0.175° s$^{-1}$ |
| CR_Ωvh | > 0 10 m s$^{-1}$ | 0.08° m$^{-1}$ | 0.23° s$^{-1}$ |

calculated at the discrete grid points $x_i$, $y_j$, and $z_k$ with $i_1$ corresponding to the first upstream grid pointAt $x = 7$ D, the vortex impact is much smaller compared to $x = 3$ D (Fig. 3), resulting in an increase of the impact of the atmospheric flow.

**4.1 Veering Wind vs. Backing Wind on both Hemispheres**

The comparison of simulations $CR$ and $CR_b$ reveals the difference in $\overline{u_A}$ and $P$ between a veering and a backing wind on the in case of a clockwise rotating rotor . As the rotational direction has a significant impact on the spanwise flow component at $x = 3$ D (Fig. 3), an impact on the streamwise flow component is also expected. The numerical results for the streamwise velocity component are presented for veering (CR, CCR) and non-veering (CR_NV, CCR_NV) inflow by $x$ - $y$ cross sections of the streamwise velocity in the top half of the rotor disc at $z = 125$ m (Fig. $7a$ shows larger $\overline{u_A}$-values if abacking wind ($CR_b$)interacts with a clockwise rotating rotor in comparison to a veering wind ($CR$). In addition, the difference of $\overline{u_A}$ between a backing and a veering wind increases downwind up to $\Delta\overline{u_A}$4(a) ≈- (d)), at hub height at $z$ 0.5 m s$^{-1}$ at 10= D. Considering a counterclockwise rotating rotor in $CCR$ and $CCR_b$, $\Delta\overline{u_A}$ is the same for $CR$ and $CCR_b$ and likewise for

[Figure]

**Figure 5.** Contours of the streamwise velocity $\overline{u_{i,j,k_*}}$ in m s$^{-1}$ at a downward position of $x = 3$ D behind the rotor of for CR in (a), CCR in (b), CR_NV in (c), and CCR_NV in (d). The blue circle represents the circumference of the actuator disc.

$CCR$ and $CR_b$. This results in larger $\overline{u_A}$-values and $P$-values in $CCR$ 100 m (Fig. 4(e) - (h)), and in the bottom half of the rotor disc at $z = 75$ m (Fig. 4(i) - (l)).

The effect of wind veer on the streamwise velocity component of clockwise rotating wind turbines is investigated by comparing CR to CR_NV, in Fig. 4(a) vs. (c) at $z = 125$ m, (e) vs. (g) at $z = 100$ m, and (i) vs. (k) at $z = 75$ m. Inflow veer causes a more rapid wake recovery at all heights, based on comparison of the velocity deficit contours. Because enhanced $\frac{\partial v_f}{\partial z} \neq 0$ in the case of veering wind, it provides a source of resolved turbulence resulting in higher entrainment in compari-

[Figure]

**Figure 6.** Vertical (first column) and horizontal profiles at $z = 75$ m (second column), $z = 100$ m (third column), and $z = 125$ m (fourth column) of the 30 min averaged streamwise velocity at $x = 7$ D downwind of the actuator for CR and CCR in (a) - (d), CR_NV and CCR_NV in (e) - (h), CR_b and CCR_b in (i) - (l), CR_u6 and CCR_u6 in (m) - (p), and CR_u14 and CCR_u14 in (q) - (t).

son to the no-veer case. Further, inflow wind veer causes wake deflection in both the top half (Fig. 4(a) vs. (c)) and the bottom half (Fig. 4(i) vs. (k)) of the rotor disc. The wake in the veered simulation CR is deflected

[Figure]

**Figure 7.** The rotor and time averaged streamwise velocity $\overline{u_A}$ presented for a downwind region of $[4\,D; 10\,D]$ with special emphasis at $x = 7$ D for the simulations CR_NV, CCR_NV, CR, CCR, CR_b and CCR_b in (a), for different geostrophic wind values in (b), for different directional shears in (c), and for different rotational frequencies in (d).

[Figure]

**Figure 8.** Schematic illustration of the rotational direction of the wake for the cases: Clockwise blade rotation CR with veering wind in NH (corresponding to backing wind in SH) in (a), counterclockwise blade rotation CCR with veering wind in NH in (b), counterclockwise blade rotation with backing wind CCR_b in NH (corresponding to veering wind in SH) in (c), and clockwise blade rotation with backing wind CR_b in NH in (d).

towards the right ($y$ D downwind turbine, this leads to an 11.5% increase in power in case of a veering wind interacting with counterclockwise rotating blades or if abacking wind interacts with clockwise rotating blades.

The rotor averaged streamwise velocity $\overline{u_A}$ and the power $P$ of a hypothetical downwind turbine are presented for a downstream region of $[4\,D; 10\,D]$ for a veering and a backing wind in the in $a$, for a veering and a backing wind in the in $b$, and for all eight simulations together in $c$.

The same investigation is shown in Fig. 7$b$ for the . Here, we also consider a veering and abacking wind with both rotational directions of the rotor, resulting in the simulations $CR_v_SH$, $CCR_v_SH$, $CR_SH$, and $CCR_SH$.The downstream behaviour and likewise $\Delta\overline{u_A}$ are similar but opposite to the results in the .The power output of a hypothetical downwind turbine at 7$\gtrsim$ D would also be larger by 11.5%, however, on the , in case of a veering wind and clockwise rotating blades or a backing

wind interacting with counterclockwise rotating blades.The minor difference near 100 D downwind between $CR_v_SH$ and $CCR_SH$ and likewise between $CCR_v_SH$ and $CR_SH$ in D) (left ($y<0$ D)) in the upper (lower) rotor part (Fig. 7$b$ results from the applied parametrization of Englberger and Dörnbrack (2018b), as the inflow wind field was extracted from a diurnal cycle LES on the (Englberger and Dörnbrack, 2018a).This assumption is supported by the following aspects: the difference is not prevalent in the northern hemispheric simulations in Fig. 7$a$, and the difference emerges far downstream, starting at $x$4(a) (Fig. 4(i))). In the non-veered simulation CR_NV, the wake is only slightly deflected towards the left in the top-tip sector (Fig. 4(c)) and towards the right in the bottom-tip sector (Fig. 4(k)). This effect is caused by the rotation of the rotor, which transports higher momentum air counterclockwise, resulting in a wake deflection to the left at $z$ >= 8125 m (Fig. 4(c)). Consequently, the opposite situation prevails at $z$ D, where the = 75 m (Fig. 4(k)). As the inflow veer contribution to wake deflection is much larger compared to the effect of a clockwise rotating rotor, the wake deflection changes from left in CR_NV (Fig. 4(c)) to the right in CR (Fig. 4(a)) in the upper rotor half and vice versa in the lower rotor half.

As a next step, the rotational direction impact in the non-veered simulations CR_NV and CCR_NV is investigated (Figs. 4(c) vs. (d), (g) vs. (h), and (k) vs. (l)). The impact of the disc rotational direction on the wake structure is rather small in comparison to the ambient flow field impact and it also increases approaching 10 D.

Schematic illustration of the rotational direction of the wake for the cases: Clockwise blade rotation with veering wind in NH ($CR$) and with backing wind in SH ($CR_SH$) in (a), counterclockwise blade rotation with veering wind in NH ($CCR$) and backing wind in SH ($CCR_SH$) in (b), counterclockwise blade rotation with backing wind in NH ($CCR_b$) and veering wind in SH ($CCR_v_SH$) in (c), and clockwise blade rotation with backing wind in NH ($CR_b$) and veering wind in SH ($CR_v_SH$) in (d).

In Fig. 7$c$, all eight simulations are shown. Here, the results of simulations $CCR$, $CR_b$, $CR_v_SH$, and $CCR_SH$ overlap and, likewise, the ones for $CR$, $CCR_b$, $CCR_v_SH$, and $CR_SH$. The resulting flow fields of the wakes are schematically shown in is limited to the wake deflection differences at the upper (Fig. 4(c), (d)) and the lower (Fig. 8. The combinations of $\frac{\partial \phi_{wind}}{\partial z}$ 4(k), (l)) rotor height, which are nearly axis-symmetric to $y$ <= 0and aclockwise blade rotation D and result from the rotational direction of the rotor. These differences in the non-veered simulations agree with results of Vermeer et al. (2003), Shen et al. (2007), Sanderse (2009), Kumar et al. (2013), Hu et al. (2013), Yuan et al. (2014), Mühle et al. (2017), and Englberger et al. (20
.

The rotational direction impact on the wake structure under veering inflow is investigated by a comparison of CCR to CR (Fig. 4(b) vs. (a), (f) vs. (e), and (j) vs. (i)). In CCR, the wake recovers more rapidly (Fig. 8$a$) or $\frac{\partial \phi_{wind}}{\partial z} > 0$ and a counterclockwise blade rotation (Fig. 8$c$) result in contrasting rotational directions of the near and far wake , referred to hereafter as 'contrasting wake cases'. The combinations of $\frac{\partial \phi_{wind}}{\partial z} < 0$ and a counterclockwise blade rotation 4(f) vs. (e)) and

the wake deflection angle is larger (Figs. 4(b) vs. (a) and (j) vs. (i)) in comparison to CR. Further, the wake width is larger in the spanwise direction in CCR in comparison to CR (Fig. 8*b*) or $\frac{\partial \phi_{wind}}{\partial z}$ 4(b) vs. (a), (f) vs. (e), and (j) vs. (i)).

The differences in the spanwise wake width and the wake deflection angle are investigated in more detail with the $y >$ - 0 and aclockwise blade rotation ($z$ crosssections at $x = 7$ D in Fig. 8*d*), however, result in a rotational direction of the wake which persists in the whole wake, referred to hereafter as 'consistent wake cases'.This contrasting and consistent behaviour of the rotational direction in the wake is valid on both hemispheres.

The 10-min time averaged streamwise velocity, representing the simulation from 10 min to 20 min, is plotted at hub height in Fig. **??** for all eight cases together with the velocity deficit (Eq. 14)as contour.The structures of the four contrasting wake cases (left row)resemble each other with narrower wakes. Similarly, the four consistent wake cases (right row)resemble each other with wider wakes. The entrainment of ambient air in the consistent wake cases is slightly less rapid in the near wake in comparison to the contrasting wake cases, whereas it is substantially enhanced in the far wake. This results in the higher $\overline{u_A}$-value in the consistent wake cases and an increase of $\Delta\overline{u_A}$ approaching downstream with rather similar values in the near wake for 5 for veering inflow (CR in (a), CCR in (b)) and no wind veer (CR_NV in (c), CCR_NV in (d)) with both rotational directions of the actuator. In the case of no veering inflow, the simulated wake at $x = 7$ D retains the shape of the rotor (Fig. 5(c)). In the case of a veering inflow, however, the wake in the lower rotor half is shifted to the left and in the upper rotor half to the right (Fig. 5(a)). The striking difference between veering and non-veering inflow simulations in combination with a clockwise rotating actuator corresponds to the inflow profile (Eqs. 6, 8), where a veering inflow is characterized by a wind component from right to left for $z <$ 4100 m and from left to right for $z$ D $>$ 100 m, whereas the spanwise inflow velocity is zero in the case of no veer in all rotor heights. The skewed wake structure under veering inflow resembles those of the simulations of Abkar and Porté-Agel (2016), Vollmer et al. (2017), Bromm et al. (2017), Churchfield and Sirnivas (2018), and Englberger and Dörnbrack (2018a).

Further, we compare the differences between a clockwise and a counterclockwise rotating actuator for non-veering and veering inflow. In the case of no wind veer, the simulated wake structures of CCR_NV (Fig. 5(d)) and CR_NV (Fig. 5(c)) show no striking difference. In the case of veering inflow, however, the skewed wake structure differs in CR and CCR (Fig. 5(a), (b)). Whereas the wake is elliptical in CR, this shape is stretched in the rotor region in CCR. This difference in shape explains the difference in the spanwise wake width at hub height (Fig. 4(f)) and also in the lower (Fig. 7) 4(j)) and the upper (Fig. 4(b)) rotor part. The wake structure dependence on the rotational direction of the rotor concurs with the results in outside the rotor region also differs between Fig. 4 of Englberger et al. (2019).

Coloured contours of the streamwise velocity $\overline{u_{i,j,k_h}}$ in m s$^{-1}$ at hub height $k_h$, averaged over the last 10 min, for $CR$ in (a), $CCR$ in (b), $CCR_b$ in (c), $CR_b$ in (d), $CCR_v_SH$ in (e), $CR_v_SH$ in (f), $CR_SH$ in (g), and $CCR_SH$ in (h). The black contours represent the velocity deficit $VD_{i,j,k_h}$ at the same vertical location.

5(a) and (b). Due to the elongation of the elliptical structure in CCR in the rotor region (Fig. 4(b)), and approximately the same vertical wake extension in CCR and CR, the wake deflection angle increases in the case of CCR (Fig. 5(b) vs. (a)), as shown in the lower rotor half in Fig. 4(j) vs. (i) and also in the upper rotor half in Fig. 4(b) vs. (a).

The evolving different wake structures result in a larger power output of the consistent wake cases in comparison to the contrasting wake cases of a downwind turbine of roughly 11% at A quantitative description of the streamwise velocity differences is presented in Fig. 6 at the downwind position $x = 7$ D approaching even to 19% at 10D. Figure 6 represents the vertical profiles at $y$ D.Considering the much higher frequency of occurrence of a veering wind in comparison to a backing wind (≈3.8 times more frequent according to two years of meteorological tower measurements in Lubbock (Texas)(Walter et al., 2009) ), a counterclockwise rotating rotor in the (and a clockwise rotating rotor in the ) would increase the power production for awaked turbine downwind.

**4.1 Strength of Stratification**

The impact of the stable stratification is tested for three different regimes, aweakly stably stratified atmosphere in $CR_th15$, a moderate stably stratified atmosphere in $CR$, and a strongly stably stratified atmosphere in $CR_th60$. The tested lapse rates are representative compared $= 0$ D in (a), and spanwise profiles of $u$ at $z = 75$ m in (b), at $z = 100$ m in (c), and at $z = 125$ m in (d) for both rotational directions CR and CCR. The heights correspond to Fig. 2 in Walter et al. (2009). The impact on $\overline{u_A}$ is presented in 4. Figure 6(e) - (h) represent the non-veering inflow simulations CR_NV and CCR_NV. Whereas the vertical and spanwise profiles of CR and CCR in the case of no inflow veer (_NV) are almost overlapping (Fig. **??**$a$. In case of a common clockwise rotating rotor, the wake lasts longer in stronger stratification 6(e) - (h)), there is a difference in the case of veering inflow (Fig. 6(a) - (d)). Firstly, the streamwise wake elongation difference of Figs. 4(f) vs. (e) is represented by larger $u$-values in the lower and the upper rotor half in the case of CCR in Fig. **??**. The wake recovers faster in $CR_th15$ compared to $CR$ and further $CR$ recovers faster compared to $CR_th60$. This differences in $\overline{u_A}$ translates in a 19% larger power output of a hypothetical downwind turbine in an during the evening transition ($CR_th15$)in comparison to the power output at night where the surface fluxes are at its minimum ($CR_th60$). Following the increase of the recovery rate from $CR_th60$ to $CR$ to $CR_th15$, it would result in an increase of 6(b) and (d). The larger wake deflection angle in CCR in comparison to CR (Fig. 4 (b) vs. (a) and (j) vs. (i)) is represented by a larger spanwise distance of the minimum of $u$ from $y = 0$ D in case of CCR in the power output for decreasing the strength of stratification.

The rotor averaged streamwise velocity $\overline{u_A}$ and the power $P$ of a hypothetical downwind turbine are presented for a downstream region of $[4\,D; 10\,D]$ for different thermal stratifications in $a$, for different strength of wind veer and rotor areas affected by the veering wind in $b$, and for different wind speeds in $c$.

Considering the same stratification with a counterclockwise rotating rotor in $CCR_th15$, $CCR$, and $CCR_th60$ in lower (Fig. 6(b)) and the upper (Fig. 6(d)) rotor half. This spanwise difference of $u_{min}$ is accompanied by larger $u$-values in the case of CCR for $y <$ -1/2 D in the lower rotor part (Fig. 6(b)) and for $y <$ 1/2 D in the upper rotor part (Fig. 6(d)). Secondly, the difference in the spanwise wake width is represented in all three heights by a larger $\Delta L_y$ with smaller $u$-values in CCR in the outermost region of the left and the right sectors in comparison to CR (Fig. **??**$a$, the values of downwind wind speed $\overline{u_A}$ are rather similar and nearly independent of the stratification. Only in the strongly stratified regime $CCR_th60$, the 6(b) - (d)).

As final step, the difference in the wake is summarized by the 30-min time and rotor area averaged streamwise velocity $\overline{u_A}$. Figure 7(a) represents the difference between clockwise and counterclockwise rotating rotors for a veering inflow and in the case of no wind veer from $x = 4$ D to 10 D. At $x = 7$ D, $\overline{u_A}$ -value slightly increases is 0.24 m s$^{-1}$ larger in the counterclockwise rotating rotor simulation CCR in comparison to the weakly (4%)and moderate (3%)regimes, resulting in a maximum power output of a hypothetical downwind turbine at night where the surface fluxes approaching its minimum. However, the impact of stratification is roughly five times smaller in comparison to the one for clockwise rotating wind turbines.

This wake behaviour results in a larger potential power output of a downwind turbine in case of a counterclockwise rotating rotor of 4% in the weakly stably stratified case, of 11.5% in the moderate stably stratified case, and of 23% in the strongly stably stratified situation at 7CR, whereas there is no difference between CCR_NV and CR_NV. According to Fig. 6(a) D. A counterclockwise blade rotation will not only enhance the power output, it will further increase the accumulated power output during the rather long nights with approximately constant surface fluxes (9 h (Walter et al., 2009, Fig. 2), 11 h (Blay-Carreras et al., 2014; Abkar et al., 2016; Englberger and Dörnbrack, 2018a, Fig. 1)). In addition, counterclockwise blade rotation would also increase the power output during the morning boundary layer regime. This regime is strongly affected by the previous nocturnal stability with an even smaller entrainment rate before the surface fluxes become positive due to the incoming solar radiation (Englberger and Dörnbrack, 2018a, Fig. 4). (d), these larger $\overline{u_A}$-values in the case of CCR result from larger $u$-values in the upper and lower sector related to the larger wake deflection angle in the case of CCR, which compensates for the larger $u$-values in the outer region of the left and right sectors resulting from a larger spanwise wake width in the case of CCR.

The previous investigations show a striking dependence of the rotational direction of the rotor on the wake under veering inflow, which is qualitatively well explained by the analysis. A schematic illustration of the deceleration or even reversion of

the spanwise flow if a clockwise rotating rotor CR interacts with a veering wind is presented in Fig. 8(a). The amplification of the spanwise flow in the case of a counterclockwise rotating rotor CCR interacting with veering inflow is presented in Fig. 8(b).

**4.1 Veering Wind vs. Backing Wind**

The contrasting power production between the clockwise and the counterclockwise rotating simulations can be explained by means of According to the analytical results (Fig. ??. The wake structure in the clockwise rotating blade simulations $CR_th15$ in $a$, $CR$ in $c$, and $CR_th60$ in $e$, behaves as known from previous studies: a less rapid wake recovery and an elongated wake for a stronger stably stratified regime (Abkar and Porté-Agel, 2014; Abkar et al., 2016; Vollmer et al., 2016; Englberger and Dörnbrack, 2018a .In contrast, the wake structures are rather similar in $CCR_th15$ in $b$ and in $CCR$ 2(e) vs. (h)), the spanwise component $v$ in the wake is expected to be comparable for a clockwise rotating rotor in veering inflow and a counterclockwise rotating rotor in backing inflow, as well as a clockwise rotating rotor in backing inflow and a counterclockwise rotating rotor in veering inflow. The wake characteristics resulting from a backing wind with both rotational directions are investigated in the simulations CR_b and CCR_b and compared to the veering wind cases CR and CCR in Fig. 9. The parameters applied in the corresponding simulations are listed in Table 1.

The behaviour in the upper and the lower rotor part in Fig. 9 can directly be compared after mirroring at $y = 0$ D, an effect resulting from the opposite sign of the directional shear and $\Delta\phi$ in Eqs. 9 and 10. A strong similarity is prevalent in the streamwise velocity component at hub height (Fig. 9(f), (g) and (e), (h)), in $d$.Only the wake width of $CCR_th60$ in $f$ differs slightly from $CCR_th15$ and$CCR$. A significant difference in the wake elongation,as in the $CR$ simulations,however, can not be detected in the $CCR$ simulations.This significant difference in the entrainment process results from the different behaviour in the wake rotation approaching downstream between the contrasting wake cases $CR$ and the consistent wake cases $CCR$ lower rotor half (Fig. 9(i), (l) and (j), (k)) as well as in the upper rotor half (Fig. 8) and is responsible for an increase of a downwind turbines power output up to 23% for counterclockwise rotating blades instead of clockwise ones.

**4.2 Strength of Veering Wind**

The impact of the strength of wind veer over the rotor is investigated for $\Delta\phi$-values of 2°9(a), (d) and (b), (c)). The more rapid wake recovery and the larger spanwise wake width for CCR and CR_b in comparison to CR and CCR_b are present in all rotor heights. The larger wake deflection angle in the upper and the lower rotor half in CCR (Fig. 9(b), (j)) is also comparable to CR_b (Fig. 9(c), (k)), whereas the smaller wake deflection angle in CR (Fig. 9(a),4°, and 8°,corresponding to a weak ($w$),moderate ($m$), or strong ($s$)change in $CR_es$,$CR$,and $CR_ew$ and plotted (i)) is comparable to CCR_b (Fig. 9(d), (l)).

[revised manuscript text omitted]

The magnitude of directional shear affects the wake elongation , however, the difference between strong, moderate, and weak wind veer is much smaller in comparison to the corresponding $CR$ simulations . In detail, the in dependence of the rotor direction, but, not to the same extent in clockwise rotating simulations in comparison to counterclockwise rotating ones. The wake elongation in $CR_cw$ CR_ds4 is much longer in comparison to $CCR_cw$CCR_ds4 (Figs. 10 - 12(a) vs. (b)). It is still larger in $CR$ CR in comparison to $CCR$.These differences between the $CR$ and the $CCR$ simulations result in the larger power output of 13% in the strong veer $CCR$ cases and of 11% in the moderate veer $CCR$ casesCCR (Figs. 10 - 12(c) vs. (d)). A further increase of the directional shear finally results in a similar wake recovery of CR_ds12 and CCR_ds12 (Figs. 10 - 12(e) vs. (f)) and a slightly more rapid wake recovery of CR_ds16 in comparison to CCR_ds16 (Figs. 10 - 12(g) vs. (h)). Comparing the very strong directional shear cases CR_ds20 and CCR_ds20 (Figs. 10 - 12(i) vs. (j)), the wake recovery is significantly faster in CR_ds20. Further, the difference between $CR_cw$ and $CR$ CR_ds4 and CR is larger in comparison to $CCR_cw$

and $CCR$.Both wake elongation trends continue for an increasing amount of veer. It finally results in a faster wake recovery of $CR_es$ in comparison to $CCR_es$, which is responsible for the slightly larger power output of $CR_es$ in comparison to $CCR_es$. CCR_ds4 and CCR (Figs. 10 - 12(a), (c) and (b), (d)). This trend continues for increasing directional shear.

**4.4 Type of Veering Wind**

[revised manuscript text omitted]

The same investigation is presented clockwise rotating rotor (Fig. 18(g), (e), (i)). This behaviour corresponds to the near wake differences in Fig. ?? for $CR_es$ vs. $CCR_es$ in 'es',$CR_ew$ vs. $CCR_ew$ in 'ew',$CR_ls$ vs. $CCR_ls$ in 'ls', and $CR_lm$ vs. $CCR_lm$ in 'lm' and $CR_u14$ vs. $CCR_u14$ and $CR_es_u14$ vs. $CCR_es_u14$.

In 2(g), (e), (i). For large enough values of the rotational frequency, the spanwise wake component reverses sign in the simulation CR_Ωh directly behind the rotor (Fig. ??, squared markers represent cases of strong entrainment processes in the wake of the upwind turbine e.g. 'th15' weak stably stratified regime, 'es' strong wind veer over the whole rotor, 'ls' strong wind veerlimited to the lower rotor part. Circles represent moderate forcings and entrainment processes like 'ref'($th30$ moderate stably stratified regime and $cm$ moderate windveer over the entire rotor) and triangles represent weak entrainment processesin 18(i)).

The comparison of the wake e.g. 'th60' strong stably stratified regime, 'ew' weak wind veer over the whole rotor. The differences in the entrainment rate can be seen in Figs. ?? and ??. From simulation results with the analysis predictions can be summarized as follows:

[revised manuscript text omitted]

Sanchez Gomez, M. and Lundquist, J.: The Effects of Wind Veer During the Morning and Evening Transitions, in: Journal of Physics: Conference Series, vol. 1452, p. 012075, 2020a.

Sanchez Gomez, M. and Lundquist, J. K.: The effect of wind direction shear on turbine performance in a wind farm in central Iowa, Wind Energy Science (Online), 5, 2020b.

Sanderse, B.: Aerodynamics of wind turbine wakes, Energy Research Center of the Netherlands (ECN), ECN-E–09-016, Petten, The Netherlands, Tech. Rep, 5, 153, 2009.

Schmidt, H. and Schumann, U.: Coherent structure of the convective boundary layer derived from large-eddy simulations, J Fluid Mech, 200, 511–562, https://doi.org/10.1017/S0022112089000753, 1989.

Shapiro, A. and Fedorovich, E.: Analytical description of a nocturnal low-level jet, Quarterly Journal of the Royal Meteorological Society, 136, 1255–1262, 2010.

Shen, W. Z., Zakkam, V. A. K., Sørensen, J. N., and Appa, K.: Analysis of counter-rotating wind turbines, in: Journal of Physics: Conference Series, vol. 75, p. 012003, IOP Publishing, https://doi.org/10.1088/1742-6596/75/1/012003, 2007.

Smolarkiewicz, P. K. and Margolin, L. G.: MPDATA: A Finite-Difference Solver for Geophysical Flows, J Comput Phys, 140, 459–480, https://doi.org/10.1006/jcph.1998.5901, 1998.

Smolarkiewicz, P. K., Sharman, R., Weil, J., Perry, S. G., Heist, D., and Bowker, G.: Building resolving large-eddy simulations and comparison with wind tunnel experiments, J Comput Phys, 227, 633–653, https://doi.org/10.1016/j.jcp.2007.08.005, 2007.

Stull, R. B.: An Introduction of Boundary Layer Meteorology, Dordrecht, Kluwer Academic, 1988.

Tomaszewski, J. M., Lundquist, J. K., Churchfield, M. J., and Moriarty, P. J.: Do wind turbines pose roll hazards to light aircraft?, Wind Energy Science (Online), 3, 2018.

Vasel-Be-Hagh, A. and Archer, C. L.: Wind farms with counter-rotating wind turbines, Sustainable Energy Technologies and Assessments, 24, 19–30, 2017.

Vermeer, L., Sørensen, J. N., and Crespo, A.: Wind turbine wake aerodynamics, Progress in aerospace sciences, 39, 467–510, https://doi.org/10.1016/S0376-0421(03)00078-2, 2003.

Vollmer, L., Steinfeld, G., Heinemann, D., and Kühn, M.: Estimating the wake deflection downstream of a wind turbine in different atmospheric stabilities: An LES study, Wind Energy Science, 1, 129–141, 2016.

Vollmer, L., Lee, J. C., Steinfeld, G., and Lundquist, J.: A wind turbine wake in changing atmospheric conditions: LES and lidar measurements, in: Journal of Physics: Conference Series, vol. 854, p. 012050, IOP Publishing, https://doi.org/10.1088/1742-6596/75/1/012003, 2017.

Walter, K., Weiss, C. C., Swift, A. H., Chapman, J., and Kelley, N. D.: Speed and direction shear in the stable nocturnal boundary layer, Journal of Solar Energy Engineering, 131, 011 013, https://doi.org/10.1115/1.3035818, 2009.

Yuan, W., Tian, W., Ozbay, A., and Hu, H.: An experimental study on the effects of relative rotation direction on the wake interferences among tandem wind turbines, Science China Physics, Mechanics & Astronomy, 57, 935–949, 2014.

Zhang, W., Markfort, C. D., and Porté-Agel, F.: Near-wake flow structure downwind of a wind turbine in a turbulent boundary layer, Exp Fluids, 52, 1219–1235, https://doi.org/10.1007/s00348-011-1250-8, 2012.

[Figure]

Coloured contours

**Figure 9.** Contours of the streamwise velocity $\overline{u_{i,j,k_h}}$ $\overline{u_{i,j,k_*}}$ in m s$^{-1}$ at hub height $k_h$, averaged over the last 10 min, for $CR_th15$ in (a), $CCR_th15$ $z = 125$ m in (b)the first two rows, $CR$ in (c)at $z = 100$ m the the third and fourth row, $CCR$ and at $z = 75$ m in (d)the last two rows for the simulations CR, $CR_th60$ in (e)CCR, CR_b and $CCR_th60$ in (f)CCR_b, each averaged over 30 min. The black contours represent the velocity deficit $VD_{i,j,k_h}$ $VD_{i,j,k_*}$ at the same vertical location.

[Figure]

**Figure 10.** Contours of the streamwise velocity $\overline{u_{i,j,k_*}}$ in m s$^{-1}$ for different directional shears at $z = 100$ m for CR_ds4 in (a), CCR_ds4 in (b), CR in (c), CCR in (d), CR_ds12 in (e), CCR_ds12 in (f), CR_ds16 in (g), CCR_ds16 in (h), CR_ds20 in (i), and CCR_ds20 in (j), each averaged over 30 min. The black contours represent the velocity deficit $VD_{i,j,k_*}$ at the same vertical location.

[Figure]

**Figure 11.** Contours of the streamwise velocity $\overline{u_{i,j,k_*}}$ in m s$^{-1}$ for different directional shears at $z = 125$ m for the same simulations as in Fig. 10. The black contours represent the velocity deficit $VD_{i,j,k_*}$ at the same vertical location.

[Figure]

**Figure 12.** Contours of the streamwise velocity $\overline{u_{i,j,k_*}}$ in m s$^{-1}$ for different directional shears at $z = 75$ m for the same simulations as in Fig. 10. The black contours represent the velocity deficit $VD_{i,j,k_*}$ at the same vertical location.

[Figure]

**Figure 13.** Vertical (first column) and horizontal profiles at $z = 75$ m (second column), $z = 100$ m (third column), and $z = 125$ m (fourth column) of the 30 min averaged streamwise velocity at $x = 7\,\mathrm{D}$ downwind of the actuator for a directional shear of $0.04^\circ\ \mathrm{m}^{-1}$ in (a) - (d), $0.08^\circ\ \mathrm{m}^{-1}$ in (e) - (h), $0.12^\circ\ \mathrm{m}^{-1}$ in (i) - (l), $0.16^\circ\ \mathrm{m}^{-1}$ in (m) - (p), and $0.20^\circ\ \mathrm{m}^{-1}$ in (q) - (t).

[Figure]

**Figure 14.** Contours of the streamwise velocity $\overline{u_{i,j,k_*}}$ in m s$^{-1}$ for different rotational frequencies at $z = 100$ m for CR_Ωl in (a), CCR_Ωl in (b), CR in (c), CCR in (d), CR_Ωh in (e), CCR_Ωh in (f), CR_Ωvh in (g), and CCR_Ωvh in (h), each averaged over 30 min. The black contours represent the velocity deficit $VD_{i,j,k_*}$ at the same vertical location.

[Figure]

**Figure 15.** Contours of the streamwise velocity $\overline{u_{i,j,k_*}}$ in m s$^{-1}$ for different rotational frequencies at $z = 125$ m for the same simulations as in Fig. 14. The black contours represent the velocity deficit $VD_{i,j,k_*}$ at the same vertical location.

[Figure]

**Figure 16.** Contours of the streamwise velocity $\overline{u_{i,j,k_*}}$ in m s$^{-1}$ for different rotational frequencies at $z = 75$ m for the same simulations as in Fig. 14. The black contours represent the velocity deficit $VD_{i,j,k_*}$ at the same vertical location.

[Figure]

**Figure 17.** Vertical (first column) and horizontal profiles at $z = 75$ m (second column), $z = 100$ m (third column), and $z = 125$ m (fourth column) of the 30 min averaged streamwise velocity at $x = 7$ D downwind of the actuator for a rotational frequency of $\Omega = 0.058°\ s^{-1}$ in (a) - (d), $\Omega = 0.12°\ s^{-1}$ in (e) - (g), $\Omega = 0.175°\ s^{-1}$ in (i) - (l), and $\Omega = 0.23°\ s^{-1}$ in (m) - (o).

[Figure]

**Figure 18.** Sector averages of $\overline{v}$ representing the top and bottom 90°-sectors for 0 m < r ≤ 50 m for clockwise and counterclockwise rotating actuators in the corresponding simulation of no veer in (b), a veering wind in (e) and a backing wind in (h). In the case of a veering wind in (e), moderate parameters of $u_g = 10$ m s$^{-1}$, $ds = 0.08°$ m$^{-1}$, and $\Omega = 0.12°$ s$^{-1}$ are applied. In the left and right column, only one parameter is changes compared to the veering wind situation in (e). Applying low parameters $u_g = 6$ m s$^{-1}$ in (a), $ds = 0.04°$ m$^{-1}$ in (d), and $\Omega = 0.058°$ s$^{-1}$ in (g), and applying high parameters $u_g = 14$ m s$^{-1}$ in (c), $ds = 0.12°$ m$^{-1}$ in (f), and $\Omega = 0.175°$ s$^{-1}$ in (i). The plot is directly comparable to Fig. 2 considering the configurations.

[Figure]

**Figure 19.** Sector averages of $\overline{u}$ representing the top and bottom 90°-sectors for 0 m < r ≤ 50 m for clockwise and counterclockwise rotating actuators for the same simulations as in Fig. 18.

[Figure]

**Figure 20.** Schematic illustration of the difference in the spanwise wake width ($L_y$) and the wake deflection angle ($\epsilon$) between clockwise CR and counterclockwise CCR rotating actuators. The parameter impact on the these wake differences is represented by $\Delta L_y$ and $\Delta\epsilon$.

[Figure]

**Figure A1.** Contours of the streamwise velocity $\overline{u_{i,j,k_*}}$ in m s$^{-1}$ at $z = 125$ m in the first two rows, at $z = 100$ m the the third and fourth row, and at $z = 75$ m in the last two rows for the simulations CR, CCR, CR_$\alpha$ and CCR_$\alpha$, each averaged over 30 min.The black contours represent the velocity deficit $VD_{i,j,k_*}$ at the same vertical location.

---

## Referee Report (RR1)

**Review of "Changing the rotational direction of a wind turbine under veering inflow: A parameter study"**

**General comments**

The authors have improved the manuscript significantly compared to the previous submission. The slightly narrowed focus of the paper, which is investigated in more detailed, is still a very compelling research question. The additional figures in the result section helped a lot to better understand the effects on the wake structure, which was my main issue with the previous manuscript. The new analysis with the simple model and its comparison to the simulations is insightful for the interactions between the wind veer and the rotational direction. The revised and more conservative conclusions are better supported by the results compared to the previous iteration. I am not qualified to evaluate, if the technical aspects of the LES highlighted by the first and third reviewer have been sufficiently addressed. This time, I have only some minor comments detailed below.

**Minor comments**

In Eq. (15) and (16) the $\vartheta$, and in Eq. (17) $x_{WT}$ and and $x_\xi$ are not introduced.

Page 8, lines 2-3: An addition to this sentence, that the simple model demonstrates the principle interactions between an idealized wake represented by a vortex and several inflow configurations, and that it will be compared with previously introduced LES at a later point would inform the reader on the section's intention.

Page 12, lines 7-10 (also relevant for abstract and conclusions): The term wake deflection is also often used to describe a horizontal displacement of the whole wake in case of a wind turbine operating with a yaw offset. Here, wake deflection is used to describe the displacement of a part of the wake relative to the wind direction at hub height (other literature coined this a skewed wake – e.g. Abkar and Porté-Agel, 2016). A sentence that clarifies the usage of wake deflection in this paper or changing the term could avoid possible misunderstandings.

Page 19, lines 4-10: It might be beyond the scope of this parameter study, but for possible future studies, it would be interesting to investigate if the increased $\bar{u}_A$ for a veering wind and a CCR rotor holds for all possible locations of a hypothetical downstream turbine. The one-sided minima of the streamwise velocity in Fig. 6(b) and Fig. 6(d) that is just outside of the rotor area might (or might not) be canceling the positive effect on $\bar{u}_A$ and that could provide insights into the robustness of possible improvement for a hypothetical downstream turbine.

Page 32, lines 32: A similar comparison to Fig. 19 for the left/right sectors from Fig. 1 would be expected at this point. If nothing interesting was learned from it, it could be mentioned in a short sentence.

Fig. 20: The schematic illustration seem to be representative of a height away from hub height due to the wake deflection. Is it correct to assume that a similar schematic illustration for the hub height would have $\epsilon$ and $\Delta\epsilon$ equal to zero and the two $\Delta L_y$ would be symmetric?

**Technical comments**

Page 1, line 4: I believe "the" is missing before "wind veer".

Page 1, line 9: I believe "inflows" should be "inflow's".

Equation (6), (8), (15), (16), and (17): The cos, sin, exp, and tan functions should be written in upright font.

Page 35, line 1: I believe is should be "…has been observed in several…" instead of "…has be observed with…".

---

## Referee Report (RR2)

**Review of revised version (R1) Changing the rotational direction of a wind turbine under veering inflow: A parameter study, previously titled as *Should wind turbines rotate in the opposite direction?* by Antonia Englberger, Julie K. Lundquist, and Andreas Dörnbrack**

Reviewer: M. Paul van der Laan, DTU Wind Energy

The authors have rewritten the article significantly, and the new version has doubled in length. I like the added Sections 3 and 4, which help understanding the interaction of rotor rotation and wind veer. Most of my previous comments are addressed appropriately and I have listed the remaining comments below, including new comments related to the newly added content. My RANS simulations still predict the opposite trend as the present article, which I do not full understand. (It is not a must that my RANS simulations should predict the same trend as your LES in order to accept the article, because my RANS simulations could be wrong.) I have listed the main comments below.

**Main comments**

1. It is nice that you have added Section 3 and I now better understand your arguments. You mention that different magnitudes of wind veer or directional wind shear ($ds$) can result in different trends of wake deflection direction. In Figure 7, you show that for small $ds$ (0.04 and 0.08 °/m), a CCW rotating rotor has less wake deficit compared to CW rotating rotor (for the Northern Hemisphere). For large values of $ds$ (0.16 and 0.20 °/m), your simulations show the opposite. Both CW and CCW show a similar wake deficit for $ds = 0.12$ °/m. I have summarized your results below (for an aligned case on the Northern Hemisphere):

| $ds$ | 0.04 | 0.08 | 0.12 | 0.16 | 0.2 |
|---|---|---|---|---|---|
| Favored rotor rotation present article | CCW | CCW | similar | CW | CW |

I have performed two additional RANS cases with different values of $ds$: 0.028, 0.045, corresponding to turbulence intensity values of 4.5 and 4% at hub height, respectively. The results are plotted in Figures A1 and A2. I had previously simulated $ds = 0.095$. (Note that there is a misunderstanding regarding my RANS simulations from the first review round. I had used the NREL-5MW wind turbine, which has a rotor diameter of 126 m. The total wind veer of 12° then represents a wind veer of about 0.095°/ m.) I have also summarized the results below for the RANS simulations (for an aligned case on the Northern Hemisphere):

| $ds$ | 0.028 | 0.04 | 0.095 |
|---|---|---|---|
| Favored rotor rotation reviewer | similar | CW | CW |

I still get the opposite results compared your LES model. If I would try to further increase the amount of directional shear in the RANS model, then the resulting boundary layer height would become smaller than the wind turbine, and

then the RANS simulation would not converge because the wind turbine operates partly in the free atmosphere where the eddy-viscosity is very small. If I decrease the amount of directional shear then the effect of rotor rotation direction on the wake is negligible at 7D. To be fair, it could be that my RANS model predicts the wrong results and your LES simulations are correct.

The favored rotor rotation is also dependent on the relative wind direction, as shown in Figures A1 and A2. If your mean wind direction at the AD is slightly off because the inflow is developing downstream, then you could have post processed a two wind turbine case with a misaligned wind direction. I would therefore recommend strongly to also post process different wind direction cases, similarly to Figures A1 and A2. For example, you could add another figure for the cases listed in Fig. 7, taken at 7D, but as function of wind direction (or $y$), similar to my RANS figures, but then for wind speed not wind turbine power. If you have saved the time averaged 3D flow fields, then you should be able to just process your LES results to obtain such a plot.

2. I now understand that the inflow is based on a parametrization of LES precursor simulations, performed in previous work, and it is good that you have added more clarifications in the revised the article. I have looked at the articles where this method was introduced and I wonder how well the parametrized inflow is in balance with the LES model when the parameters are changed significantly. For example, there are quite large deviations of the fitted LES profiles (as shown in Fig 1. of your previous work Englberger and Dörnbrack (2018)). I am aware that this review should be focused on the current work, but it is important that your inflow profiles are in balance with the LES model, especially when the goal is look at relatively small differences caused by rotor rotation directions. If the inflow profiles are not in balance, then the wake deflections can also be caused by a downstream development of the inflow. On other hand, an LES inflow is never fully in balance with the 3D domain because of the transient nature of the model. So it could be that your parametrized inflow model is just as good (or bad) as using a traditional LES precursor inflow. It is very good that you have added a few lines in the conclusion, where you discuss the short comings and possible issues with using a parametrized LES inflow model compared to using a non-parametrized LES inflow model.

3. Section 2: You mention extremely small rotational frequencies of the rotor: 0.058 - 0.23 °/s, which would correspond to a time of 103 and 26 minutes for a full rotation, respectively. This does make sense to me. Are you sure these numbers are reported correctly? You could also report the corresponding tip speed ratios, which are more common to use in wind energy. A common tip speed ratio of MW-sized wind turbines below rated wind speeds is 7.5, so then you rotor rotational frequency would be 86 °/ s for a wind speed of 10 m/s.

4. I think you should clarify the point of view a bit better in the caption of Figures 3 and 4. Instead of *This picture is looking downwind on the wake* you could use: *This picture is looking from upwind towards downwind*.

5. Figure 7:

   – You could also normalize the wind speed by the rotor averaged wind speed taken from either the inflow, or a distance upstream of the AD or at the AD location without AD.

   – Should case *_ds18* be *_ds16*?

6. Section 4.3: You could mention that the main influence of wind speed is the wind turbine thrust coefficient $C_T$ through your AD controller. In addition, the shape of the mean inflow profile is dependent on the geostrophic wind speed, which most likely follows a Rossby similarity, see for example van der Laan et al. (2020b). (Without Coriolis and buoyancy related sources terms, your simulations should be Reynolds-number or wind speed independent for a fixed $C_T$, turbulence intensity and turbulence length scale, see for example van der Laan et al. (2020a)).

**Minor comments**

1. Figure 4 caption: ... the the ...

2. The x and y labels often include a variable and a unit that are separated by a divide or slash symbol. It would be more clear to use square brackets, for example: x [m].

**References**

Englberger, A. and Dörnbrack, A.: A Numerically Efficient Parametrization of Turbulent Wind-Turbine Flows for Different Thermal Strati-
fications, Boundary-Layer Meteorology, 169, 505–536, https://doi.org/10.1007/s10546-018-0377-z, 2018.

van der Laan, M. P., Andersen, S. J., Kelly, M., and Baungaard, M. C.: Fluid scaling laws of idealized wind farm simulations, J. Phys.: Conf.
Ser., 1618, 1–10, https://doi.org/10.1088/1742-6596/1618/6/062018, 2020a.

van der Laan, M. P., Kelly, M., Floors, R., and Peña, A.: Rossby number similarity of an atmospheric RANS model using limited-length-
scale turbulence closures extended to unstable stratification, Wind Energy Science, 5, 355–374, https://doi.org/10.5194/wes-5-355-2020,
https://wes.copernicus.org/articles/5/355/2020/, 2020b.

**Appendix A: Additional Reynolds-averaged Navier-Stokes simulations from reviewer**

[Figure]

**Figure A1.** Power of downstream wind turbine for clockwise and counter-clockwise rotor rotation using an atmospheric boundary inflow
with three different magnitudes of directional shear for the Northern Hemisphere.

[Figure]

**Figure A2.** Relative difference in power of downstream wind turbine for clockwise and counter-clockwise rotor rotation using an atmospheric boundary layer inflow with three different magnitudes of directional shear for the Northern Hemisphere.

---

## Author Response (AR2)

**Comments on the Review of Changing the rotational direction of a wind turbine under veering inflow: A parameter study - Reviewer 1**

Antonia Englberger[1], Julie K. Lundquist[2,3], and Andreas Dörnbrack[1]

[1]German Aerospace Center, Institute of Atmospheric Physics, Oberpfaffenhofen, Germany
[2]Department of Atmospheric and Oceanic Sciences, University of Colorado Boulder, Boulder, USA
[3]National Renewable Energy Laboratory, Golden, Colorado, USA

**Correspondence:** Antonia Englberger (antonia.englberger@dlr.de)

Dear Dr. M. Paul van der Laan,

Thank you for taking the time to carefully review our completely modified version of the original manuscript. We read your review in detail and appreciate you sharing your own simulation results. Here is our response to your comments:

**Main comments**

It is nice that you have added Section 3 and I now better understand your arguments. You mention that different magnitudes of wind veer or directional wind shear (ds) can result in different trends of wake deflection direction. In Figure 7, you show that for small ds (0.04 and 0.08° m$^{-1}$), a CCW rotating rotor has less wake deficit compared to CW rotating rotor (for the Northern Hemisphere). For large values of ds (0.16 and 0.20° m$^{-1}$), your simulations show the opposite. Both CW and CCW show a similar wake deficit for ds = 0.12° m$^{-1}$. I have summarized your results below (for an aligned case on the Northern Hemisphere):

I have performed two additional RANS cases with different values of ds: 0.028, 0.045, corresponding to turbulence intensity values of 4.5 and 4% at hub height, respectively. The results are plotted in Figures A1 and A2. I had previously simulated ds = 0.095. (Note that there is a misunderstanding regarding my RANS simulations from the first review round. I had used the NREL-5MW wind turbine, which has a rotor diameter of 126 m. The total wind veer of 12° then represents a wind veer of about 0.095° m$^{-1}$.) I have also summarized the results below for the RANS simulations (for an aligned case on the Northern Hemisphere):

Here is a misunderstanding. In the first manuscript version we apply CW for Clockwise Wake and CCW for CounterClockwise Wake. In the revised version we change to CR for a Clockwise Rotor rotation (corresponding to counterclockwise rotating wake CCW) and CCR for a CounterClockwise Rotor rotation (corresponding to clockwise rotating wake CW), as we thought it is more intuitive to think about the turbines rotation. Furthermore, we avoid the misunderstanding of interpreting CCW as CounterClockWise and CW as ClockWise. We are sorry if we caused any confusion here. So in your tables CCW should be CCR and CW should be CR.

| ds | 0.04 | 0.08 | 0.12 | 0.16 | 0.20 |
|---|---|---|---|---|---|
| EULAG favoured rotor rotation | CCR | CCR | similar | CR | CR |

Comparing this to your results:

| ds | 0.028 | 0.04 | 0.095 |
|---|---|---|---|
| Reviewers favoured rotor rotation | similar | CR | CR |

I still get the opposite results compared your LES model. If I would try to further increase the amount of directional shear in the RANS model, then the resulting boundary layer height would become smaller than the wind turbine, and then the RANS simulation would not converge because the wind turbine operates partly in the free atmosphere where the eddy-viscosity is very small. If I decrease the amount of directional shear then the effect of rotor rotation direction on the wake is negligible at 7D. To be fair, it could be that my RANS model predicts the wrong results and your LES simulations are correct.

Probably there are other influence factors between EULAG and your RANS simulation, leading to a deviation of your results from the prediction of the theoretical analysis and the EULAG results in our manuscript. E.g. looking at the equations in your papers, you apply external forces on the right hand side. Dependent on your boundary conditions etc. they could result in a large scale pressure gradient, which affects the meridional velocity component. This could explain the deviation from the theoretical analysis and the EULAG results. Here, we think further personal discussion between the Reviewer and the authors is necessary.

Further, the difference between clockwise and counterclockwise rotating actuators should not depend on the applied parametrization in this work. In a previous paper Englberger et al. (2019) (which will be published soon), we investigated the rotational direction impact for an SBL case, applying 2D slices of $u$, $v$, $w$, and $\Theta$ at each time step, which result from a precursor SBL LES. In Englberger et al. (2019), the wind veer is limited to the lower rotor half with $0.28°$ m$^{-1}$ between 10 m and 115 m. In this case, if the actuator rotates counterclockwise the wake width and the wake deflection angle are larger and the maximum velocity deficit is smaller (more rapid wake recovery downstream). Therefore, the preferred rotational direction of the rotor in this study is counterclockwise, as it results in a higher downstream $u$-value up to at least 10 D. Comparing the directional shear value to this study, also the vertical extend of veering inflow matters. Different conditions of the mean wind field could also lead to differences in the directional shear value, at which both wakes behave similar regarding their velocity deficit.

The favored rotor rotation is also dependent on the relative wind direction, as shown in Figures A1 and A2. If your mean wind direction at the AD is slightly off because the inflow is developing downstream, then you could have post processed a two wind turbine case with a misaligned wind direction. I would therefore recommend strongly to also post process different wind direction cases, similarly to Figures A1 and A2. For example, you could add another figure for the cases listed in Fig. 7, taken at 7D, but as function of wind direction (or y), similar to my RANS figures, but then for wind speed not wind turbine power. If you have saved the time averaged 3D flow fields, then you should be able to just process your LES results to obtain such a plot.

In our simulations, we apply a homogeneous surface and prescribe the background wind field, which is only modified by turbulence (and the WT itself). Further, we do not apply any additional external forcing acting as large scale pressure gradient and therefore the wind direction in front of the wind turbine is constant in each simulation with $\overline{v(z_h)}_{time} = 0$ (see Eq. 8). This

allows us to only reproduce your plot for 270° (see Fig. 7).

2. I now understand that the inflow is based on a parametrization of LES precursor simulations, performed in previous work, and it is good that you have added more clarifications in the revised the article. I have looked at the articles where this method was introduced and I wonder how well the parametrized inflow is in balance with the LES model when the parameters are changed significantly. For example, there are quite large deviations of the fitted LES profiles (as shown in Fig 1. of your previous work Englberger and Dörnbrack (2018)). I am aware that this review should be focused on the current work, but it is important that your inflow profiles are in balance with the LES model, especially when the goal is look at relatively small differences caused by rotor rotation directions. If the inflow profiles are not in balance, then the wake deflections can also be caused by a downstream development of the inflow. On other hand, an LES inflow is never fully in balance with the 3D domain because of the transient nature of the model. So it could be that your parametrized inflow model is just as good (or bad) as using a traditional LES precursor inflow. It is very good that you have added a few lines in the conclusion, where you discuss the short comings and possible issues with using a parametrized LES inflow model compared to using a non-parametrized LES inflow model.

The solution of the LES inside the area is disturbed by the 2 D inflow profiles of $u$, $v$, and $w$ that change over time due to the applied turbulence parametrization. The pressure solver ensures that the solution is in equilibrium. To verify this, I repeated the reference simulation CR, however, applying the inflow perturbations only every twentieth time step. Comparing the residual divergence (domain min, max, average), the difference after 10 min is rather small.

| Simulation | $div_{av}$ |
|---|---|
| reference CR | -0.4e-8 |
| 20th dt CR | -0.8e-8 |

3. Section 2: You mention extremely small rotational frequencies of the rotor: 0.058 - 0.23 ° /s, which would correspond to a time of 103 and 26 minutes for a full rotation, respectively. This does make sense to me. Are you sure these numbers are reported correctly? You could also report the corresponding tip speed ratios, which are more common to use in wind energy. A common tip speed ratio of MW-sized wind turbines below rated wind speeds is 7.5, so then you rotor rotational frequency would be 86 ∘ / s for a wind speed of 10 m/s.

Thank you very much for detecting this typo. It should be only 0.058 - 0.23 $s^{-1}$, which is the number of revolutions per second. We apply 7 revolutions per minute for the reference case, resulting in 0.117 revolutions per second.

4. I think you should clarify the point of view a bit better in the caption of Figures 3 and 4. Instead of This picture is looking downwind on the wake you could use: This picture is looking from upwind towards downwind.

We added it. Further, the way of looking from upwind, downwind on the rotor is explained in Fig. 1 and section 2.3.

5. Figure 7:
You could also normalize the wind speed by the rotor averaged wind speed taken from either the inflow, or a distance upstream

of the AD or at the AD location without AD.

As $\overline{u_{inflow}}$ is the same in case of CCR and CR, the plot itself only modifies for part (b) (Fig. 1). We agree, for part (b) the decrease of the difference between clockwise and counterclockwise is better represented by plotting the normalized values instead of absolute ones. However, we decide not to use this plot, as we discuss absolute velocity values in the wake in the paper in the other figures and Fig. 7 is the summary of all other ones. Therefore, the explanation of the differences is more straight forward applying Fig. 7 with absolute values instead of normalized ones and the text is also based on absolute figures.

Should case _ds18 be _ds16?

Thank you! It was a typo, we changed it.

Section 4.3: You could mention that the main influence of wind speed is the wind turbine thrust coefficient $C_T$ through your AD controller. In addition, the shape of the mean inflow profile is dependent on the geostrophic wind speed, which most likely follows a Rossby similarity, see for example van der Laan et al. (2020b). (Without Coriolis and buoyancy related sources terms, your simulations should be Reynolds-number or wind speed independent for a fixed $C_T$ , turbulence intensity and turbulence length scale, see for example van der Laan et al. (2020a)).

We see your argument. Our attempt with increasing the wind speed, however, was to change the atmospheric inflow. Therefor, we decide not to include this comment.

[Figure]

**Figure 1.** The rotor and time averaged streamwise velocity $\overline{u_{A_{wake}}}$ normalized by the rotor and time averaged streamwise velocity at the inflow $\overline{u_{A_{inflow}}}$ presented for a downwind region of $[4\,D; 10\,D]$ with special emphasis at $x = 7\,D$ for the simulations CR_NV, CCR_NV, CR, CCR, CR_b and CCR_b in (a), for different geostrophic wind values in (b), for different directional shears in (c), and for different rotational frequencies in (d).

**References**

Englberger, A., Dörnbrack, A., and Lundquist, J. K.: Does the rotational direction of a wind turbine impact the wake in a stably stratified atmospheric boundary layer?, Wind Energy Science Discussions, 2019, 1–24, https://doi.org/10.5194/wes-2019-45, https://www.wind-energ-sci-discuss.net/wes-2019-45/, 2019.

**Comments on the Review of Changing the rotational direction of a wind turbine under veering inflow: A parameter study - Reviewer 2**

Antonia Englberger[1], Julie K. Lundquist[2,3], and Andreas Dörnbrack[1]

[1]German Aerospace Center, Institute of Atmospheric Physics, Oberpfaffenhofen, Germany
[2]Department of Atmospheric and Oceanic Sciences, University of Colorado Boulder, Boulder, USA
[3]National Renewable Energy Laboratory, Golden, Colorado, USA

**Correspondence:** Antonia Englberger (antonia.englberger@dlr.de)

Dear Reviewer 2,

Thank you for taking the time to carefully review our completely modified version of the paper. We are happy to hear that you think it is a very compelling research question. We read your review in detail. Here is our response to your comments:

**Minor comments**

In Eq. (15) and (16) the , and in Eq. (17) x WT and and x  are not introduced.

Thank you for noticing. We included the explanations.

Page 8, lines 2-3: An addition to this sentence, that the simple model demonstrates the principle interactions between an idealized wake represented by a vortex and several inflow configurations, and that it will be compared with previously introduced LES at a later point would inform the reader on the section's intention.

We added your first suggestion in: 'In this section, we construct a simple analytical model of the interaction of the rotating wake of a wind turbine with a sheared ambient flow. The rotating wake is prescribed by a Rankine vortex. The ambient flow is described by three different inflow conditions (no veer, veering wind, backing wind). In the case of a veering inflow, the relations are also evaluated for three different parameters (wind speed, directional shear, rotational velocity).'

Your second suggestion is listed in the end of the introduction giving the main overview of the idea of the manuscript. Therefore, we did not list it in section 3.

Page 12, lines 7-10 (also relevant for abstract and conclusions): The term wake deflection is also often used to describe a horizontal displacement of the whole wake in case of a wind turbine operating with a yaw offset. Here, wake deflection is used to describe the displacement of a part of the wake relative to the wind direction at hub height (other literature coined this a skewed wake – e.g. Abkar and Porté-Agel, 2016). A sentence that clarifies the usage of wake deflection in this paper or changing the term could avoid possible misunderstandings.

We use skewed wake structure for describing the wake structure difference in the y-z plane between veering and no-veering

inflow in Fig. 5. Applying 'wake deflection to the left/right', we think it is more telling in comparison to skewed wake. However, we understand your point to get confused with yaw offset. Therefore, we added an explanation the first time we use wake deflection in the text. '... wind veer causes wake deflection in relation to a vertical plan through the nacelle at $y=0$ ...'. Now the meaning of wake deflection in this manuscript should be clear.

Page 19, lines 4-10: It might be beyond the scope of this parameter study, but for possible future studies, it would be interesting to investigate if the increased $\overline{u_A}$ for a veering wind and a CCR rotor holds for all possible locations of a hypothetical downstream turbine. The one-sided minima of the streamwise velocity in Fig. 6(b) and Fig. 6(d) that is just outside of the rotor area might (or might not) be canceling the positive effect on $\overline{u_A}$ and that could provide insights into the robustness of possible improvement for a hypothetical downstream turbine.

We get a similar comment from Reviewer 1 on the first manuscript version. In the response to him we added a plot (Fig. 1) investigating this. However, as the manuscript is already very long, we decide to not include this discussion in the manuscript. But we will refer to it in subsequent work.

Considering the horizontal profiles in the lower and the upper rotor half at $z=75$ m and at $z=125$ m in Figs. 5, 6, and 7, the wake is deflected in the lower rotor part towards the left (right) and in the upper rotor part towards the right (left) in case of veering (backing) inflow. As the lateral wake position depends on the inflow wind angle, the spanwise wake position approaches away from $y=0$ D for increasing directional shear. This is presented in Fig. 1 (here). In case of no wind veer (Fig. 1(b)), there is no difference. In case of veering inflow (Fig. 1(e)), at $y=-1/2$ D, there is a small rotational direction difference in the bottom rotor part, and at $y=1/2$ D, there is a difference in the top rotor part. The top left ($y<0$) and the bottom right rotor parts are unaffected by the rotational direction, as there is no wake in these sectors. In case of a backing wind (Fig. 1(h)) the situation is the opposite. In case of a veering inflow, increasing the geostropic wind (Fig. 1(c)) or the directional shear (Fig. 1(f)) increases the difference in $\overline{u}$, especially in the top right rotor part. The same is valid for an increase of the rotational frequency (Fig. 1(i)). Decreasing the atmospheric or vortex strength, the difference decreases. Therefore, there is an impact at $y=0$ D and likewise in the wake affected sectors to the right or the left. In the considered idealized simulations of this work, the impact on $\overline{u}$ has therefore the same tendency in case of staggered or unstaggered arrangements of the hypothetical downwind turbines.

Page 32, lines 32: A similar comparison to Fig. 19 for the left/right sectors from Fig. 1 would be expected at this point. If nothing interesting was learned from it, it could be mentioned in a short sentence.

The right/left comparison plot (Fig. 2 (here) does not provide any new insight. We mention the much less distinct difference between clockwise and counterclockwise rotating actuators for the left and right sectors in comparison to the top and bottom ones.

Fig. 20: The schematic illustration seem to be representative of a height away from hub height due to the wake deflection. Is it correct to assume that a similar schematic illustration for the hub height would have $\epsilon$ and $\Delta\epsilon$ equal to zero and the two $\Delta L_y$ would be symmetric?

[Figure]

**Figure 1.** Sector averages of $\overline{u}$ representing the top and bottom 90°-sectors for 0 m $< r \leq$ 50 m for clockwise and counterclockwise rotating actuators for the same simulations as in Fig. 19 of the manuscript at $y = 0$ D and in addition shifted by D/2 in both lateral directions. The indices 'b' and 't' at the top x-axis represent the corresponding bottom or top sectors.

At hub height, $\epsilon = \Delta\epsilon = 0$, however, $\Delta L_y \neq 0$ all over the rotor height. $L_y$ of the counterclockwise rotating rotor will be larger in comparison to $L_y$ of a clockwise rotating rotor also at hub height (see Fig. 9c vs. f). We added this in the caption of Fig. 20.

[Figure]

**Figure 2.** Sector averages of $\overline{u}$ representing the left and right $90°$-sectors for $0\,\mathrm{m} < r \le 50\,\mathrm{m}$ for clockwise and counterclockwise rotating actuators for the same simulations as in Fig. 18 (manuscript).